



**1    Modelling soil CO₂ production and transport with dynamic source and diffusion terms:**

**2          Testing the steady-state assumption using DETECT v1.0**

4          Edmund Ryan[1,2]*, Kiona Ogle[2,3,4,5], Heather Kropp[6], Kimberly E. Samuels-Crow[3],

5                        Yolima Carrillo[7], Elise Pendall[7]

[1]Lancaster Environment Centre, Lancaster University, Lancaster, UK
[2]School of Life Sciences, Arizona State University, Tempe, Arizona, USA
[3]School of Informatics, Computing, and Cyber Systems, Northern Arizona University, Flagstaff,
Arizona, USA
[4]Center for Ecosystem Science and Society, Northern Arizona University, Flagstaff, Arizona,
USA
[5]Department of Biological Sciences, Northern Arizona University, Flagstaff, Arizona, USA
[6]Department of Geography, Colgate University, Hamilton, NY, USA
[7]Hawkesbury Institute for the Environment, Western Sydney University, NSW, Australia
*Corresponding author:
Lancaster Environment Centre,
Lancaster,
Lancashire. LA1 4YW
United Kingdom.
Tel: +44 (0)1524 594009
Email: edmund.ryan@lancaster.ac.uk





**Abstract**
The flux of $CO_2$ from the soil to the atmosphere (soil respiration, $R_{soil}$) is a major component of
the global carbon cycle.  Methods to measure and model $R_{soil}$, or partition it into different
components, often rely on the assumption that soil $CO_2$ concentrations and fluxes are in steady
state, implying that $R_{soil}$ is equal to the rate at which $CO_2$ is produced by soil microbial and root
respiration.  Recent research, however, questions the validity of this assumption.  Thus, the aim
of this work was two-fold: (1) to describe a non-steady state (NSS) soil $CO_2$ transport and
production model, DETECT, and (2) to use this model to evaluate the environmental conditions
under which $R_{soil}$ and $CO_2$ production are likely in NSS. The backbone of DETECT is a non-
homogeneous, partial differential equation (PDE) that describes production and transport of soil
$CO_2$, which we solve numerically at fine spatial and temporal resolution (e.g., 0.01 m increments
to 1 m, every 6 hours). Production of soil $CO_2$ is simulated for every depth and time increment as
the sum of root respiration and microbial decomposition of soil organic matter, both of which
can be driven by current and antecedent soil water content and temperature, which can also vary
by time and depth. We also analytically solved the ordinary differential equation (ODE)
corresponding to the steady-state (SS) solution to the PDE model. We applied the DETECT NSS
and SS models to the 6-month growing season period representative of a native grassland in
Wyoming. Simulation experiments were conducted with both model versions to evaluate factors
that could affect departure from SS: (1) varying soil texture; (2) shifting the timing or frequency
of precipitation; and (3) with and without the environmental antecedent drivers.  For a coarse-
textured soil, $R_{soil}$ from the SS model closely matched that of the NSS model.  However, in a
fine-textured (clay) soil, growing season $R_{soil}$ was ~3% higher under the assumption of NSS
(versus SS).  These differences were exaggerated in clay soil at daily time-scales whereby $R_{soil}$





under the SS assumption deviated from NSS by up to ~20% in the 10 days following a major
precipitation event. Moreover, incorporation of antecedent drivers increased the magnitude of
$R_{soil}$ by 15% to 37% for coarse- and fine-textured soils, respectively.  However, the responses of
$R_{soil}$ to the timing of precipitation and antecedent drivers did not differ between SS and NSS
assumptions.  In summary, the assumption of SS conditions can be violated depending on soil
type and soil moisture status, as affected by precipitation inputs, and the DETECT model
provides a framework for accommodating NSS conditions to better predict $R_{soil}$ and associated
soil carbon cycling processes.
*Keywords*: antecedent soil water content, DETECT, diffusion model, modelling soil $CO_2$, non-
steady-state, precipitation frequency, soil respiration, soil texture, steady-state.

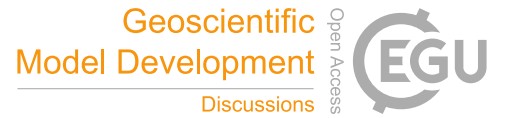



## 1. Introduction

The flux of $CO_2$ to the atmosphere from the soil (i.e., soil respiration, $R_{soil}$) is one of the largest

fluxes in the global C cycle and is approximately ten times the annual amount of $CO_2$ emitted by

fossil fuel burning (Friedlingstein et al., 2014; Hashimoto et al., 2015).  Moreover, global change

experiments and predictions from models agree that $R_{soil}$ is expected to increase in a future

climate of elevated $CO_2$ and warming (Cox, 2001; Davidson and Janssens, 2006; Pendall et al.,

2013; Piao et al., 2009; Ryan et al., 2015). Therefore, monitoring $R_{soil}$ is important for

quantifying and modeling the global C cycle.

Commonly, $R_{soil}$ is monitored by directly measuring surface soil $CO_2$ fluxes using various

chamber methods (Luo and Zhou, 2010; Risk et al., 2011) or by estimating $R_{soil}$ from soil $CO_2$

concentrations measured at multiple depths using probe methods (Pendall et al., 2003; Tang et

al., 2003; Vargas et al., 2010).  The probe methods employ diffusion equations that often rely on

the assumption that $R_{soil}$ at the surface is in steady state (SS) with subsurface $CO_2$ production by

roots and micro-organisms (Baldocchi et al., 2006; Lee et al., 2004; Luo and Zhou, 2010;

Šimůnek et al., 2012; Tang et al., 2003; Vargas et al., 2010).  That is, the SS assumption

essentially assumes that $CO_2$ produced by roots and microbes within the soil profile is

instantaneously respired from the soil surface, effectively neglecting delays due to $CO_2$ transport

times.  Partitioning $R_{soil}$ (surface flux) into its different components (e.g., sub-surface

heterotrophic [microbes] versus autotrophic [root or rhizosphere] respiration) using isotope

methods (Hui and Luo, 2004; Ogle and Pendall, 2015), trenching methods (Šimůnek and Suarez,

1993), or soil $CO_2$ models (Vargas et al., 2010) also relies on the SS assumption. Even

simulations of the vertical movement of soil $CO_2$ through snow have employed a SS diffusion

model (Monson et al. 2006). Recent work, however, calls into question whether this SS





assumption is valid most of the time or in most systems (Maggi and Riley, 2009; Nickerson and
Risk, 2009).

3        Given the use of the SS assumption in a diverse range of settings, the aim of this study

was to determine the meteorological and site specific conditions under which the SS assumption
is valid, and the circumstances under which a non-steady state (NSS) model substantially
improves our understanding of subsurface processes that lead to observed $R_{soil}$. We focused on
soil texture because it is a critical factor underlying soil porosity and tortuosity, which, in turn,
control soil $CO_2$ diffusion rates (Bouma and Bryla, 2000). For example, coarse (e.g., high sand
content) soils generally facilitate fast diffusion rates, especially under low soil moisture
conditions associated with high air-filled porosity (Bouma and Bryla, 2000); the opposite is
expected for finer-grained (e.g., silt or clay) soils. Thus, we expected coarse soils to generally
induce SS conditions for soil $CO_2$, whereas fine-grained soils would likely produce frequent and
longer duration NSS conditions, especially following rain pulses that decrease air-filled porosity,
thereby reducing $CO_2$ diffusivity.

15       We also focused on the impacts of precipitation variability given that the timing and

magnitude of precipitation pulses can have large effects on $R_{soil}$ (Borken and Matzner, 2009;
Cable et al., 2008; Huxman et al., 2004; Ogle et al., 2015; Schwinning et al., 2004; Sponseller,
2007). Precipitation indirectly impacts $R_{soil}$ via its influence on soil moisture dynamics, and soil
moisture and soil texture affect both diffusivity (physical process) and $CO_2$ production (primarily
biological process governed by roots and microbes). Moreover, as precipitation pulses infiltrate
the soil, filling of pore spaces with water can result in a displacement of $CO_2$, which may be seen
as a transient spike in $R_{soil}$ (e.g., Lee et al., 2004). Such transient spikes, however, may also be
attributable to changes in decomposition, microbial growth, and/or C substrate availability in





response to wetting (Birch, 1958; Borken et al., 2003; Jarvis et al., 2007; Meisner et al., 2013;
Xiang et al., 2008). This transient response may be followed by a depression in $R_{soil}$ since water-
filled pores will ultimately slow $CO_2$ diffusion and transport (Bouma and Bryla, 2000).  These
linked effects imply that precipitation pulses and their effects on soil moisture are likely to
impose NSS soil $CO_2$ conditions, but the manner in which such pulses impact these processes is
temporally dynamic and spatially complex, and therefore difficult to measure directly.

7         We evaluated the importance of soil texture and precipitation variability on SS versus

NSS soil $CO_2$ behavior via a simulation-based approach. To allow for the possibility of both SS
and NSS behavior, we implemented a depth- and time-varying $CO_2$ transport and production
model that builds on the groundbreaking work of Fang and Moncrieff (1999), Hui and Luo
(2004), Nickerson and Risk (2009), Moyes et al. (2010) and Risk et al. (2012).  These processes
are captured by a partial differential equation (PDE) model that is grounded in diffusion theory,
and solved numerically. Some current NSS models make simplifying assumptions such as
assuming depth-invariant production rates (e.g., Fang and Moncrieff, 1999), or assuming that
production only responds to concurrent environmental conditions (e.g., Nickerson and Risk,
2009). Such simplifications may make it difficult to evaluate physical and biological conditions
leading to SS versus NSS behavior.

18        We addressed the aforementioned shortcomings of existing NSS models with the

DETECT (DEconvolution of Temporally varying Ecosystem Carbon componenTs) model,
version 1.0 (v1.0), which implemented four improvements. First, we simulated soil $CO_2$ at 0.01
m increments—from the surface to a depth of 1 m—to ensure numerical accuracy of the
solutions (Haberman, 1998). Second, we drove the PDE model with output from a dynamic,
physically-based model of soil water and temperature in porous media (HYDRUS; Simunek et



al., 2005; Šimůnek et al., 2008) that predicts sub-daily soil environmental conditions in each 0.01
m increment. Third, we simulated the production of $CO_2$ by microbial and root respiration in
each increment by linking these processes to the depth-specific soil conditions using existing
respiration models that are typically applied to "bulk" soil (Cable et al., 2008; Davidson et al.,
2012; Lloyd and Taylor, 1994; Todd-Brown et al., 2012).  Fourth, a growing body of evidence
suggests that antecedent or past environmental or meteorological conditions are important for
predicting soil and ecosystem $CO_2$ fluxes (Barron-Gafford et al., 2014; Cable et al., 2013; Ryan
et al., 2015), and we accounted for these antecedent effects in the respiration (or production)
submodels. For example, observed increases in ecosystem and soil respiration rates after a rain
event are generally greater if the rain event occurs during a dry period compared to a wet period
(Cable et al., 2013; Cable et al., 2008; Sponseller, 2007; Thomas et al., 2008; Xu et al., 2004).
Such antecedent effects may underlie the importance of biological versus physical processes in
governing the transition between SS and NSS behavior.
After describing the DETECT model, we subsequently use it to explore the effects of soil
texture, precipitation pulses, and antecedent conditions on the relative importance of NSS soil
$CO_2$ behavior, and to identify the factors giving rise to such behavior. We simulated soil $CO_2$
concentrations, $CO_2$ production, and $R_{soil}$ under four different soil textures and three different
precipitation regimes. For each scenario, we implemented the DETECT model under the
assumption that soil $CO_2$ production is affected by antecedent moisture and temperature versus
the assumption that only concurrent conditions matter. Data from the Wyoming Prairie Heating
and $CO_2$ Enrichment (PHACE) experiment (e.g., Carrillo et al., 2014a; Mueller et al., 2016;
Pendall et al., 2013; Ryan et al., 2015; Zelikova et al., 2015) were used to parameterize the
model and motivated the selection of the texture and precipitation scenarios.  Under the different





scenarios, we compared $R_{soil}$ and $CO_2$ production rates predicted from the DETECT model to
that of a simpler SS model, and evaluated the relative impact of SS assumptions on inferring
subsurface processes (e.g., $CO_2$ production by roots and microbes) and surface $CO_2$ fluxes (i.e.,
$R_{soil}$).

## 2. Methods

Our DETECT model simulates $CO_2$ production by roots and microbes at 100 0.01-m depth
intervals at 6-hourly time increments, but it is flexible enough to accommodate finer or coarser
depth and time intervals. The backbone of DETECT is a partial differential equation (PDE)
model motivated by diffusion theory, and solved numerically.  In this study, we used the
DETECT model to simulate time- and depth-varying soil $CO_2$ concentration and $CO_2$ production
rates and time-varying surface soil $CO_2$ efflux ($R_{soil}$) over 183 days demarking the growing
season in a mixed-grass prairie in south-central Wyoming. The model is driven by environmental
data collected at the Prairie Heating and $CO_2$ Enrichment (PHACE) experiment near Cheyenne,
Wyoming (Pendall et al., 2013). PHACE data were also used to derive realistic parameter values
associated with the submodels for $CO_2$ production and diffusivity, and additional data were used
to qualitatively evaluate model predictions.

17       In this section, we describe key components of the DETECT model (section 2.1), the

numerical solution approach (section 2.2), and the SS solution to the underlying PDE (section
2.3). We follow this by a description of how we parameterized the model to be representative of
a "real" system by drawing upon information from the PHACE experiment (section 2.4). This
leads us to provide an overview of the PHACE experiment and relevant datasets used to
parameterize, drive, and informally test the model. We conclude this methods section by
outlining the simulation experiments that we conducted with the DETECT model and its SS





counterpart to evaluate the influence of soil texture and precipitation variability on SS versus
NSS soil $CO_2$ behavior (section 2.5).
**2.1 Description of the Non Steady State DETECT Model**
The PDE that underlies the DETECT model (v1.0) accounts for time- and depth-varying $CO_2$
diffusivity and $CO_2$ production by root and microbial respiration (Fang & Moncrieff, 1999). We
use a pair of PDEs, one describing the soil $CO_2$ derived from root respiration (subscripted with
$R$), and the other for $CO_2$ derived from microbial respiration ($M$) such that for $K = R$ or $M$:

$$\frac{\partial c_K(z,t)}{\partial t} = \frac{\partial}{\partial z}\left(D_{gs}(z,t)\frac{\partial c_K(z,t)}{\partial z}\right) + S_K(z,t) \tag{1}$$

$c_K(z,t)$ is $CO_2$ concentration (mg $CO_2$ m$^{-3}$), $D_{gs}(z,t)$ is the effective diffusivity of $CO_2$ through the
soil (m$^2$ s$^{-1}$), and $S_K(z,t)$ is the source (or production) term (mg $CO_2$ m$^{-3}$) (Fig. 1b), all of which
vary by depth $z$ (meters) and time $t$ (hours). Note that $D_{gs}$ is assumed to be the same for root- and
microbial-derived $CO_2$ and is thus not indexed by $K$. In this version of the model, we assumed
that $CO_2$ transport within the soil profile and over time is solely governed by gaseous diffusion,
and we ignored other types of $CO_2$ transport—such as diffusion in the liquid state, convection,
and bulk transport via vertical movement of water—that have been shown to have a negligible
contribution (Fang and Moncrieff, 1999; Kayler et al., 2010). Total soil $CO_2$ and total $CO_2$
production are given as $c(z,t) = c_M(z,t) + c_R(z,t)$ and $S(z,t) = S_M(z,t) + S_R(z,t)$, respectively. Below
we describe the two main components of the PDE model: (1) $CO_2$ diffusivity, $D_{gs}$, and (2) the
production terms, $S_R(z,t)$ and $S_M(z,t)$.
*2.1.1 Soil $CO_2$ diffusivity sub-model*
The diffusivity of $CO_2$ within the soil ($D_{gs}$) depends on soil structure and water content; we





modeled $D_{gs}$ using the Moldrup function (Moldrup et al., 2004; Sala et al., 1992). We chose this
formulation because it is more accurate than other common models, such as the Millington and
Quirk (2000) and Penman (1981) models (Moldrup et al., 2004). Based on Moldrup et al. (2004),
$D_{gs}$ (m$^2$ s$^{-1}$) is defined as:

$$D_{gs}(z,t) = D_{g0}(z,t) \cdot \left( 2\phi_{g100}(z)^3 + 0.04\phi_{g100}(z) \right) \cdot \left( \frac{\phi_g(z,t)}{\phi_{g100}(z)} \right)^{2+\frac{3}{b(z)}} , \qquad (2)$$

where $D_{g0}(z,t) = D_{stp} \cdot \left( \dfrac{T_s(z,t)}{T_0} \right)^{1.75} \cdot \left( \dfrac{P_0}{P(t)} \right)$ and $D_{stp} = 1.39 \times 10^{-5}$ m$^2$ s$^{-1}$ is the diffusion
coefficient for $CO_2$ in air at standard temperature ($T_0$, 273 K) and pressure ($P_0$, 101.325 kPa);
$T_s(z,t)$ is the soil temperature (Kelvin) at depth $z$ and time $t$, and $P(t)$ is the air pressure (kPa) just
above the soil surface at time $t$. The remaining terms in Eqn 2 include $\phi_g(z,t)$, the air-filled soil
porosity, which is related to the total soil porosity ($\phi_T$) and volumetric soil water content ($\theta$)
according to $\phi_g(z,t) = \phi_T(z) - \theta(z,t)$, and $\phi_T(z)$ is defined as $1 - BD(z)/PD$, where BD and PD are
the bulk density and particle density of the soil, respectively; $\phi_{g100}(z)$ is the air-filled porosity at a
soil water potential ($\Psi$) of -100 cm H$_2$O (about -10 kPa); $b(z)$ is a unitless parameter that is
related to the pore size distribution of the soil based on the water retention curve given by $\Psi =$
$\Psi_e(\theta/\theta_{sat})^{-b}$, where $\Psi_e(z)$ is the air-entry potential and $\theta_{sat}(z)$ is the saturated soil water content
(v/v).
*2.1.2 CO$_2$ source (production) terms*
Soil $CO_2$ can be produced in the soil ($S$ term in Eqn. 1) by five different biological processes: (i)
root growth respiration, (ii) root maintenance respiration, (iii) consumption of rhizodeposits by
root-associated microorganisms and associated microbial respiration, (iv) microbial

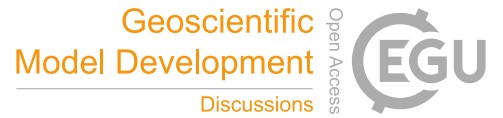



decomposition of newly produced plant litter that has been incorporated into the soil matrix, and
(v) microbial decomposition of older soil organic matter (SOM) (Pendall et al., 2004). Due to the
general lack of sufficient data and process understanding to accurately separate all five sources,
the DETECT model treats $CO_2$ production as the sum of two main contributions: $CO_2$ respired
by (1) roots and closely associated microorganisms (the sum of (i)-(iii)), giving $S_R(z,t)$, and (2)
free-living soil microorganisms (the sum of (iv)-(v)), giving $S_M(z,t)$. Such simplification based on
root and microbial sources is common in models of soil $CO_2$ transport and production (Fang and
Moncrieff, 1999; Hui and Luo, 2004; Šimůnek and Suarez, 1993).  Although DETECT v1.0
assumes that root and microbial respiration are independent of one another, they both depend on
the same environmental data (e.g., $\theta$ and $T_s$).

11        $CO_2$ production by root respiration is represented as the product of three terms: (i) the

mass-specific base respiration rate ($R_{Rbase}$) at a reference soil temperature of $T_s = T_{ref}$, and at
average soil water and antecedent temperature conditions, (ii) root mass expressed as the amount
of root carbon, $C_R(z,t)$, and (iii) functions that rescale $R_{Rbase}$ to account for the effect of soil water
($\theta$), temperature ($T_s$), and their antecedent counterparts, which are determined separately for
roots and microbes. For roots, antecedent soil water and temperature are denoted as $\theta_R^{ant}$ and
$T_s^{ant}$, respectively. In general, $S_R(z,t)$ is given by:

$$S_R(z,t) = R_{Rbase} \cdot C_R(z,t) \cdot f(\theta(z,t), \theta_R^{ant}(z,t)) \cdot g(T_S(z,t), T_S^{ant}(z,t)) \qquad (3)$$

The functional form of $C_R(z,t)$ is informed by field data on root biomass C (see Appendix S1 for
complete details).  The functions $f$ and $g$ are given by:

$$f\left(\theta, \theta_R^{ant}\right) = \exp\left(\alpha_1 \theta(z,t) + \alpha_2 \theta_R^{ant}(z,t) + \alpha_3 \theta(z,t) \cdot \theta_R^{ant}(z,t)\right) \qquad (4a)$$

$$g\left(T_S, T_R^{ant}\right) = \exp\left( E_o(z,t) \cdot \left( \frac{1}{T_{ref} - T_o} - \frac{1}{T_S(z,t) - T_o} \right) \right) \qquad (4b)$$





$$E_o(z,t) = E_o^* + \alpha_4 T_S^{ant}(z,t) \qquad (4c)$$

$\alpha_1$, $\alpha_2$, $\alpha_3$, $\alpha_4$, $T_o$, and $E_o^*$ are parameters that require numerical values (Table 1; Ryan et al.
2015), $\theta$ and $T_s$ are informed by field data, and $\theta_R^{ant}$ and $T_s^{ant}$ are computed from the field data
(described below). The temperature scaling function, $g$ (Eqn 4b), is motivated by Lloyd and
Taylor (1994) and has been successfully used to describe soil and ecosystem respiration (Cable
et al., 2013; Luo and Zhou, 2010; Ryan et al., 2015). $E_o(z,t)$ is analogous to an energy of
activation term that governs the apparent temperature sensitivity of $S_R$ (Cable et al., 2011;
Davidson and Janssens, 2006; Tucker et al., 2013); we assume $E_o$ responds to antecedent
temperature, reflecting a potential thermal acclimation response (Atkin and Tjoelker, 2003; Ryan
et al., 2015). $T_o$ is also related to the apparent temperature sensitivity (Cable et al., 2011), and
we assume that it is invariant with depth and time (Barron-Gafford et al., 2014; Cable et al.,
2013; Lloyd and Taylor, 1994; Ryan et al., 2015). An exponential function is also used for the
moisture scalar, $f$, to ensure $f > 0$ (Eqn 4a). While the functional forms and choice of
environmental drivers used for $f$ and $g$ were motivated by previous analyses (Barron-Gafford et
al., 2014; Cable et al., 2013), the exact functions and parameter values were based on Ryan et al.
(2015) and Cable et al. (2013).
CO$_2$ production by microbial respiration and SOM decomposition is represented by a
modified version of the Dual Arrhenius and Michaelis-Menten (DAMM) model (Davidson et al.,
2012). We exclude the O$_2$ term, rendering the model relevant to systems that are typically
unlimited by O$_2$ availability, such as the semi-arid site that we focus on, but we accounted for a
microbial C pool ($C_{MIC}$) and a soluble soil-C pool ($C_{SOL}$) (Todd-Brown et al., 2012) such that:

$$S_M(z,t) = V_{max}(z,t) \cdot \frac{C_{SOL}(z,t)}{K_m + C_{SOL}(z,t)} \cdot C_{MIC}(z,t) \cdot (1 - CUE) \qquad (5)$$





Decomposition is assumed to be an enzymatic process that follows Michaelis-Menten kinetics,
where $V_{max}$ is the maximum potential decomposition rate, and $K_m$ (the half-saturation constant) is
the amount of substrate required for the decomposition rate to reach half of $V_{max}$. Carbon-use
efficiency (CUE) represents the proportion of total C assimilated by microbes that is allocated
for microbial growth (Tucker et al., 2013). We excluded a microbial death rate term (Todd-
Brown *et al.*, 2012) because we had insufficient data on death rates, and $C_{MIC}$ is only ~1% of
$C_{SOL}$ at our study site (Carrillo and Pendall, in review).

8        In contrast to the original DAMM formulation, we allowed $S_M(z,t)$ and $V_{max}(z,t)$ to vary

by depth and time, whereas existing applications of the DAMM model are generally applied to
"bulk" soil (i.e., do not vary with $z$). We also modeled $V_{max}$ according to the modified energy of
activation function described in Lloyd and Taylor (1994), which essentially parallels Eqns 4b-4c:

$$V_{max}(z,t) = V_{Base} \cdot f\left(\theta,\theta_M^{ant}\right) \cdot \exp\left( E_o(z,t) \cdot \left( \frac{1}{T_{ref}-T_o} - \frac{1}{T_S(z,t)-T_o} \right) \right) \qquad (6)$$

$V_{Base}$ is the 'base' $V_{max}$ at a reference soil temperature of $T_{ref}$ and at mean values of current $\theta$ and
antecedent $\theta$ and $T_S$ (i.e., mean values of $\theta_M^{ant}$ and $T_s^{ant}$). $E_o(z,t)$ and $f\left(\theta,\theta_M^{ant}\right)$ follow the same
functional forms and interpretation as described for the root respiration submodel (Eqns 3 and
4a-c), except that $\theta_M^{ant}$ and $T_M^{ant}$ are used instead of $\theta_R^{ant}$ and $T_R^{ant}$, respectively, and different
values are specified for the parameters $\alpha_1$, $\alpha_2$, $\alpha_3$, $\alpha_4$, $T_o$, and $E_o^*$ to reflect microbial respiration
(see Table 1).

19        Finally, $C_{SOL}$ is modeled as a function of soil organic C content at depth $z$, $C_{SOM}(z)$, based

on the fraction, $p$, of $C_{SOM}(z)$ that is soluble and the diffusivity of the substrate in liquid, $D_{liq}$
(Davidson et al., 2012).  The equation for $C_{SOL}$ is given by:

$$C_{SOL}(z,t) = C_{SOM}(z) \cdot p \cdot \theta(z,t)^3 \cdot D_{liq} \qquad (7)$$





The values of $p$ and $D_{liq}$ were taken from laboratory analysis (see § 2.4.5) and Davidson et al.
(2012), respectively. We assumed that $C_{SOM}(z)$ and $C_{MIC}(z)$ (see Eqn 5) are constant over time
given the relatively short simulation periods we explored here (a single growing season); but the
model could be easily modified to allow for time-vary $C_{SOM}$ and $C_{MIC}$. Here, $C_{SOM}(z)$ and $C_{MIC}(z)$
are simple, empirical functions that were informed by data (see Appendix S1 for details).
Moreover, while assumption of time invariant $C_{SOM}(z)$ and $C_{MIC}(z)$ is an implicit SS assumption
about biological factors affecting soil $CO_2$ dynamics, this assumption allows us to isolate the
importance of NSS conditions that are primarily due to physical $CO_2$ transport characteristics.
*2.1.3 Soil respiration*
The efflux of $CO_2$ from the soil surface (soil respiration, $R_{soil}$) is computed as:

$$R_{soil}(t) = \frac{D_{gs}(z = 0.01, t)}{\Delta z}\left(c(z = 0.01, t) - c_{atm}(t)\right) \tag{8}$$

$D_{gs}(z=0.01, t)$ is the diffusivity of $CO_2$ in the soil and $c(z=0.01, t)$ is the total $CO_2$ concentration
(microbial- and root-derived), respectively, at $z = 0.01$ m depth and time $t$; $c_{atm}(t)$ is the $CO_2$
concentration in the atmosphere above the soil surface; and $\Delta z$ is the depth increment that the
model solves for soil $CO_2$ concentration (here, $\Delta z = 0.01$ m).
**2.2 Numerical implementation of the DETECT model**
The numerical solution to the NSS version of the DETECT model v1.0, as described in Eqns 1-8,
requires an initial condition (IC) and two boundary conditions (BCs), which we specified as:

IC:
$$c(z, t = 0) = c_0(z) \tag{9a}$$

Upper BC:
$$c(z = 0, t) = c_{atm}(t) \tag{9b}$$

Lower BC:
$$\frac{\partial c(z = 1, t)}{\partial z} = 0 \tag{9c}$$



The function $c_0(z)$ is determined and parameterized using observed soil $CO_2$ concentration data
at three depths (Appendix S2, supplemental material); we set $c_{atm}(t)$ equivalent to 356 ppm for all
$t$, which was the average near-surface, ambient atmospheric $CO_2$ concentration measured at the
PHACE site in the 2008 growing season. Following methods of Haberman (2015), we adopted a
zero-flux lower BC (Eqn 9c) due to the lack of data at or near a depth of 1 m.

6            We numerically solved the non-linear PDE (Eqn. 1) by employing a forward Euler

discretization with a centered difference method for the depth derivative at a depth increment of
$\Delta z = 0.01$ m.  To ensure numerical stability, we calculate model outputs at a numerical time-step
of $\Delta t = \frac{dt}{Ndt}$, where $dt$ is the time step at which the predicted outputs are stored (6 hours), and
$Ndt$ is the number of numerical time-steps.  $Ndt$ is computed based on the fastest (largest)
diffusion coefficient at each time step such that $Ndt \geq \frac{dt \times \max(D_{gs})}{0.5 \times (\Delta z)^2}$, where $\max(D_{gs})$ is the
maximum $D_{gs}$ across all depth increments at time $t$ (Haberman, 1998).  We solved Eqn. 1
separately for both root- and microbial-derived $CO_2$ concentrations, such that for $K = R$ or $M$:
$$\frac{c_K(z, t + \Delta t) - c_K(z, t)}{\Delta t} = D_{gs}(z, t)\left(\frac{c_K(z + \Delta z, t) - 2c_K(z, t) + c_K(z - \Delta z, t)}{(\Delta t)^2}\right)$$
$$+ \left(\frac{D_{gs}(z + \Delta z, t) - D_{gs}(z - \Delta z, t)}{2\Delta z}\right)\left(\frac{c_K(z - \Delta z, t) - c_K(z + \Delta z, t)}{2\Delta z}\right)$$
$$+ S_K(z, t) \tag{10}$$
We rearranged Eqn. 10 to solve for $c_K(z, t + \Delta t)$, which was iterated forward for all time-steps and
depth increments; total $CO_2$ concentration at each time step and depth is calculated as $c(z, t + \Delta t) =$
$c_R(z, t + \Delta t) + c_M(z, t + \Delta t)$.  We programmed the DETECT model v.10 and the numerical solution
method in Matlab (Mathworks, 2016).



**2.3 Steady-state (SS) solution to the DETECT model**
A primary goal of this work was to test if soil $CO_2$ and associated $R_{soil}$ predicted from the non-
steady-state (NSS) model (DETECT) could be distinguished from that of the steady-state (SS)
solution. The SS version of Eqn 1, which we refer to as the SS-DETECT model, can be solved
analytically as an ordinary differential equation (ODE) by setting the $\partial c / \partial z$ term to zero
(Amundson et al., 1998).  As with the NSS model, we found the SS solution to Eqn. 1 separately
for root- and microbial-derived $CO_2$ concentrations, $c_R^*(z,t)$ and $c_M^*(z,t)$, respectively. Using the
upper and lower boundary conditions described for the NSS model (Eqns 9b and 9c), the
analytical SS solutions at time $t$ and depth $z$ are derived by Amundson et al. (1998) and given by,
for $K = R$ and $M$ :

$$c_K^*(z,t) = \frac{S_K^*(t)}{D_{gs}(z,t)}\left(z - \frac{z^2}{2}\right) + c_{atm}(t) \tag{11a}$$

$$S_K^*(t) = \frac{1}{100}\sum_{z=0.01}^{1m} S_K(z,t) \tag{11b}$$

$S_K^*(t)$ is the depth-averaged source term for microbial or root production (averaging over 100
0.01-m increments).  The soil $CO_2$ diffusivity term, $D_{gs}(z,t)$, and upper boundary condition,
$c_{atm}(t)$, are the same as previously defined (Eqns 2 and 9b, respectively; Amundson *et al.* (1998)).
**2.4 Application of the DETECT and SS-DETECT models to the PHACE site**
To address our research questions related to the relative importance of NSS versus SS conditions
for understanding and modeling soil $CO_2$ transport and fluxes (e.g., $R_{soil}$), we applied the
DETECT and SS-DETECT models to data obtained from a semi-arid grassland in Wyoming. We
expected that the precipitation pulse regimes characteristic of this and other semi-arid
ecosystems would likely lead to NSS soil $CO_2$ conditions given the impacts of such pulse





regimes on soil water dynamics (Bachman et al., 2010; Kemp et al., 1997; Reynolds et al., 2004;
Sala et al., 1992; Sala et al., 1981).  For example, pulse-driven, semi-arid ecosystems can
experience long periods of dry soil conditions, under which SS soil $CO_2$ conditions likely
operate, interrupted by moisture pulses that we would expect to cause transient NSS conditions
due to rapid changes in soil air-filled porosity, temperature, and associated soil $CO_2$ diffusivity.
These NSS conditions would also likely be associated with high flux rates (e.g., high $CO_2$
production rates and high $R_{soil}$,), potentially making such transient conditions proportionally
more important over the long-term (Jarvis et al., 2007).

9       Thus, in this subsection, we provide an overview of the study site, including the PHACE

experiment, and relevant data sources from PHACE that we used to drive the DETECT and SS-
DETECT models. We also summarize how we calibrated the models in the context of the
PHACE site, and we highlight data that we used to informally validate the general behavior of
the models. We conclude by describing the simulation experiments that we conducted to test the
effects of soil texture and precipitation variability on the importance of NSS versus SS soil $CO_2$
conditions.
*2.4.1 Field site and PHACE experiment*
The Prairie Heating and $CO_2$ Enrichment (PHACE) field experiment is located in south-central
Wyoming (latitude 41º 50'N, longitude 104º 42'W, elevation = 1930 m). The site is a mixed-
grass prairie with a semi-arid climate characterized by long winters (mean January temperature =
-2.5 °C) and warm summers (mean July temperature = 17.5 °C), with mean annual precipitation
of 384 mm (Morgan et al., 2011).  The vegetation is predominantly composed of two $C_3$ grasses,
western wheatgrass (*Pascopyrum smithii* (*Rydb.*) *A. Löve*) and needle-and-thread grass
(*Hesperostipa comata Trin and Rupr*), and a $C_4$ perennial grass, blue grama (*Bouteloua gracilis*



(*H.B.K.*) *Lag*). The soil is a fine-loamy, mixed, mesic Aridic Argiustoll, and biological crusts are
not present (Bachman et al., 2010).
*2.4.2 Environmental driving data*
We simulated the transport and production of soil $CO_2$ for each 0.01 m depth increment, from the
surface (0 m) to 1 m, across all 732 time steps (i.e., 4 time steps per day [every 6 hours] for 183
days from April-September).  To do this, we required soil environmental data consisting of water
content ($\theta$) and temperature ($T_S$) and meteorological data including precipitation, air temperature,
and air pressure.  The PHACE study provided these data, or data that were used to create the
driving data at the necessary spatial and temporal resolution.

10        The PHACE experiment involved an incomplete factorial of $CO_2$, warming, and

irrigation (6 treatment levels total), with five replicate plots per treatment level, resulting in a
total of 30 instrumented plots. One of the five plots from the control treatment—ambient $CO_2$,
temperature (no heating), and precipitation (no supplemental irrigation)—was chosen at random
and had a system installed to measure soil $CO_2$ concentrations continuously for three different
soil depths (3, 10, and 20 cm). This plot, therefore, provided the data for driving the DETECT
and SS-DETECT models. Data that we used were collected during the growing season (March-
September) of 2008; $\theta$ was measured hourly at three depths (5-15, 15-25, and 35-45 cm;
EnvironSMART probe, Sentek Sensor Technologies, Stepney, Australia), and we used daily
averages to drive the models. $T_S$ was measured hourly at two depths (3 and 10 cm) using type-T
thermocouples. Likewise, hourly precipitation (mm), air temperature (ºC), relative humidity (%),
and surface barometric air pressure (kPa) were recorded by an automated weather station at the
site.





*2.4.3 High resolution environmental data*
To accommodate the 0.01 m depth increments specified for the DETECT model, we used the
coarse resolution field data (above) to create finer resolution driving data. For example, temporal
gap-filling of the $\theta$, $T_S$, and micrometeorological datasets was required due to gaps that occurred
during a small number of days (<1%, 6%, and 2.5%, respectively) as a result of instrument
failure.  We used data from other nearby plots to estimate the values of the missing data, but we
also used cubic spline interpolation where gaps remained.  Details of these gap-filing methods
can be found in Ryan *et al.* (2015).
We used HYDRUS-1D v4.16.0090 to simulate $\theta$ and $T_S$ in 0.01 m increments from a
depth of 0.01 m to 1 m (Chou et al., 2008; Piao et al., 2009; Šimůnek et al., 2008) based on
precipitation data at the site. HYDRUS simulates the movement of water by solving the
Richards' equation for water movement (Chou et al., 2008; Richards, 1931; Sitch et al., 2008)
and heat transport via Fickian based advection-dispersion equations.  Soil hydraulic and heat
transport parameters were estimated in HYDRUS using the inverse mode to solve for parameter
values based on the PHACE $\theta$ (5-10, 15-25, and 35-45 cm) and $T_S$ (3 and 10 cm) data.
HYDRUS was then run in forward mode based on the tuned soil hydraulic parameters to
estimate $\theta$ and $T_S$ at all 100 0.01-m depth increments at 6-hourly time intervals.  For consistency,
HYDRUS-derived $\theta$ and $T_S$ were used as the environmental input data to the DETECT models,
even at the depths for which PHACE data were available.
*2.4.4 Antecedent soil water and soil temperature conditions*
We explicitly evaluated the impact of antecedent (past) $\theta$ and $T_S$ conditions on $CO_2$ production
by roots and microbes, motivated by prior work that estimated the relative importance of
antecedent conditions and their time-scales of influence on soil and ecosystem $CO_2$ efflux





(Barron-Gafford et al., 2014; Cable et al., 2013; Ogle et al., 2015; Ryan et al., 2015). Antecedent
soil water content and antecedent soil temperature—$\theta_K^{ant}(z,t)$ and $T_s^{ant}(z,t)$, respectively, for $K =$
$R$ (roots) and $M$ (microbes)—were computed as weighted averages of the HYDRUS-produced
$\theta(z,t)$ and $T_S(z,t)$ data, respectively. These calculations were done external to the DETECT
model, and the antecedent variables were supplied as driving variables to DETECT. For
example, for each 0.01 m increment ($z$) and time period ($t$), antecedent soil water associated with
microbial $CO_2$ production was calculated as:

$$\theta_M^{ant}(z,t) = \sum_{j=1}^{J} w(j) \cdot \theta(z, t-j) \tag{12}$$

The $w$'s are the antecedent importance weights, which sum to 1 from $j = 1$ (previous time period)
to $j = J$ ($J$ time periods previous). The weights were informed by results from an analysis of
ecosystem respiration at the PHACE site (Ryan et al., 2015). For microbes, $J = 4$ days and $w =$
(0.75, 0.25, 0, 0), indicating the strong importance of $\theta$ conditions occurring yesterday ($j = 1$)
(Oikawa et al., 2014). Similar equations were used to compute $\theta_R^{ant}(z,t)$ and $T_s^{ant}(z,t)$, each with
their own set of weights ($w$'s) and time-scales ($J$'s). For example, the time step and $J$ for $\theta$ differ
among microbes (2 days) and roots (3 weeks); for roots, $\theta_R^{ant}(z,t)$ was computed as a weighted
average of past, average weekly values of $\theta$, with $j$ denoting weeks into the past, for $J = 4$ weeks,
and $w =$ (0.2, 0.6, 0.2, 0), indicating a strong lag response to $\theta$ conditions occurring two weeks
ago (Cable et al., 2013; Ryan et al., 2015). For antecedent soil temperature, we assumed that
each of the past four days were equally important by setting the $w$ vectors to (0.25, 0.25, 0.25,
0.25), for both microbes and roots (Ryan et al., 2015). The specification of $J$ and the $w$'s are
independent of the DETECT model formulation and can be varied by the user. Specifically, the
antecedent drivers become input variables that are specified outside of the DETECT machinery.



*2.4.5 Overview of parameterization approach using PHACE data*
In general, our aim was to specify realistic values for the parameters in the DETECT model. We
did not formally "fit" the DETECT model to data, but rather, we simply determined reasonable
values based on simple analyses of relevant PHACE data sets, results published for the PHACE
site, or results from other relevant studies. For clarity, we categorize the parameters into four
groups (Table 1). The first group consists of parameters used solely for the microbial source or
production term ($S_M$, Eqns 5-6), and the second group of parameters belong solely to the root
respiration source term ($S_R$, Eqns 3-4). Parameters in the third group are shared between the $S_M$
and $S_R$ submodels; the fourth group contains parameters associated with calculating $CO_2$
diffusivity ($D_{gs}$, Eqn 2). The full list of parameters is given in Table 1, and below we describe the
logic behind specifying specific values in Table 1.

12        The depth-distributions of root biomass C ($C_R$, Eqn 3), soil microbial biomass C ($C_{MIC}$,

Eqn 5), and soil organic C ($C_{SOM}$, Eqn 7) are expressed in terms of a total C content in a 1 m
deep soil column (e.g., $R^*$, $M^*$, and $S^*$, respectively; mg C cm$^{-2}$), multiplied by the proportion of
that C that occurs at depth $z$ (e.g., $f_R$, $f_M$, and $f_S$, respectively). For example, $C_R(z,t) =$
$R^* \cdot f_R(z) \cdot G(t)$, where $G(t)$ is a function that scales $R^*$ by an index of time-varying vegetation
activity based on vegetation greenness estimates (Pendall et al., 2001; Appendix S1). The depth-
varying distributions of C contents were approximated by fitting a simple exponential function to
data on root, microbial, and soil organic C content, thus providing estimates for $R^*$, $M^*$, $S^*$, $f_R(z)$,
$f_M(z)$, and $f_S(z)$ (see Appendix S1 for details; Table 1). Regarding the data, soil organic C was
determined by combustion of acidified, root-free soil collected from 0-5, 5-15, 15-30, 30-45, 45-
75, and 75-100 cm depths, using a Costech Elemental Analyzer. Microbial biomass C was
determined by the chloroform fumigation and extraction in 0.05 M $K_2SO_4$ (Carrillo et al.,



2014b).  Extracts were analysed for total C on a total organic carbon analyzer (Shimadzu TOC-
VCPN; Shimadzu Scientific Instruments, Wood Dale, IL, USA) after treating with 1 M H3PO4
(1 µl per 10 ml of extract) to remove any carbonates. Root biomass C was estimated from ash-
free root biomass and elemental analysis (Carrillo et al., 2014a; Mueller et al., 2016).  The
solubility parameter, $p$, was estimated as the ratio of $C_{SOL}$ to $C_{SOM}$ (Eqn 7), and was based on
unfumigated extracts obtained for microbial biomass estimations as above ($C_{SOL}$) and on total C
concentration in soil ($C_{SOM}$).
The values used for the base microbial respiration rates and the half-saturation constant
($V_{Base}$ [Eqn 6] and $K_m$ [Eqn 5]; Table 1) were estimated by fitting the microbial respiration
submodel, but without the $C_{MIC}$ or $CUE$ terms (Eqn 5), to microbial respiration data from the
PHACE control plots (Fig. S6).  In the absence of root respiration data, we assumed that base
root respiration ($R_{Rbase}$ [Eqn 3]; Table 1) was proportional to the microbial base rate term
(Hanson et al., 2000).  The parameters denoting the effects of current soil moisture (e.g., $\alpha_1$; Eqn
4a), antecedent moisture ($\alpha_2$), and the interaction between current and antecedent moisture ($\alpha_3$)
on root and microbial respiration were derived from Ryan *et al.* (2015), also based on an analysis
of ecosystem respiration ($R_{eco}$) data from PHACE.  However, we adjusted the values (Table 1) to
reflect the expectation that the effects of current soil moisture should be stronger for microbial
compared to root respiration because microbes tend to respond more rapidly to precipitation
pulses (Risk et al., 2008), whereas root respiration is likely to show a delayed response that
depends more strongly on past moisture conditions (Cable et al., 2013; Cable et al., 2008).  Of
the remaining two parameters describing $S_M$ (Eqns 5-6; Table 1), the value of $CUE$ was based on
results from a soil incubation study conducted at a nearby site (Tucker et al., 2013), whilst our
value for $D_{liq}$ was taken from Davidson *et al.* (2012).  Three parameters ($E_o^*$, $T_o$, and $\alpha_4$; Eqns





4a-b) were shared between the $S_R$ and $S_M$ submodels, and their values were also obtained from
Ryan *et al.* (2015). Finally, the parameters used for $CO_2$ diffusivity (*b*, *BD*, and $\phi_{g100}$; Eqn 2)
were based on published, site-specific data (Morgan et al., 2011).
*2.4.6 Informal model validation with soil respiration measurements*
We evaluated the accuracy of the DETECT model by comparing (1) predicted $R_{soil}$ (Eqn 8)
against plot-level measurements of ecosystem respiration ($R_{eco}$) (see below) and (2) predicted
soil $CO_2$ concentrations, $c(z,t)$, versus observed concentrations; all observed data were from the
PHACE study. Since we did not rigorously parameterize the DETECT model with PHACE data,
we were simply looking for reasonable, qualitative agreement between the modelled variables
and the observations (e.g., similar order of magnitude, comparable temporal trends). Observed
$R_{eco}$ was measured on control plots every 2-4 weeks during the target growing season, using a
canopy gas exchange chamber, and instantaneous fluxes were scaled to daily rates using a linear,
empirical function (Bachman et al., 2010; Jasoni et al., 2005).  We assumed that $R_{soil}$ was similar
to $R_{eco}$ given that aboveground biomass was <20% of total plant biomass (Mueller et al., 2016).
Glyphosate herbicide was applied to small subplots in May, 2008, limiting ecosystem $CO_2$ efflux
to microbial sources (Pendall et al., 2013), and non-steady state soil chambers were used to
estimate surface soil fluxes every two weeks around midday (Ogle et al., 2016; Oleson et al.,
2013); these data provided observations of microbial respiration. Soil $CO_2$ concentrations were
also measured with non-dispersive infrared sensors (Vaisala GM222, Finland) installed at 3, 10,
and 20 cm below the soil surface, averaged on an hourly basis (Brennan, 2013; Risk et al., 2008;
Vargas et al., 2011). Observations of soil $[CO_2]$ for control plots were compared against
predictions of $c(z,t)$ at $z = 0.03$, 0.1, and 0.2 m and at the corresponding times.



**2.5 Simulation Experiments**
We evaluated the impact of three potentially important factors that could affect the frequency of
NSS (Eqns 1 and 9a-c) relative to SS (Eqn 10) conditions: (1) soil texture, (2) precipitation
patterns, and (3) importance of antecedent conditions. In the control (*Ctrl*) scenario, we
calculated the source terms and diffusion terms ($S_K$ and $D_{gs}$ in Eqns 1 and 2) based on soil
environmental ($\theta$ and $T_S$), soil texture (sandy clay loam: 60% sand, 20% silt, 20% clay), and
meteorological data (e.g., precipitation) measured at the PHACE site in 2008. We varied soil
texture, relative to that of the site, by varying the relative amounts of sand, silt, and clay, giving
three levels (Table 3): 80% sand, 10% silt, and 10% clay (sandy loam, scenario denoted as *ST-*
*Sa*); 20% sand, 60% silt, and 20% clay (silt loam, *ST-Si*); 20% sand, 20% silt, and 60% clay
(clay, *ST-Cl*). The control (*Ctrl*) scenario was also paired with the observed daily precipitation
data for 2008. We explored three additional precipitation scenarios, under the control soil
texture, by shifting the daily precipitation to occur one month earlier, or one month later, or by
using precipitation data from 2009 (scenarios *P-E*, *P-L* and *P-FM*, respectively; Table 3). For *P-*
*FM*, we chose 2009 because it had approximately the same total precipitation between April and
September as 2008, but it fell as more frequent events of smaller magnitudes. For each texture
and precipitation scenario, HYDRUS was used to compute the corresponding $T_S$ and $\theta$ at the
required depth and time intervals. All above scenarios assumed that antecedent conditions are not
important, which was achieved by setting all antecedent effects parameters ($\alpha_2$, $\alpha_3$, and $\alpha_4$; Table
1) equal to zero. We contrasted these scenarios against ones that included antecedent conditions
(thus, computed $\theta_K^{ant}$ and $T_s^{ant}$ in Eqs 3 and 6) in the calculation of soil $CO_2$ production by roots
($K=R$) and microbes ($K=M$); all such scenario names were appended with "*ant*" (Table 3, Fig.



1a).  For each scenario summarized in Table 3, we evaluated the potential for NSS conditions by
comparing the predicted $R_{soil}$ produced by the DETECT model versus the SS-DETECT model.

## 3. Results

Our analysis indicated that soil texture was the strongest predictor of whether soil $CO_2$ and
associated $R_{soil}$ were in steady state or not, particularly during periods of high precipitation. Thus,
precipitation patterns played a secondary role.  The inclusion of antecedent soil water and soil
temperature effects in the model resulted in a significant increase in predicted annual $R_{soil}$ but
only for the control and fine textured soil scenarios, and resulted in predicted soil $CO_2$ being
closer to soil $CO_2$ measurements from the PHACE site.  Below, we summarize key results from
this study.

### 3.1 Control Scenarios

Soil $CO_2$ was in steady state (SS) during most of the growing season under the control soil
texture (sandy clay loam) and precipitation conditions that assumed no antecedent affects (*Ctrl*
scenario).  For example, soil respiration ($R_{soil}$) predicted by the DETECT model was
approximately equal to $R_{soil}$ predicted by the SS-DETECT model during times of no or little
precipitation (Fig. 2a, days < 218 or > 230). Conversely, $R_{soil}$ predicted by the SS-DETECT
model was temporarily greater and more variable than that predicted by the DETECT model
immediately following a large precipitation event (Fig. 2a, days 218-229).  However, the total
cumulative $R_{soil}$ between days 92 to 274 – hereafter 'total growing season $R_{soil}$' – under SS (497
g C m$^{-2}$) versus NSS (498 g C m$^{-2}$) assumptions was approximately equal (a difference of
~0.2%).





The effects of antecedent conditions (*Ctrl-ant* scenario; Fig. 2b) were generally consistent
with the control scenario without antecedent conditions. However, the magnitude of $R_{soil}$
predicted by both the DETECT and SS-DETECT models was up to 9 gC m$^{-2}$ day$^{-1}$ greater during
days following the major rain event (i.e., during days 230-243) when antecedent conditions were
considered. Moreover, the incorporation of antecedent effects led to a longer delay between the
timing of the major rain event and the maximum $R_{soil}$, which occurred ~5 days later than when
only current conditions were considered (Fig. 2a vs. 2b).  As a result, total growing season $R_{soil}$
was ~15% higher under the *Ctrl-ant* scenario (e.g., 571 gC m$^{-2}$ under NSS assumptions, Fig. 2b)
compared to the *Ctrl* scenario (e.g., 498 gC m$^{-2}$ under NSS, Fig. 2a). This increase in predicted
$R_{soil}$ under the *Ctrl-ant* scenario for days 230-243 was primarily driven by greater root respiration
(Fig. 2a vs 2b).
**3.2 Effects of soil texture**
Varying soil texture resulted in the greatest difference in daily $R_{soil}$ between the DETECT and
SS-DETECT models; however, integrated over the growing season, these differences were very
small (Fig. 3a,b,c).  In particular, total growing season $R_{soil}$ predicted by SS-DETECT was ~1.5%
less than predicted by DETECT for soils consisting primarily of sand and silt (*ST-Sa* and *ST-Si*
scenarios; Fig. 3a,b), but was ~3.3% less for a clay dominated soil (*ST-Cl* scenario; Fig. 3c red
versus grey bars). These differences in $R_{soil}$ under NSS versus SS assumptions were
approximately the same for the scenarios involving antecedent effects (Figs. 3d,e,f).  Despite the
minor differences at the growing season scale, notable differences emerged at the daily scale.
For example, with the largest precipitation event of the year and the 10 days that followed (days
218-248), daily $R_{soil}$ predicted by the DETECT model was on average ~2.5% less than daily $R_{soil}$
from the SS-DETECT model for the *ST-Sa* and *ST-Si* scenarios (Fig. S1). $R_{soil}$ from DETECT





was 4% greater than DETECT-SS $R_{soil}$ for the *ST-Cl* scenario, but when antecedent variables
were included in the models, this difference increased to 10% (Figs. 3 and S1).

3        Soil texture also affected the magnitude of predicted $R_{soil}$ compared to the control

scenarios, both with and without antecedent effects (*Ctrl-ant* and *Ctrl*, respectively). In
particular, we found that total growing season $R_{soil}$, whether from the DETECT or the SS-
DETECT model, was ~30% and ~60% higher for the *ST-Si* and *ST-Cl* scenarios relative to the
*Ctrl* scenario (Figs. 3b, 3c, 4a). The change in $R_{soil}$ was negligible, however, when the sand
content was increased from 60% (*Ctrl*) to 80% (*ST-Sa*) for both models (Fig. 3a, Fig. 4a). The
antecedent versions of the fine-textured scenarios (*ST-Si-ant* and *ST-Cl-ant*) resulted in ~45%
and ~95% increases in total growing season $R_{soil}$, respectively, compared to the *Ctrl-ant* scenario
(Figs. 3e, 3f, 4b). Greater root respiration (Figs. 3e,f), following the end of the second
precipitation period between days 230 and 245, primarily drove the larger percentage increases
for the *SL-Si-ant* and *SL-Cl-ant* scenarios compared to the non-antecedent versions (Fig. 4b vs
Fig. 4a; Fig. 4e).
**3.3 Effects of precipitation regimes**
Based on the four different precipitation scenarios that we explored in the context of the control
soil texture (sandy clay loam), varying the timing, frequency, or magnitude of precipitation led to
little difference between $R_{soil}$ predicted by the DETECT and SS-DETECT models (Fig. S2).
However, the precipitation regime did affect the magnitude of $R_{soil}$ predicted under NSS or SS
conditions. For example, total growing season $R_{soil}$ predicted under the alternative precipitation
scenarios was lower relative to the *Ctrl* scenario. This decrease was relatively small (5-10%) for
the non-antecedent versions of the models (Fig. 4c), but was comparatively larger (15-22%) for
the antecedent versions (Fig. 4d). This reduction appears to be driven by the amount of time





over which daily $R_{soil}$ responded to the second precipitation period, which occurred around day
220, 190, and 250 in the *Ctrl*, *P-E*, and *P-L* scenarios, respectively. Following this precipitation
event, daily $R_{soil}$ achieved values around 10 g C m$^{-2}$ day$^{-1}$ for about 20 days in the *Ctrl* scenario
(Fig. 2a, days 220-240), but for only about five days in the *P-E* and *P-L* scenarios (Fig S2a,b,
after days 190 and 250, respectively). Increasing the frequency of precipitation while retaining
approximately the same annual amount (i.e., scenario *P-FM*) resulted in daily $R_{soil}$ being
consistently less than that of the *Ctrl* scenario, which led to a reduction in total growing season
$R_{soil}$ in the *P-FM* scenario (Fig. S2c and S2e).
**3.4 Effects of antecedent responses**
When antecedent soil water content and soil temperature were included in the DETECT model
we found that predicted $R_{soil}$ was 15% greater for the control scenario and 29-37% greater for the
fine textured soil scenarios, compared to the corresponding scenarios that did not include
antecedent conditions.  When the sand content was 80% or for any of the different precipitation
regimes, there was a negligible difference between $R_{soil}$ predicted by the antecedent versus non-
antecedent parametrizations of DETECT.

16       Daily $R_{soil}$ predicted by the DETECT model based on the *Ctrl* and *Ctrl-ant* scenarios

agreed well with observed ecosystem respiration ($R_{eco}$), but $R_{eco}$ was slightly higher than
predicted $R_{soil}$ (Fig. 2a,b), which was expected since $R_{eco} = R_{soil}$ + aboveground autotrophic
respiration.  For the most part, this data-model agreement was similar whether the antecedent
model terms were included (Fig. 2b) or not (Fig. 2a). Unfortunately, $R_{eco}$ data were not available
during the time period (days 230-250) associated with the greatest disagreement between the *Ctrl*
and *Ctrl-ant* scenarios.  During this period, frequent hourly measurements of soil [$CO_2$] were in
better agreement with predicted soil $CO_2$ from the *Ctrl-ant* scenario compared to the *Ctrl*



scenario (Fig. 5a,b, S3a,b).  After day ~250, based on the DETECT model, both scenarios (*Ctrl*
and *Ctrl-ant*) under-predicted the observed soil [$CO_2$] by ~ 50% (Fig. 5).
**4. Discussion**
The DETECT and SS-DETECT models provide a framework for evaluating the circumstances
under which steady-state (SS) assumptions of soil $CO_2$ production and surface soil respiration
($R_{soil}$) are valid, and to identify the major physical (i.e., soil texture, soil moisture) and/or
biological (i.e., root and microbial respiration responses) factors that lead to non-steady-state
(NSS) conditions.
**4.1 Steady-state versus non-steady-state conditions**
At the seasonal scale, there was reasonable agreement between total growing season $R_{soil}$
predicted under the assumption of SS versus NSS conditions, but the strength of this agreement
depended on soil texture (see §4.2).  At the daily scale, $R_{soil}$ predicted by the DETECT model
deviated from values expected under the assumption of SS conditions for 11 days or 4% of the
days during the April-September growing season (Fig 2, days 218-228).  These discrepancies,
attributed to NSS conditions, were generally limited to periods following large rain events.  For
applications that assume SS conditions, such as isotopic partitioning studies (Hui and Luo, 2004;
Ogle and Pendall, 2015), the SS assumption seemed reasonable during periods of minimal or no
precipitation, representative of times during which soil water content changes very little or
gradually.  For sites or time periods characterized by pulsed precipitation patterns, our results
suggested that NSS conditions would be more likely over longer periods of time.
**4.2 Effect of varying soil texture**
Our results indicated that soil texture exerts the strongest control over the prevalence of NSS soil



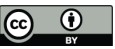

$CO_2$ conditions. For a predominantly (e.g., 60%) sandy or silty soil, soil $CO_2$ transport and efflux
generally aligned with the SS assumption (Fig. 2, Fig. 3a-b). This was consistent with previous
work that used SS models to predict $R_{soil}$ for similar soil types (Baldocchi et al., 2006; Vargas et
al., 2010).

5        For very fine-texture soil dominated by clay, however, SS assumptions were far less

appropriate.  The larger difference in $R_{soil}$ predicted under SS versus NSS conditions for fine-
texture (i.e., 60% clay) soil was apparent at both the growing season scale and the daily scale
following a large precipitation event (Fig. 3c,e, from day 218).  In general, the DETECT model
predicted that $R_{soil}$ should be higher in clay compared to sandy soil after precipitation events, a
result supported by field experiments (Cable et al., 2008), but this texture effect is muted under
assumptions of SS.  Moreover, recovery of $R_{soil}$ to SS rates after a large rain event took ~30 days
in the clay soil (Fig. 3c, days 218 to 248) compared to ~10 days for the other coarser soil texture
scenarios (Fig. 2, Fig. 3a-b, days 218 to ~230). These effects of soil texture on the prevalence of
NSS conditions can be attributed to soil physical properties and their effects on air-filled porosity
and $CO_2$ diffusivity. Fine textured soils have smaller pores and tend to retain water for longer
(Bouma and Bryla, 2000), which has the effect of decreasing soil $CO_2$ diffusivity (Fig. 6). Thus,
under moist conditions that follow a rain event, it may take 15 minutes or so for a $CO_2$ molecule
produced at 0.5 m to diffuse to the surface in a clay soil compared to only 1-2 minutes for a
sandy soil. Moreover, fine-textured soils have slower infiltration rates (Hillel, 1998), delaying
the exposure of more deeply distributed roots and microbes to increased moisture availability.
While this effect may not directly impact the SS assumption, it would lead to greater time lags
between precipitation pulses and $R_{soil}$ peaks.



These findings have important implications for studies that rely on the SS assumption to
predict subsurface soil $CO_2$ production. The SS assumption may be sufficient for systems
defined by coarse-textured soils, but it may lead to erroneous conclusions if applied to fine-
textured soils. Our simulation experiments made the simplifying assumption that soil texture is
constant with depth, but in many ecosystems, texture may vary greatly with depth (Ogle et al.,
2004).  An important next step is to extend the simulations to explore the impacts of depth-
varying soil texture on SS versus NSS conditions.  The DETECT model can easily accommodate
such modifications; allowing soil texture to vary by depth would have a direct effect on soil
water content, which is simulated outside of DETECT using HYDRUS (Chou et al., 2008; Piao
et al., 2009; Šimůnek et al., 2008), that can accommodate such depth variation.
**4.3 Effect of varying the timing or frequency of precipitation**
Unlike soil texture, varying the timing, frequency, and magnitude of precipitation resulted in
predicted $R_{soil}$ that was almost identical under SS and NSS assumptions, both at the growing
season and daily time-scales (Fig. S2).  We had anticipated that such changes in the precipitation
regime would impact SS conditions via impacts on soil air-filled porosity and potentially by
changing the covariance between soil water and soil temperature, both of which affect soil $CO_2$
diffusivity (e.g., see Eqn 2). We did not explore, however, the effect of decreasing the frequency
while simultaneously increasing the magnitude of individual pulses. We hypothesize that this
latter scenario could produce more exaggerated or extended NSS conditions given that large rain
events would infiltrate deeper, reducing $CO_2$ diffusivity across greater soil depths, thus slowing
the transport of more deeply derived $CO_2$. Increasing the number of small events, as done in the
*P-FM* scenario, would generally confine water inputs to shallow layers, from which $CO_2$ has



shorter distances to travel to reach the surface, creating less opportunity for $R_{soil}$ to exhibit NSS
behavior.

3        While the precipitation regimes that we explored (Table 3) did not notably impact SS

versus NSS behavior, they did influence the magnitude of growing season $R_{soil}$, with the
alternative regimes resulting in a decrease in growing season $R_{soil}$ between 6% and 11%
compared to the control scenario (Fig 4c,d, Fig. S2). The decrease in $R_{soil}$ as a result of
increasing the frequency while decreasing the magnitude of precipitation (*P-FM* and *P-FM-ant*
scenarios) is consistent with previous research (Borken and Matzner, 2009; Cable et al., 2008;
Huxman et al., 2004; Ogle et al., 2015; Schwinning et al., 2004; Sponseller, 2007). For example,
Cable et al. (2008) found that soil respiration is insensitive to small precipitation pulses (<7mm
in size), while Borken and Matzner (2009) found that an increase in the drying and wetting
frequency reduced the cumulative C mineralization. Likewise, shifting precipitation to one
month later (*P-L* scenario), which resulted in an abundance of rainfall at the end of the growing
season, resulted in lower growing season $R_{soil}$. In contrast, Chou et al. (2008) found that
abundant end of growing season precipitation increased growing season $R_{soil}$ (Fig. 4c,d; Fig.
S2b,e). These differences in findings could be due to a number of reasons, such as dissimilar
daily precipitation patterns, different study systems, and different approaches to quantifying $R_{soil}$.
In Chou et al. (2008), the late season precipitation in one year allowed soil water content to
remain high for longer, resulting in high growing season $R_{soil}$ compared to a different year with
greater overall annual precipitation but without the late surge in precipitation. In our analysis,
precipitation in the study year (2008) was concentrated in two relatively short time windows, and
thus, our scenarios are not directly comparable to those observed by Chou et al. (2008). Our
analysis could be extended, however, by exploring the impacts of more varied precipitation



regimes; given the flexibility of simulation approaches, we could use the DETECT model to
explicitly evaluate $R_{soil}$ responses to any particular observed or hypothetical precipitation regime.
**4.4 Effect of antecedent conditions**
The inclusion or exclusion of antecedent soil moisture and temperature effects on $CO_2$
production rates had little to no impact on the balance between SS versus NSS behavior of $R_{soil}$.
However, incorporating antecedent effects generally increased the magnitude of $R_{soil}$ as
microbial respiration was stimulated more during the initial onset of the main precipitation
period when antecedent effects were considered (Fig. 2b vs Fig 2a, day 218, blue line). This is
expected because the instantaneous response of microbes to a rain event is expected to be greater
following a dry period compared to during a wet period (Cable et al., 2013; Cable et al., 2008;
Sponseller, 2007; Thomas et al., 2008; Xu et al., 2004). These dynamics are incorporated in the
antecedent version of the models when the parameter corresponding to the interaction between
current and antecedent soil water content is negative (e.g., $\alpha_3$, Table 1).  Secondly, root
respiration was greatly enhanced following the end of this period of precipitation (Fig. 2b vs Fig.
2a, days ~230-250, green line), despite there being little precipitation after day 230 (Fig. 2b).
This likely occurred because our DETECT model assumed that soil water over relatively longer
time periods (past 1-2 weeks, Eqn. 12) affects current root respiration rates.  This partly reflects
the mechanism that roots are able to take up more soil water that has infiltrated to deeper depths
(Cable et al., 2013).  The microbes, however, are coupled to past conditions over comparatively
short time periods (a couple days).

21         The importance and benefit of including antecedent terms for modelling soil respiration

or ecosystem respiration has been well documented (Barron-Gafford et al., 2014; Cable et al.,
2013; Ryan et al., 2015).  Thus, we encourage future studies to include influences of past



conditions when modelling subsurface and surface $CO_2$ fluxes. Fortunately, our simulation
experiments suggest that the lagged responses of microbial and root respiration to soil moisture
and temperature do not have a notable impact on the SS assumption.
**4.4 Comparison of modelled soil $CO_2$ with data**
The good agreement between modeled and observed soil $CO_2$ concentrations—particularly when
including antecedent effects—was very encouraging because the DETECT model was not
rigorously tuned or calibrated to fit data on soil $[CO_2]$ or ecosystem $CO_2$ fluxes ($R_{eco}$) (Figs. 5,
S3). However, there remained discrepancies between the predicted and observed $CO_2$ fluxes,
particularly after rain events. These discrepancies could be an artifact of the input data used to
calculate $CO_2$ production (i.e., the source term). Some parameter values were drawn from the
literature and others were estimated by fitting a non-linear regression model to data. For
example, the parameters describing the current and antecedent soil water content effects ($\alpha$'s)
were obtained by fitting a non-linear model to $R_{eco}$ data (Ryan et al., 2015). While measured $R_{eco}$
represents both root respiration and microbial respiration contributions, it also reflects
aboveground respiration, which is not currently treated in the DETECT model. Moreover, we
made further assumptions about how the $R_{eco}$ parameter estimates translate to component
processes (root and microbial responses), and we relied on literature information about how
microbes and roots respond to precipitation events (e.g., the timing, magnitude, and lags). Future
studies could rigorously fit the DETECT model to field data, such as observations of $R_{soil}$, soil
$CO_2$ concentrations, and $^{13}C$ isotope fluxes. Using a Bayesian methodology to do this would
allow one to incorporate multiple data sets to inform all parameters in DETECT.
**4.5 Non-steady state model of soil $CO_2$ transport and production**
An important contribution of this this study was the development of a non-steady state (NSS)



model of soil $CO_2$ transport and production (the DETECT model version 1.0), which is
particularly useful for systems that may frequently experience NSS conditions. Other comparable
NSS models exist (e.g., Fang and Moncrieff, 1999; Hui and Luo, 2004; Šimůnek and Suarez,
1993), but they generally treat the production (source) terms—root/rhizosphere respiration and
microbial decomposition of soil organic matter—simplistically, and accompanying model code
is not available. Our DETECT v1.0 model includes more detailed submodels for the production
terms, inspired by recent studies (E.g. Carrillo et al., 2014a; Davidson et al., 2012; Lloyd and
Taylor, 1994; Pendall et al., 2003; Todd-Brown et al., 2012); in contrast to these studies, which
essentially described models for "bulk" soil, we applied the $CO_2$ production models to every
depth increment. Additionally, we have provided model code, implemented in Matlab (see *Code*
*Availability* section), with the goal of making the DETECT model, and ability to accommodate
NSS conditions, more accessible to potential users.
**5. Conclusions**
Determining the conditions under which steady-state (SS) assumptions are appropriate for
modeling soil $CO_2$ production, transport, and efflux is crucial for accurately modeling the
contribution of soils to the carbon cycle. We found that soil texture exerted the greatest control
over whether SS assumptions are appropriate. When the soil at a site is coarse (60% or more
sand), SS assumptions appeared to be appropriate, and one could apply a simpler, more
computationally efficient SS model, such as SS-DETECT (see also Amundson et al., 1998). As
the soil texture becomes increasingly finer, SS assumptions start to break down, especially
following large precipitation events that can greatly impact soil water content and associated soil
air-filled porosity, thus affecting $CO_2$ diffusivity.  Under such conditions, the more complex and
computationally demanding NSS model (DETECT) is preferred. We found that precipitation



regime characteristics and/or the inclusion of antecedent soil moisture and temperature
conditions had little singular effect on whether SS or NSS assumptions were appropriate.
However, while these factors do not directly impact SS versus NSS behavior, they were found to
be important for accurately modeling the soil carbon cycle because they notably impacted the
magnitude of the soil $CO_2$ efflux.
**Code availability**
All of the Matlab script files for running the DETECT model can be accessed via
http://doi.org/10.5281/zenodo.927501.  These Matlab script files are set up so that the model
runs at the PHACE field site.  The above weblink also provides a user manual which gives
instructions for running DETECT at either the PHACE site or at a user specified field site.  We
also provide Matlab script files for creating a time series of predicted versus observed soil
respiration (figure 1) and a time series of predicted versus observed soil $CO_2$ (figure 5).  These
can be found via http://doi.org/10.5281/zenodo.927313.  Following publication, these Matlab
files and the data files (see next section) will be available to download from the Ogle lab website
via http://jan.ucc.nau.edu/ogle-lab/.
**Data availability**
Measurement data made at the PHACE field site, which are required as inputs for the DETECT
model, are available via http://doi.org/10.5281/zenodo.926064.
**Acknowledgements**
We thank Dan LeCain, David Smith, and Erik Hardy for implementing and managing the
PHACE experiment, and Jack Morgan for project leadership. This material is based upon work
supported by the US Department of Agriculture Agricultural Research Service Climate Change,





1 Soils & Emissions Program, USDA-CSREES Soil Processes Program (#2008-35107-18655), US

2 Department of Energy Office of Science (BER), through the Terrestrial Ecosystem Science

3 program (#DE-SC0006973) and the Western Regional Center of the National Institute for

4 Climatic Change Research, and by the National Science Foundation (DEB#1021559). Any

5 opinions, findings, and conclusions or recommendations expressed in this material are those of

6 the author(s) and do not necessarily reflect the views of the National Science Foundation.

## 8 References

9 Amundson, R., Stern, L., Baisden, T., and Wang, Y.: The isotopic composition of soil and soil-
10 respired CO 2, Geoderma, 82, 83-114, 1998.

11 Atkin, O. K. and Tjoelker, M. G.: Thermal acclimation and the dynamic response of plant
12 respiration to temperature, Trends in plant science, 8, 343-351, 2003.

13 Bachman, S., Heisler-White, J. L., Pendall, E., Williams, D. G., Morgan, J. A., and Newcomb, J.:
14 Elevated carbon dioxide alters impacts of precipitation pulses on ecosystem photosynthesis and
15 respiration in a semi-arid grassland, Oecologia, 162, 791-802, 2010.

16 Baldocchi, D., Tang, J., and Xu, L.: How switches and lags in biophysical regulators affect
17 spatial-temporal variation of soil respiration in an oak-grass savanna, Journal of Geophysical
18 Research: Biogeosciences (2005–2012), 111, 2006.

19 Barron-Gafford, G. A., Cable, J. M., Bentley, L. P., Scott, R. L., Huxman, T. E., Jenerette, G. D.,
20 and Ogle, K.: Quantifying the timescales over which exogenous and endogenous conditions
21 affect soil respiration, New Phytologist, 202, 442-454, 2014.

22 Birch, H.: The effect of soil drying on humus decomposition and nitrogen availability, Plant and
23 Soil, 10, 9-31, 1958.

24 Borken, W., Davidson, E., Savage, K., Gaudinski, J., and Trumbore, S. E.: Drying and wetting
25 effects on carbon dioxide release from organic horizons, Soil Science Society of America
26 Journal, 67, 1888-1896, 2003.

27 Borken, W. and Matzner, E.: Reappraisal of drying and wetting effects on C and N
28 mineralization and fluxes in soils, Global change biology, 15, 808-824, 2009.

29 Bouma, T. J. and Bryla, D. R.: On the assessment of root and soil respiration for soils of different
30 textures: interactions with soil moisture contents and soil CO2 concentrations, Plant and Soil,
31 227, 215-221, 2000.

32 Brennan, A.: Vegetation and Climate Change Alter Ecosystem Carbon Losses at the Prairie
33 Heating and CO2 Enrichment Experiment in Wyoming, Department of Botany, University of
34 Wyoming, 2013.



Cable, J. M., Ogle, K., Barron-Gafford, G. A., Bentley, L. P., Cable, W. L., Scott, R. L., Williams, D. G., and Huxman, T. E.: Antecedent conditions influence soil respiration differences in shrub and grass patches, Ecosystems, 16, 1230-1247, 2013.

Cable, J. M., Ogle, K., Lucas, R. W., Huxman, T. E., Loik, M. E., Smith, S. D., Tissue, D. T., Ewers, B. E., Pendall, E., and Welker, J. M.: The temperature responses of soil respiration in deserts: a seven desert synthesis, Biogeochemistry, 103, 71-90, 2011.

Cable, J. M., Ogle, K., Williams, D. G., Weltzin, J. F., and Huxman, T. E.: Soil texture drives responses of soil respiration to precipitation pulses in the Sonoran Desert: implications for climate change, Ecosystems, 11, 961-979, 2008.

Carrillo, Y., Dijkstra, F. A., LeCain, D., Morgan, J. A., Blumenthal, D., Waldron, S., and Pendall, E.: Disentangling root responses to climate change in a semiarid grassland, Oecologia, 1-13, 2014a.

Carrillo, Y., Dijkstra, F. A., Pendall, E., LeCain, D., and Tucker, C.: Plant rhizosphere influence on microbial C metabolism: the role of elevated CO2, N availability and root stoichiometry, Biogeochemistry, 117, 229-240, 2014b.

Carrillo, Y. and Pendall, E.: Combined effects of elevated CO2 and warming on soil carbon and microbial C use, in review.

Chou, W. W., Silver, W. L., Jackson, R. D., Thompson, A. W., and ALLEN-DIAZ, B.: The sensitivity of annual grassland carbon cycling to the quantity and timing of rainfall, Global Change Biology, 14, 1382-1394, 2008.

Cox, P. M.: Description of the TRIFFID dynamic global vegetation model, 2001.

Davidson, E. A. and Janssens, I. A.: Temperature sensitivity of soil carbon decomposition and feedbacks to climate change, Nature, 440, 165-173, 2006.

Davidson, E. A., Samanta, S., Caramori, S. S., and Savage, K.: The Dual Arrhenius and Michaelis–Menten kinetics model for decomposition of soil organic matter at hourly to seasonal time scales, Global Change Biology, 18, 371-384, 2012.

Fang, C. and Moncrieff, J. B.: A model for soil CO 2 production and transport 1:: Model development, Agricultural and Forest Meteorology, 95, 225-236, 1999.

Friedlingstein, P., Andrew, R. M., Rogelj, J., Peters, G., Canadell, J. G., Knutti, R., Luderer, G., Raupach, M. R., Schaeffer, M., and Van Vuuren, D. P.: Persistent growth of CO2 emissions and implications for reaching climate targets, Nature geoscience, 7, 709, 2014.

Haberman, R.: Elementary applied partial differential equations (Third Edition), Prentice Hall Englewood Cliffs, NJ, 1998.

Hanson, P., Edwards, N., Garten, C., and Andrews, J.: Separating root and soil microbial contributions to soil respiration: a review of methods and observations, Biogeochemistry, 48, 115-146, 2000.

Hashimoto, S., Carvalhais, N., Ito, A., Migliavacca, M., Nishina, K., and Reichstein, M.: Global spatiotemporal distribution of soil respiration modeled using a global database, Biogeosciences, 12, 4121-4132, 2015.



Hillel, D.: Environmental soil physics: Fundamentals, applications, and environmental considerations, Academic press, 1998.

Hui, D. and Luo, Y.: Evaluation of soil CO2 production and transport in Duke Forest using a process-based modeling approach, Global Biogeochemical Cycles, 18, 2004.

Huxman, T. E., Snyder, K. A., Tissue, D., Leffler, A. J., Ogle, K., Pockman, W. T., Sandquist, D. R., Potts, D. L., and Schwinning, S.: Precipitation pulses and carbon fluxes in semiarid and arid ecosystems, Oecologia, 141, 254-268, 2004.

Jarvis, P., Rey, A., Petsikos, C., Wingate, L., Rayment, M., Pereira, J., Banza, J., David, J., Miglietta, F., and Borghetti, M.: Drying and wetting of Mediterranean soils stimulates decomposition and carbon dioxide emission: the "Birch effect", Tree physiology, 27, 929-940, 2007.

Jasoni, R. L., Smith, S. D., and Arnone, J. A.: Net ecosystem CO2 exchange in Mojave Desert shrublands during the eighth year of exposure to elevated CO2, Global Change Biology, 11, 749-756, 2005.

Kayler, Z. E., Sulzman, E. W., Rugh, W. D., Mix, A. C., and Bond, B. J.: Characterizing the impact of diffusive and advective soil gas transport on the measurement and interpretation of the isotopic signal of soil respiration, Soil Biology and Biochemistry, 42, 435-444, 2010.

Kemp, P. R., Reynolds, J. F., Pachepsky, Y., and Chen, J. L.: A comparative modeling study of soil water dynamics in a desert ecosystem, Water Resources Research, 33, 73-90, 1997.

Lee, X., Wu, H. J., Sigler, J., Oishi, C., and Siccama, T.: Rapid and transient response of soil respiration to rain, Global Change Biology, 10, 1017-1026, 2004.

Lloyd, J. and Taylor, J.: On the temperature dependence of soil respiration, Functional ecology, 315-323, 1994.

Luo, Y. and Zhou, X.: Soil respiration and the environment, Academic press, 2010.

Maggi, F. and Riley, W. J.: Transient competitive complexation in biological kinetic isotope fractionation explains nonsteady isotopic effects: Theory and application to denitrification in soils, Journal of Geophysical Research: Biogeosciences, 114, 2009.

Mathworks: The MathWorks, Inc. Natick, Massachusetts, USA MATLAB and Statistics Toolbox Release 2016b. 2016.

Meisner, A., Bååth, E., and Rousk, J.: Microbial growth responses upon rewetting soil dried for four days or one year, Soil Biology and Biochemistry, 66, 188-192, 2013.

Moldrup, P., Olesen, T., Yoshikawa, S., Komatsu, T., and Rolston, D. E.: Three-porosity model for predicting the gas diffusion coefficient in undisturbed soil, Soil Science Society of America Journal, 68, 750-759, 2004.

Morgan, J. A., LeCain, D. R., Pendall, E., Blumenthal, D. M., Kimball, B. A., Carrillo, Y., Williams, D. G., Heisler-White, J., Dijkstra, F. A., and West, M.: C4 grasses prosper as carbon dioxide eliminates desiccation in warmed semi-arid grassland, Nature, 476, 202-205, 2011.





Moyes, A. B., Gaines, S. J., Siegwolf, R. T., and Bowling, D. R.: Diffusive fractionation complicates isotopic partitioning of autotrophic and heterotrophic sources of soil respiration, Plant, cell & environment, 33, 1804-1819, 2010.

Mueller, K. E., Blumenthal, D. M., Pendall, E., Carrillo, Y., Dijkstra, F. A., Williams, D. G., Follett, R. F., and Morgan, J. A.: Impacts of warming and elevated $CO_2$ on a semi-arid grassland are non-additive, shift with precipitation, and reverse over time, Ecology letters, 19, 956-966, 2016.

Nickerson, N. and Risk, D.: Physical controls on the isotopic composition of soil-respired $CO_2$, Journal of Geophysical Research: Biogeosciences (2005–2012), 114, 2009.

Ogle, K., Barber, J. J., Barron-Gafford, G. A., Bentley, L. P., Young, J. M., Huxman, T. E., Loik, M. E., and Tissue, D. T.: Quantifying ecological memory in plant and ecosystem processes, Ecology letters, 18, 221-235, 2015.

Ogle, K. and Pendall, E.: Isotope partitioning of soil respiration: A Bayesian solution to accommodate multiple sources of variability, Journal of Geophysical Research: Biogeosciences, 2015.

Ogle, K., Ryan, E., Dijkstra, F. A., and Pendall, E.: Quantifying and reducing uncertainties in estimated soil $CO_2$ fluxes with hierarchical data-model integration, Journal of Geophysical Research: Biogeosciences, 2016.

Ogle, K., Wolpert, R. L., and Reynolds, J. F.: Reconstructing plant root area and water uptake profiles, Ecology, 85, 1967-1978, 2004.

Oikawa, P., Grantz, D., Chatterjee, A., Eberwein, J., Allsman, L., and Jenerette, G.: Unifying soil respiration pulses, inhibition, and temperature hysteresis through dynamics of labile soil carbon and $O_2$, Journal of Geophysical Research: Biogeosciences, 119, 521-536, 2014.

Oleson, K., Lawrence, D., Bonan, G., Drewniak, B., Huang, M., Koven, C., Levis, S., Li, F., Riley, W., and Subin, Z.: Technical description of version 4.5 of the Community Land Model (CLM). Ncar Tech. Note NCAR/TN-503+ STR. National Center for Atmospheric Research, Boulder, CO, 422 pp. doi: 10.5065/D6RR1W7M., 2013.

Pendall, E., Bridgham, S., Hanson, P. J., Hungate, B., Kicklighter, D. W., Johnson, D. W., Law, B. E., Luo, Y., Megonigal, J. P., and Olsrud, M.: Below-ground process responses to elevated $CO_2$ and temperature: A discussion of observations, measurement methods, and models, New Phytologist, 162, 311-322, 2004.

Pendall, E., Del Grosso, S., King, J., LeCain, D., Milchunas, D., Morgan, J., Mosier, A., Ojima, D., Parton, W., and Tans, P.: Elevated atmospheric $CO_2$ effects and soil water feedbacks on soil respiration components in a Colorado grassland, Global Biogeochemical Cycles, 17, 2003.

Pendall, E., Heisler-White, J. L., Williams, D. G., Dijkstra, F. A., Carrillo, Y., Morgan, J. A., and LeCain, D. R.: Warming reduces carbon losses from grassland exposed to elevated atmospheric carbon dioxide, PloS one, 8, e71921, 2013.

Pendall, E., Leavitt, S. W., Brooks, T., Kimball, B. A., Pinter Jr, P. J., Wall, G. W., LaMorte, R. L., Wechsung, G., Wechsung, F., and Adamsen, F.: Elevated $CO_2$ stimulates soil respiration in a FACE wheat field, Basic and Applied Ecology, 2, 193-201, 2001.



Piao, S., Ciais, P., Friedlingstein, P., de Noblet-Ducoudre, N., Cadule, P., and al., e.: Spatiotemporal patterns of terrestrial carbon cycle during the 20th century, 23, 2009.

Reynolds, J. F., Kemp, P. R., Ogle, K., and Fernández, R. J.: Modifying the 'pulse–reserve' paradigm for deserts of North America: precipitation pulses, soil water, and plant responses, Oecologia, 141, 194-210, 2004.

Richards, L. A.: Capillary conduction of liquids through porous mediums, Physics, 1, 318-333, 1931.

Risk, D., Kellman, L., and Beltrami, H.: A new method for in situ soil gas diffusivity measurement and applications in the monitoring of subsurface CO2 production, Journal of Geophysical Research: Biogeosciences, 113, 2008.

Risk, D., Nickerson, N., Creelman, C., McArthur, G., and Owens, J.: Forced Diffusion soil flux: A new technique for continuous monitoring of soil gas efflux, Agricultural and forest meteorology, 151, 1622-1631, 2011.

Risk, D., Nickerson, N., Phillips, C., Kellman, L., and Moroni, M.: Drought alters respired δ 13 CO 2 from autotrophic, but not heterotrophic soil respiration, Soil Biology and Biochemistry, 50, 26-32, 2012.

Ryan, E. M., Ogle, K., Zelikova, T. J., LeCain, D. R., Williams, D. G., Morgan, J. A., and Pendall, E.: Antecedent moisture and temperature conditions modulate the response of ecosystem respiration to elevated CO2 and warming, Global Change Biology, 2015.

Sala, O., Lauenroth, W., and Parton, W.: Long-Term Soil Water Dynamics in the Shortgrass Steppe, Ecology, 73, 1175-1181, 1992.

Sala, O. E., Lauenroth, W., Parton, W., and Trlica, M.: Water status of soil and vegetation in a shortgrass steppe, Oecologia, 48, 327-331, 1981.

Schwinning, S., Sala, O. E., Loik, M. E., and Ehleringer, J. R.: Thresholds, memory, and seasonality: understanding pulse dynamics in arid/semi-arid ecosystems, Oecologia, 141, 191-193, 2004.

Šimůnek, J. and Suarez, D. L.: Modeling of carbon dioxide transport and production in soil: 1. Model development, Water Resources Research, 29, 487-497, 1993.

Simunek, J., Van Genuchten, M. T., and Sejna, M.: The HYDRUS-1D software package for simulating the one-dimensional movement of water, heat, and multiple solutes in variably-saturated media, University of California-Riverside Research Reports, 3, 1-240, 2005.

Šimůnek, J., van Genuchten, M. T., and Šejna, M.: Development and applications of the HYDRUS and STANMOD software packages and related codes, Vadose Zone Journal, 7, 587-600, 2008.

Šimůnek, J., Van Genuchten, M. T., and Šejna, M.: HYDRUS: Model use, calibration, and validation, Trans. Asabe, 55, 1261-1274, 2012.

Sitch, S., Huntingford, C., Gedney, N., Levy, P., Lomas, M., Piao, S., Betts, R., Ciais, P., Cox, P., and Friedlingstein, P.: Evaluation of the terrestrial carbon cycle, future plant geography and climate-carbon cycle feedbacks using five Dynamic Global Vegetation Models (DGVMs), Global Change Biology, 14, 2015-2039, 2008.

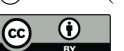



Sponseller, R. A.: Precipitation pulses and soil CO2 flux in a Sonoran Desert ecosystem, Global
Change Biology, 13, 426-436, 2007.
Tang, J., Baldocchi, D. D., Qi, Y., and Xu, L.: Assessing soil CO 2 efflux using continuous
measurements of CO 2 profiles in soils with small solid-state sensors, Agricultural and Forest
Meteorology, 118, 207-220, 2003.
Thomas, A. D., Hoon, S. R., and Linton, P. E.: Carbon dioxide fluxes from cyanobacteria crusted
soils in the Kalahari, Applied Soil Ecology, 39, 254-263, 2008.
Todd-Brown, K. E., Hopkins, F. M., Kivlin, S. N., Talbot, J. M., and Allison, S. D.: A
framework for representing microbial decomposition in coupled climate models,
Biogeochemistry, 109, 19-33, 2012.
Tucker, C. L., Bell, J., Pendall, E., and Ogle, K.: Does declining carbon-use efficiency explain
thermal acclimation of soil respiration with warming?, Global Change Biology, 19, 252-263,
13  2013.

Vargas, R., Baldocchi, D. D., Allen, M. F., Bahn, M., Black, T. A., Collins, S. L., Yuste, J. C.,
Hirano, T., Jassal, R. S., and Pumpanen, J.: Looking deeper into the soil: biophysical controls
and seasonal lags of soil CO2 production and efflux, Ecological Applications, 20, 1569-1582,
17  2010.

Vargas, R., Carbone, M. S., Reichstein, M., and Baldocchi, D. D.: Frontiers and challenges in
soil respiration research: from measurements to model-data integration, Biogeochemistry, 102,
20  1-13, 2011.

Xiang, S.-R., Doyle, A., Holden, P. A., and Schimel, J. P.: Drying and rewetting effects on C and
N mineralization and microbial activity in surface and subsurface California grassland soils, Soil
Biology and Biochemistry, 40, 2281-2289, 2008.
Xu, L., Baldocchi, D. D., and Tang, J.: How soil moisture, rain pulses, and growth alter the
response of ecosystem respiration to temperature, Global Biogeochemical Cycles, 18, 2004.
Zelikova, T. J., Williams, D. G., Hoenigman, R., Blumenthal, D. M., Morgan, J. A., and Pendall,
E.: Seasonality of soil moisture mediates responses of ecosystem phenology to elevated CO2 and
warming in a semi-arid grassland, Journal of Ecology, 103, 1119-1130, 2015.

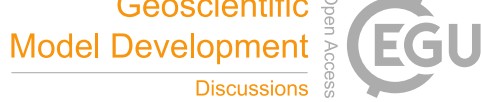

# 1 Figures

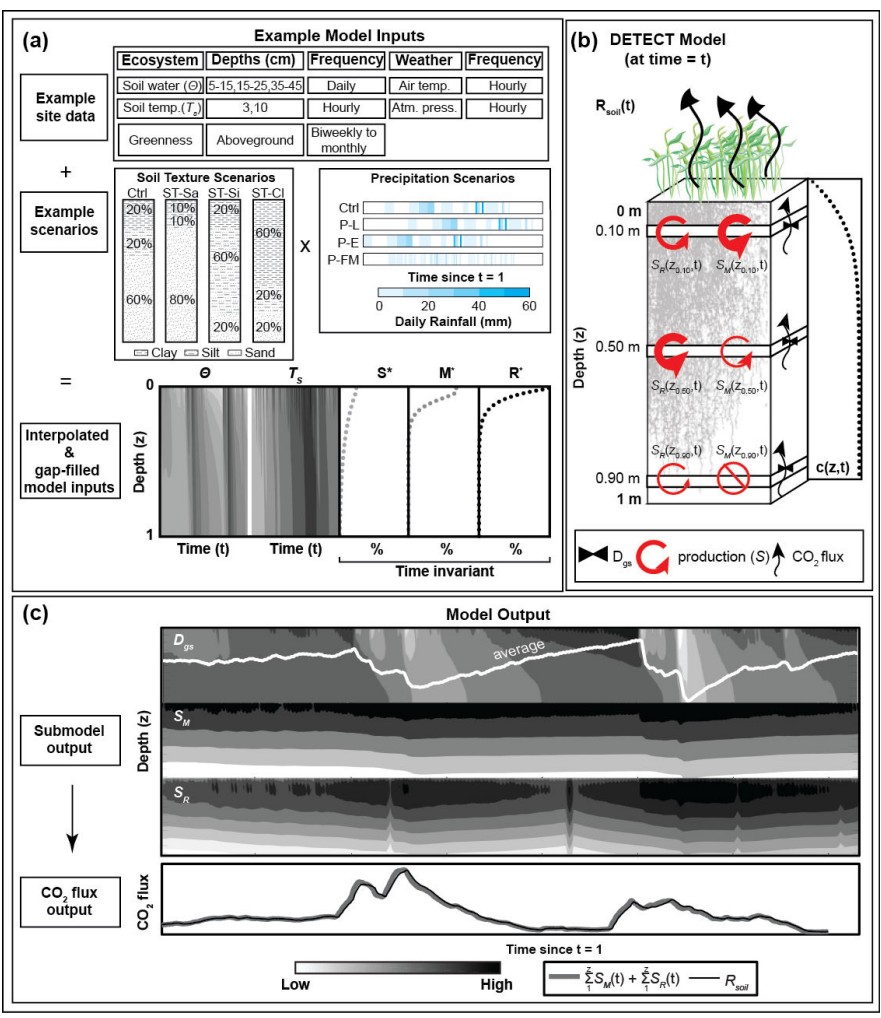

**Figure 1.** Graphical representation of (A) the required inputs to the DETECT model and the
associated scenarios implemented in this study, (B) the components of the DETECT model at a
particular time $t$, indicating depth-dependent production, $CO_2$ concentrations, and $CO_2$ fluxes,
and (C) example model outputs, such as temporally and spatially varying $CO_2$ diffusivity and
$CO_2$ production, and temporally varying bulk $CO_2$ fluxes. In this study, the model inputs in (A)
include interpolated and gap-filled environmental drivers derived from field measurements
(example from the Wyoming PHACE site) [*Ryan et al.*, 2015], combined with soil texture and
precipitation scenarios. The gap-filled data drives the DETECT model in (B), which numerically
solves for soil $CO_2$ at each depth $z$ and time $t$ given diffusivity and production submodels at each
$z$ and $t$, providing a complete picture of time- and depth-varying $CO_2$ production by microbes
($S_M$) and roots ($S_R$) production, diffusivity ($D_{gs}$), and bulk $CO_2$ fluxes as illustrated in (C).



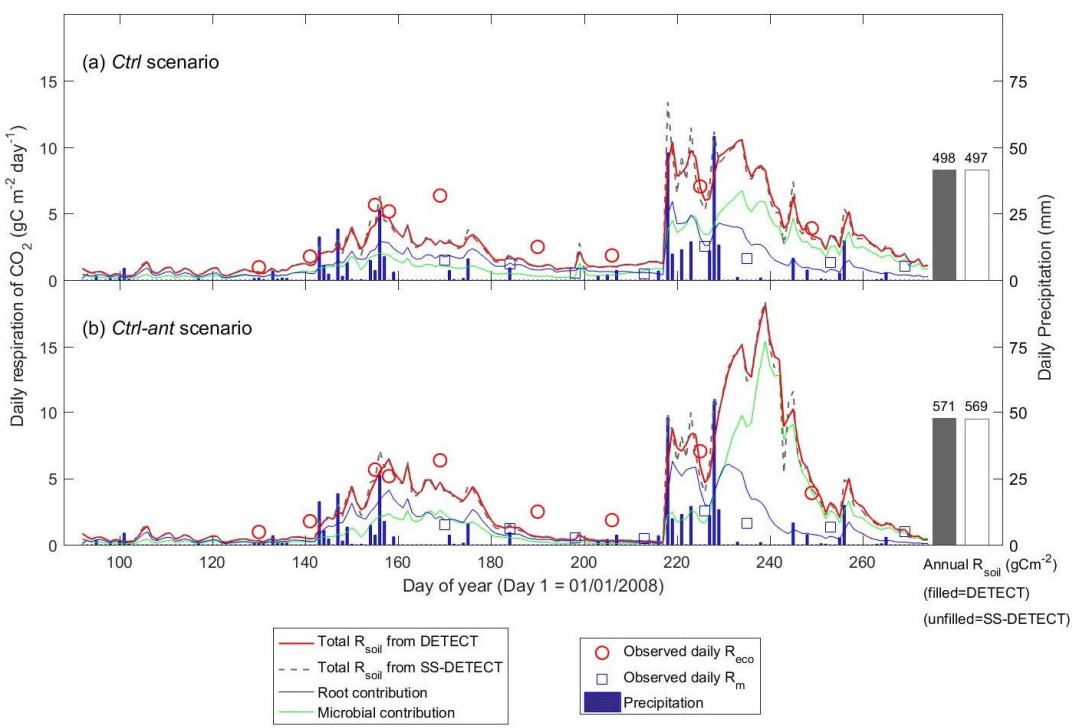

**Figure 2** Time-series of daily surface soil $CO_2$ fluxes ($R_{soil}$) predicted by the non-steady-state (DETECT) and steady-state (SS-
DETECT) models over the growing season (1$^{st}$ April – 30$^{th}$ September), based on the control scenarios (a) without (*Ctrl*) and (b) with
(*Ctrl-ant*) antecedent effects (see Table 2). Only total $R_{soil}$ is shown for the SS-DETECT model, whereas $R_{soil}$ and its root and
microbial contributions are shown for the DETECT model. The predicted fluxes are overlaid with observed ecosystem respiration
($R_{eco}$; $R_{soil}$ + aboveground plant respiration) and microbial respiration ($R_m$; based on plots where vegetation was removed).

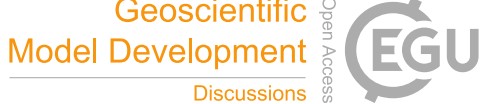



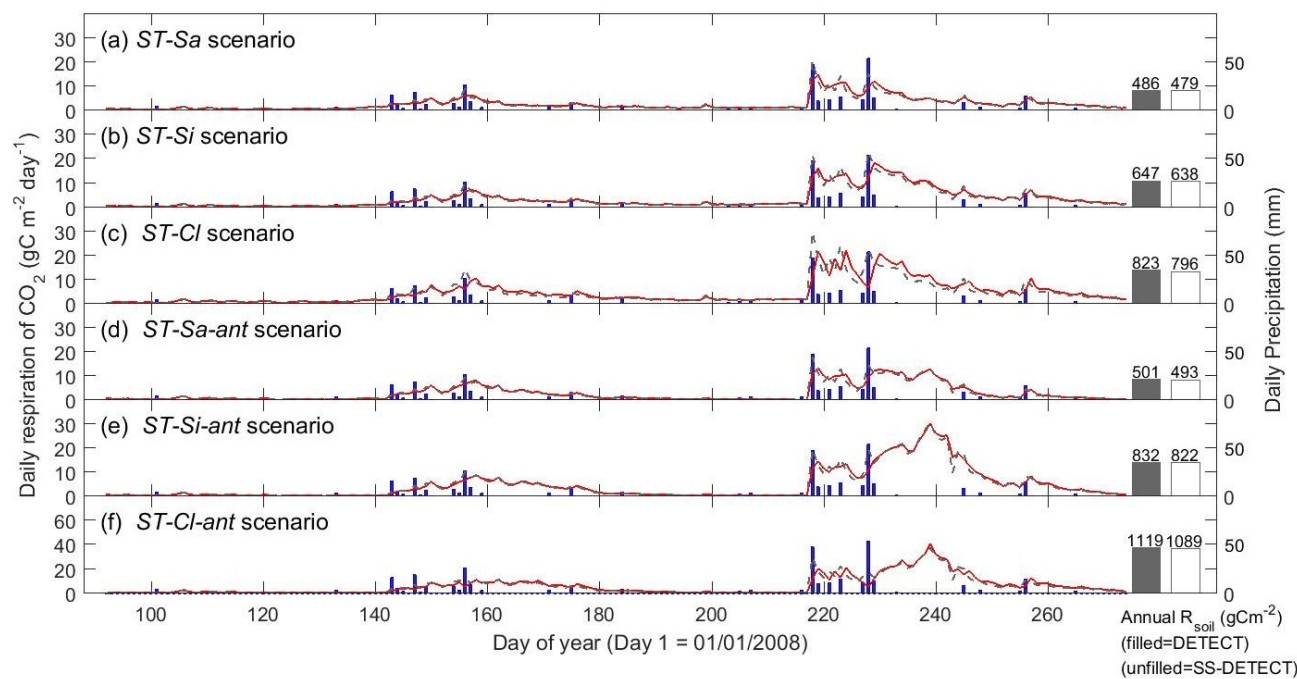

**Figure 3** Time-series of daily surface soil respiration ($R_{soil}$) predicted from the non-steady-state (NSS) DETECT model (red solid
lines) and the steady-state (SS-DETECT) model (grey dashed lines), for different soil texture scenarios. The first three scenarios are
the same as the control (*Ctrl*), except they assume a different soil texture: (a) more sandy soil, (b) more silty soil, or (c) more clayey
soil. Panels (d), (e), and (f) show the $R_{soil}$ predictions from the same soil texture scenarios as in (a)-(c), but also including antecedent
effects of soil moisture and temperature. See Table 2 for descriptions of each scenario. $R_{soil}$ predictions are overladied with daily
precipitation.





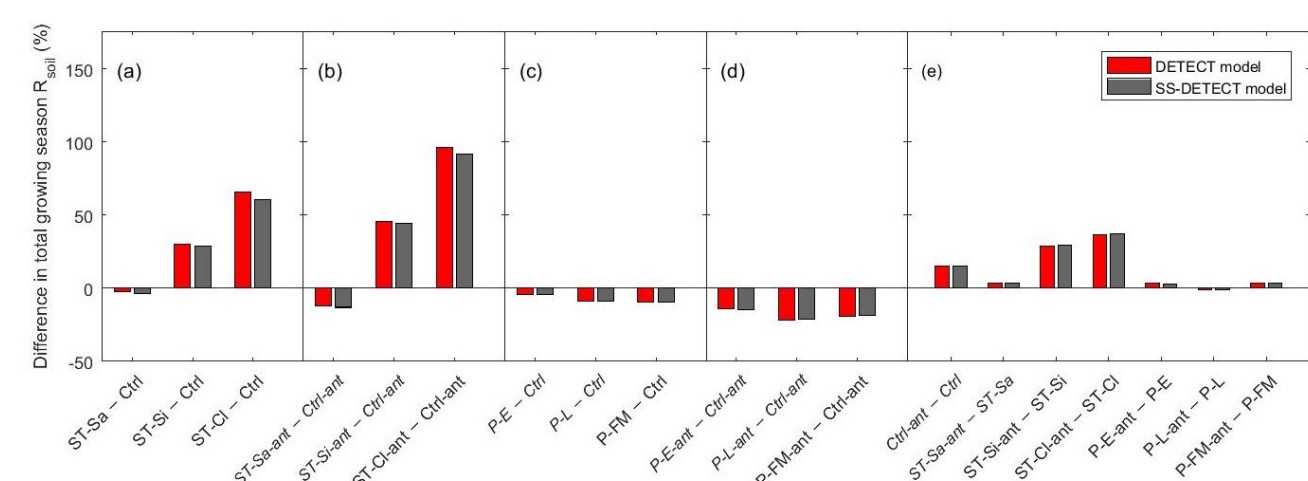

**Figure 4** Differences of total growing season (April-September) soil respiration ($R_{soil}$) as predicted by the non-steady-state (DETECT)
and steady-state (SS-DETECT) models, for different pairs of scenarios. Comparisons are grouped such that they quantify the effects of
(a) soil texture without antecedent effects, (b) soil texture with antecedent effects, (c) precipitation without antecedent effects, (d)
precipitation with antecedent effects, and (e) antecedent effects. See Table 2 for descriptions of each scenario .





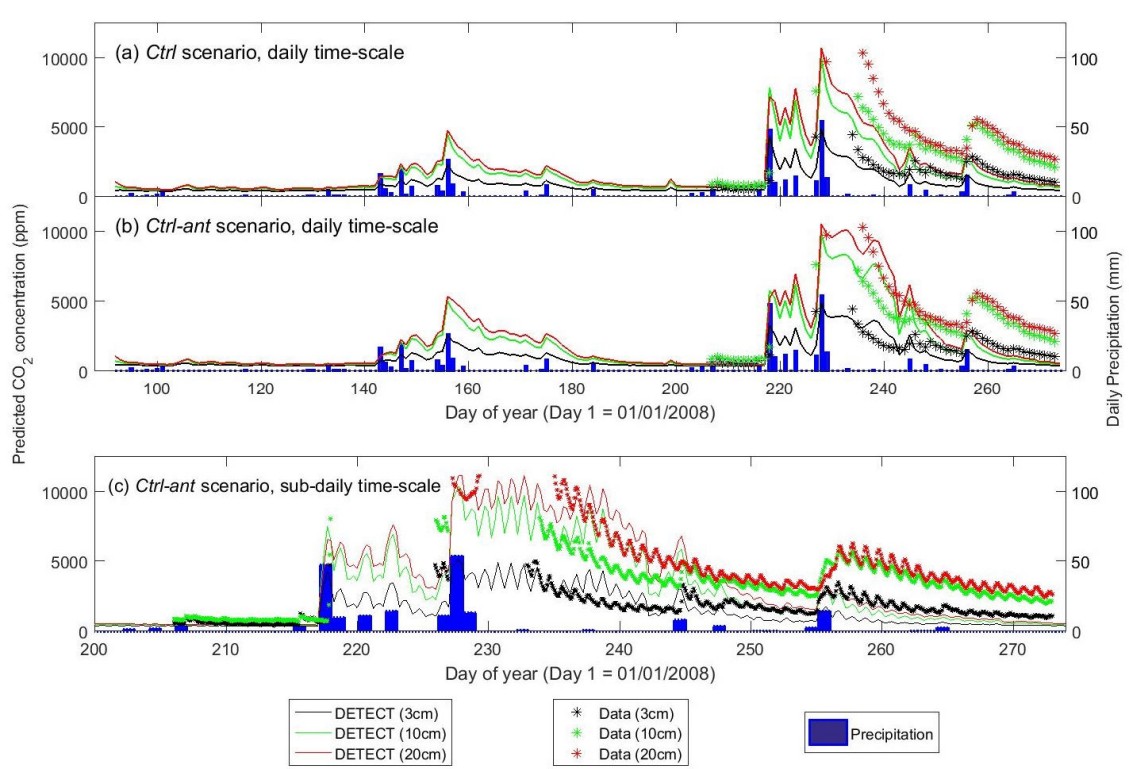

**Figure 5** Time-series of predicted versus observed soil $CO_2$ concentrations for depths 3 cm, 10 cm, and 20 cm, where the predictions
are based on the non-steady-state (NSS) DETECT model. Predicted [$CO_2$] is shown for the daily time-scale for the control scenarios
(a) without (*Ctrl*) and (b) with (*Ctrl-ant*) antecedent effects, and for (c) the subdaily (every 6 hours) time scale for the *Ctrl-ant*
scenario. Units are in parts per million (ppm).





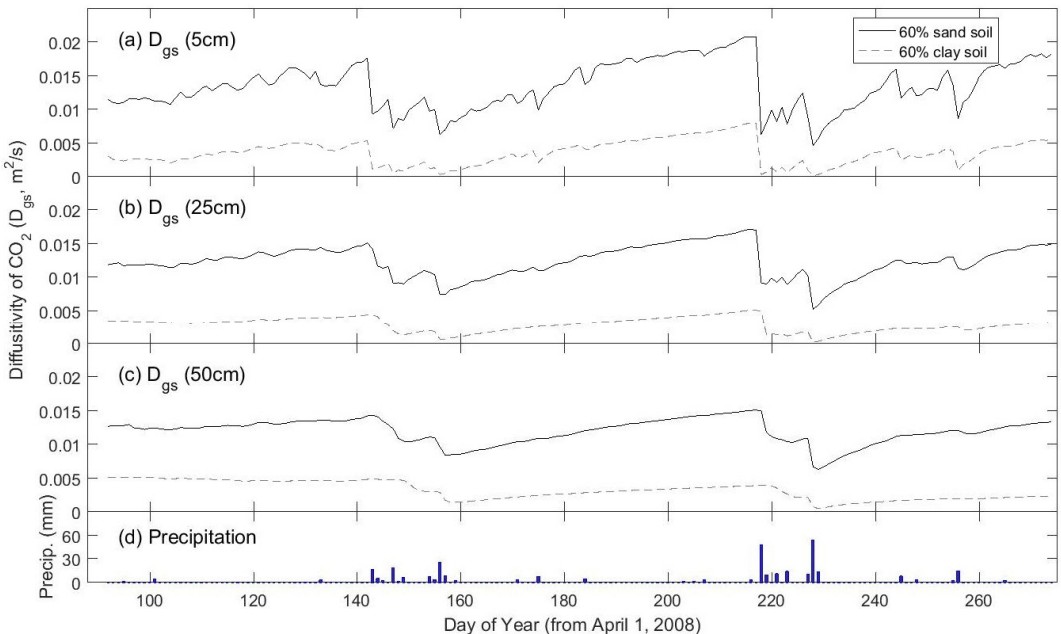

**Figure 6** Time series of how the modelled diffusivity of $CO_2$ ($D_{gs}$) at three different depths (5, 25, and 50 cm) varies between a predominantly sandy soil (solid line) and a predominantly clay soil (dashed line). Predictions are from the non-steady state (DETECT) model for the *Ctrl* (60% sand) and *ST-Cl* (60% clay) scenarios; see Table 2 for a description of the scenarios.





**Table 1** Summary of scalar parameters used in the non-steady-state (DETECT) model, arranged
into four groups: parameters unique to the microbial respiration submodel for $S_M(z,t)$ (group 1);
parameters unique to the root respiration submodel for $S_R(z,t)$ (group 2); parameters that are
shared between the $S_M(z,t)$ and $S_R(z,t)$ submodels (group 3); parameters used to calculate soil
$CO_2$ diffusivity, $D_{gs}$ (group 4).

| Symbol | Description | Value | Units | Eqn(s). |
|---|---|---|---|---|
| **Group 1** | | | | |
| $R*$ | Total root biomass C in a 1 m deep by 1 cm$^2$ soil column | 111.5 | mg C cm$^{-2}$ | 3 |
| $R_{RBase}$ | Root mass-base respiration rate at 10 ºC and mean environmental conditions | $6\times10^{-5}$ | mg C cm$^{-3}$ hr$^{-1}$ | 3 |
| $\alpha_{1(R)}$ | The effect of soil water content ($\theta$) on root respiration | 11.65 | unitless | 3, 4a |
| $\alpha_{2(R)}$ | The effect of antecedent $\theta$ ($\theta_R^{ant}$) on root respiration | 20.7 | unitless | 3, 4b |
| $\alpha_{3(R)}$ | The interactive effect of $\theta$ and $\theta_R^{ant}$ on root respiration | -164.2 | unitless | 3, 4c |
| **Group 2** | | | | |
| $S*$ | Total soil organic C in a 1 meter deep by 1 cm$^2$ soil column | 711.6 | mg C cm$^{-2}$ | 5 |
| $M*$ | Total microbial biomass C in a 1 meter deep by 1cm$^2$ column of soil | 12.3 | mg C cm$^{-2}$ | 5 |
| $V_{Base}$ | Value of $V_{max}$ at 10 ºC and mean environmental conditions | 0.0015 | mg C cm$^{-3}$ hr$^{-1}$ | 5, 6 |
| $\alpha_{1(M)}$ | The effect of $\theta$ on microbial respiration | 14.05 | unitless | 5, 6 |
| $\alpha_{2(M)}$ | The effect of antecedent $\theta$ ($\theta_M^{ant}$) on microbial respiration | 11.05 | unitless | 5, 6 |
| $\alpha_{3(M)}$ | The interactive effect of $\theta$ and $\theta_M^{ant}$ on microbial respiration | -87.6 | unitless | 5, 6 |
| $K_m$ | Michaelis-Menton half saturation constant | $10^{-5}$ | mg C cm$^{-3}$ hr$^{-1}$ | 5 |
| $CUE$ | Microbial carbon-use efficiency | 0.8 | mg C / mg C | 5 |
| $p$ | Fraction of soil organic C that is soluble | 0.004 | — | 7 |
| $D_{liq}$ | Diffusivity of soil C substrate in liquid | 3.17 | unitless | 7 |
| **Group 3** | | | | |
| $E_o*$ | Temperature sensitivity parameter, somewhat analogous to an energy of activation | 324.6 | Kelvin | 4c |
| $T_o$ | Temperature sensitivity-related parameter | 227.5 | Kelvin | 4c |
| $\alpha_4$ | The effect of antecedent soil temperature ($T_S^{ant}$) on root and microbial respiration | -4.7 | unitless | 4c |
| **Group 4** | | | | |
| $\alpha_{3(R)}$ | Absolute value of the slope of the line relating log($\Psi$) versus log($\theta$) | 4.547 | unitless | 2 |
| $BD$ | Soil bulk density | 1.12 | g cm$^{-3}$ | 2 |
| $\phi_{g100}$ | Air-filled porosity at soil water potential of -100 cm H$_2$0 (~10 kPa) | 18.16 | % | 2 |
| $PD$ | Particle density | | | |





**Table 2** Summary of quantities in the non-steady-state (DETECT) model that vary by depth only
($z$), or by depth ($z$) and time ($t$).  Those in group 1 represent input variables (derived prior to the
running of the DETECT model), while group 2 contains the modeled quantities (used as part of
the operation of the DETECT model).  Equation S1 can be found in Appendix S2 in the
supplemental material.

| Symbol | Description | Units | Eqn(s). |
|---|---|---|---|
| \multicolumn Group 1 | | | |
| $f_R(z)$ | A function describing the distribution by depth of root carbon. | unitless | S1 |
| $C_R(z,t)$ | The amount of root carbon. | mg C cm$^{-3}$ hr$^{-1}$ | 3, S1 |
| $f_S(z)$ | A function describing the distribution by depth of carbon from soil organic matter (SOM) | unitless | S1 |
| $C_{SOM}(z)$ | The amount of carbon from SOM. | mg C cm$^{-3}$ hr$^{-1}$ | 7, S1 |
| $f_M(z)$ | A function describing the distribution by depth of microbial carbon | unitless | S1 |
| $C_{MIC}(z)$ | The amount of microbial carbon. | mg C cm$^{-3}$ hr$^{-1}$ | 3, S1 |
| $\theta(z,t)$ | Soil water content | m$^3$ m$^{-3}$ | 3, 6, 7 |
| $\theta_R^{ant}(z,t)$ | Antedecent soil water content (used in $S_R$ function) calculated as a weighted average of soil water content from the previous 4 days.  The weights are w=(0.75,0.25,0,0). | m$^3$ m$^{-3}$ | 3 |
| $\theta_M^{ant}(z,t)$ | Antedecent soil water content (used in $S_M$ function) calculated as a weighted average of soil water content from the previous 4 days.  The weights are w=(0.2,0.6,0.2,0). | m$^3$ m$^{-3}$ | 6 |
| $T_S(z,t)$ | Soil temperature | Kelvin | 3, 6 |
| $T_S^{ant}(z,t)$ | Antedecent soil temperature calculated as a weighted average of soil temperature from the previous 4 weeks.  The weights are w=(0.25,0.25,0.25,0.25). | Kelvin | 3, 6 |
| \multicolumn Group 2 | | | |
| $c(z,t)$ | Total soil $CO_2$. | mg $CO_2$ m$^{-3}$ | 1 |
| $c_r(z,t)$ | Soil $CO_2$ derived from root sources. | mg $CO_2$ m$^{-3}$ | 1 |
| $S_r(z,t)$ | Source term describing the production of soil $CO_2$ from root respiration. | mg $CO_2$ m$^{-3}$ | 1 |
| $c_m(z,t)$ | Soil $CO_2$ derived from microbial sources. | mg $CO_2$ m$^{-3}$ | 1 |
| $S_m(z,t)$ | Source term describing the production of soil $CO_2$ from microbial respiration. | mg $CO_2$ m$^{-3}$ | 1 |
| $D_{gs}(z,t)$ | Diffusivity of soil $CO_2$ | m$^2$ s$^{-1}$ | 1, 2 |
| $\phi_g(z,t)$ | Air-filled soil porosity. | m$^3$ m$^{-3}$ | 1, 2 |
| $C_{SOL}(z,t)$ | The amount of soluble carbon from SOM. | mg C cm$^{-3}$ hr$^{-1}$ | 5, 7 |
| $V_{max}(z,t)$ | Maximum potential decomposition rate (microbial carbon). | mg C cm$^{-3}$ hr$^{-1}$ | 6 |
| $E_o(z,t)$ | Analogous to energy of activation. | Kelvin | 4c |





**Table 3** The scenario code, description, and summary of results associated with each model scenario; the 14 scenarios below were
applied to both the DETECT and SS-DETECT models. The scenarios involved a non-factorial combination of different soil texture,
precipitation regimes, and inclusion/exclusion of antecedent effects on the root and microbial $CO_2$ production rates.

| Scenario | Description | Primary result(s) |
|---|---|---|
| **Scenarios that assume no antecedent effects** | | |
| *Ctrl* (control) | Uses soil texture (sandy clay loam: 60% sand, 20% clay) and precipitation (for 2008) data from the PHACE site; $CO_2$ production only responds to concurrent environmental conditions. | $R_{soil}$ was very similar under SS and NSS soil $CO_2$ assumptions. |
| Soil texture scenarios | | |
| *ST-Sa* | Same as *Ctrl*, but the soil texture is set to sandy loam (80% sand, 10% clay). | For *ST-Cl*, $R_{soil}$ was greater in magnitude and more different under SS vs NSS conditions, due to NSS conditions producing greater $R_{soil}$ after a major precipitation event. The results are similar, but muted, for the *ST-Si* scenario. |
| *ST-Si* | Same as *Ctrl*, but the soil texture is set to silt loam (20% sand, 20% clay). | |
| *ST-Cl* | Same as *Ctrl*, but the soil texture is set to clay (20% sand, 60% clay). | |
| Precipitation scenarios | | |
| *P-E* | Same as *Ctrl*, but daily precipitation was shifted to occur one month earlier. | Varing the timing or magnitude of precipitation pulses had little effect on the magnitude of $R_{soil}$ or on the difference between SS and NSS predictions of $R_{soil}$. |
| *P-L* | Same as *Ctrl*, but daily precipitation was shifted to occur one month later. | |
| *P-FM* | Same as *Ctrl*, but daily precipitation was based on data from 2009, which is characterized by more frequent, smaller events. | |
| **Scenarios that incorporate antecedent effects on $CO_2$ production rates** | | |
| *Ctrl-ant* *ST-Sa-ant* *ST-Si-ant* *ST-Cl-ant* *P-E-ant* *P-L-ant* *P-FM-ant* | All scenarios parallel those described about, except both current and antecedent conditions (past soil water and past soil temperature) are used in the calculation of the source terms (i.e., root and microbial $CO_2$ production rates). | $R_{soil}$ was generally greater in magnitude under both SS and NSS conditions, especially for *ST-Si-ant* and *ST-Cl-ant* (relative to *ST-Si* and *ST-Cl*). |

