# Peer review of "Manuscript under review for journal Geosci. Model Dev."

_Geoscientific Model Development, 2017_

## Referee Comment (RC1) · F. Moyano (Referee) · 20 Oct 2017

Manuscript Review

Modeling soil CO2 production and transport with dynamic source and diffusion terms: Testing the steady-state assumption using DETECT v1.0

General comments

The manuscript describes a modeling study with the main objective of determining the

significance of non-steady states for determining and understanding soil respiration fluxes.

The paper is well written, with a logical structure and clear sentences. Apart from some minor comments, I find the abstract correctly describes the study. The introduction is also complete and informative. The same is valid for the methods, which require a detailed description given the amount of equations and assumptions used. Overall, the study succeeds as posing a defined set of questions and methods that are then used to obtain the results. By making the data and model code available the authors make a valuable contribution to the community.

The study is valid and provides some informative results as it is. However, the conclusions could be stronger with a slightly different focus. This considered, the below can be taken as suggestions for improvement unless a direct question or concern is stated.

Generally, the study could focus more on the specific question posed, i.e. when are NSS conditions relevant. It could discuss less the scenario comparisons not related to this, which make the article longer than required, since they are affected by a number of factors that are not analyzed properly. For example, some discussions on the response of Rsoil that are due to the source part of the model (SK) require a more detailed analysis of the functions used and could be left out. This includes precipitation effects not related to CO2 transport (as I comment below). On the other hand, a closer look at how concentrations change in soils, the amounts of air-filled pore-space and how much/fast CO2 is displaced upon wetting would be a nice addition.

Since the NSS and SS models do not differ in the production or source of CO2, the only difference should be where this CO2 remains after being produced. So it would be very informative to include the storage state variable, i.e. how much CO2 is in the soil. The total (Rsoil + storage) should be equal for both models (otherwise there is a mass-balance problem, as there is no other output flux for CO2). This also makes clearer that a NSS is always a temporal condition, so any difference (at daily or seasonal scales)

should be explained by changes in storage.

Because changes in CO2 storage can affect the net Rsoil, initial conditions that lead to a change in storage can affect the outcome. In that case it is better to get the model equilibrium to use as initial conditions instead of values fitted from data. Further questions and suggestions are given below as specific comments.

Specific comments (Numbers are for the page and line)

3/47 The term moreover here does not seem to connect the two sentences. The second does not add to the previous.

3/50-51 Integration time will surely also play a role, and NSS and SS differences will decrease for longer periods. Only a feedback of [CO2] on respiration or as a flux of dissolved inorganic C to groundwater (neither modeled) would result in different accumulated long-term Rsoil.

4/4 A comparison with fossil fuels is misleading if not better clarified. Rsoil is part of the fast C cycle. Not necessarily a net addition of C.

5/22 The hypothesis that the Rsoil spike after re-wetting is caused by pores filling with water and displacing CO2, is presented here, but not quite tested in the study.

10/15 How is $\Psi e(z)$ calculated? Is $\theta sat(z)$ not the same as $\varphi T$?

11/20 It is rather unusual to model the effects of volumetric moisture on respiration activity as an exponential function. This usually is an OK approximation only at the dry end of moisture content. Also strange is that when the $\theta$ and $\theta ant$ terms are 0 the function would equal 1. How does this make sense for a completely dry soil? There doesn't seem to be any information here or in the cited studies of why this function type was chosen (other than that it uses both current and antecedent inputs). Changes in the dynamics of soil moisture induced by modifying precipitation patterns will affect Rsoil largely as a result of the shape of this function. It's non-linear shape partly would explain why changing the frequency of precipitation with the same total amount would

lead to different seasonal fluxes. The discussion of those differences should include this.

13/eq.7 Here is another function that directly affects respiration activity and is strongly non-linearly related to moisture, as it includes the multiplier $\theta 3$. As with the f($\theta$, $\theta$ant) function, it changes Rsoil in response to changes in precipitation. This needs mentioning in the discussion.

14/4 'time-varying'

15/11 The expression is not an equality so it does not say how exactly Ndt is calculated.

15/eq.10 Would be nice to see this derived in the appendix.

16/9 Should actually cite the original derivation (by Cerling 1984)

16/eq.11 Since the only output is to the atmosphere, I'm guessing the depth terms are irrelevant and could be ignored in this model, unless the storage amount is of interest.

19/15 A reference for this procedure would be useful.

22/5 Parameter p probably has a strong impact on Rsoil. Uncertainties in this parameter would be informative.

22/10 Why without Cmic and CUE?

24 The paper makes texture a central point of the scenarios and discussion. However, the methods section did not make at all clear how texture affects the outcomes in the model. Presumably, texture is used in the HYDRUS model, thus affecting $\theta$. Maybe also affecting eq.2 (but it was not specified how). Given the discussion related to texture, this should be made clearer.

26/1-2 The first sentence here is not clear. What effects?

32/3-23 This paragraph almost seems too out of topic. While the model could be used to explain some of the dynamics of post-wetting Rsoil, this does not seem to be the

focus of the study. As commented above, these differences induced by changes in precipitation are strongly affected by the functions using $\theta$, which are not really analyzed here. Since the paper is rather long, it would seem preferable to leave a more careful analysis of this topic for another paper.

---

## Referee Comment (RC2) · Anonymous Referee #2 · 2 Nov 2017

In their manuscript, Ryan et al. study under which conditions soil CO2 production is in steady state with CO2 fluxes at the soil surface using a modelling approach, in which they focus on the effects of grain size and antecedent temperature and soil moisture conditions. Therefore, the authors present a new model of non-steady-state soil CO2 production (DETECT v1.0) and compare the model results with a simplified version of the model which assumes steady state conditions (no delay between sub-soil production of CO2 and CO2 the flux at the soil surface), by applying the model to an experimental site in Wyoming (PHACE).

[Figure]

The authors address some important questions: which environments factors control subsoil CO2 production and how can these processes be correctly simulated using a modelling approach. Overall, the manuscript is well-written and has a good structure. The abstract is informative and provides a good overview of the questions the authors address and a brief overview of the set-up of the study. The introduction gives an overview of the studied subject and existing knowledge, although it could be shortened in my opinion (see specific comments). The methodology provides a complete overview of the structure of the DETECT model and the equations it uses. At some points, however, some information is still missing (see specific comments). In the results section the authors present how they applied the model to assess the effect of different environmental factors supported by clear graphs. In the discussion section, in my opinion, the authors should focus more on the processes lying at the basis of their observations, such as the effect of soil moisture on microbial and root CO2 respiration (see specific comments). The fact that the authors provide the codes of their model together with a clear user manual increases the impact of their contribution.

Although I believe that this manuscript provides a valuable contribution to existing knowledge on how to model CO2 production in soils, I have some concerns and suggestions, as formulated below and in the specific comments.

A main concern is that most of the different amounts of modelled Rsoil between the scenarios arise from the effect that soil moisture has on the production of CO2 from both sources (roots and microbes), e.g. as shown in Figure 2 between days 220 and 240. The effect of soil moisture on CO2 production by both roots and microbes is regulated by equation 4a, which assumes an exponential relationship between $\theta$ and the amount of CO2 respiration. The conclusion that precipitation regime characteristics and/or including antecedent soil moisture and temperature conditions have an impact on the magnitude of the soil CO2 efflux (as formulated in the conclusion) is thus greatly affected by the use of eq. 4a. Using a different equation in which e.g. CO2 respiration rates decrease at very high soil moisture contents, might thus lead to

a different conclusion. E.g., using a soil moisture – respiration response function in which $CO_2$ production is inhibited at very high soil moisture levels might lead to less $CO_2$ respiration using NSS conditions. Therefore, I would encourage a more elaborate discussion (in addition to P33 L11-13) on the effect of this equation on your results or, better, an assessment of how including a different soil moisture - respiration response function affects the model results. Moreover, it should be more clearly explained how eq. 4a and 4b affect the produced $CO_2$ by roots and microbes, so this is more easily understandable for the reader.

The authors state that a correct simulation of $CO_2$ respiration in soils can improve modelling soil C processes. Therefore it would be interesting to assess the effect of the NSS vs SS approach on the total SOC pool: does the increase in $CO_2$ respiration using the NSS conditions lead to substantially decreasing SOC pool, or is this effect limited? Or in other words, is a correct simulation (NSS vs SS) of $CO_2$ respiration necessary in order to correctly model changes in the total SOC pool? Other suggestions and remarks are formulated in the specific comments below.

Specific comments

P 4 L17-18: in addition to delays due to $CO_2$ transport times, is also something known about the effect on additional $CO_2$ production (as this is one of the outcomes of the study)?

P5 L21: please clarify what you mean with 'displacement of $CO_2$'

P6-7 L18-13: In my opinion, this detailed explanation of your set-up can be formulated much shorter here, as this is explained in detail in the methods section

P8 L6-16: this is mostly a repeat of the last paragraph of the introduction and can be removed

P8 L17 – P9 L2: If you want to shorten the manuscript I would remove this part, as this is also clear from the introduction and the rest of the methods section.

P 11 L 20: please provide a reference for this equation

P11 eq 3: It's not clear to me how you obtained the value for RRbase, can this be stated explicitly?

P13 L16-17: how were these different values for the constants obtained? Please provide a reference if appropriate

P14 L13-14: please provide the value for the atmospheric $CO_2$ concentration that was used here.

P16 L17 – P17 L8: This paragraph belongs to the introduction, not to the materials and methods section.

P18 L8: please be more specific about the data that was created

P20 L9 – 22: It would be good if you could summarize the values of these parameters in a supplementary table, this would increase the readability and reduce the amount of text.

P21 L5 – 10: This can be removed in my opinion, this is also explained in the caption of the table

P21 L12 – 20: this is also explained in Appendix S1, this can be removed either in the text or in the appendix.

P22 L5 – 7: Here you state that you obtained a value for the parameter p as the ratio of Csol to Csom. However, in eq7 you state that you calculate Csol from the p parameter. This is rather confusing: is eq. 7 actually used in the model?

P22 L9: It is not clear how both parameters (Vbase and Km) were obtained through fitting the microbial respiration submodel to data. Please clarify. Also, why are Cmic and CUE left out?

P22 L16: please clarify how these values were adjusted.

P24 L6: I agree with the comment from reviewer 1 here: please clarify how texture affect the model outcomes.

P24 L15: please provide the amount of precipitation in 2009 here.

P25 L4 – 10: In my opinion, it's strange to already summarize the results before you have presented them, I would remove this paragraph as this is also clear from the rest of the results section

P 26 L7 – 8: the fact that Rsoil is larger when including the antecedent effect is likely to be a result of relationship between soil moisture and respiration (eq 4a), another formulation of this relationship could lead to a different results, see comment above.

P26 L9 – 11: You attribute the greater Rsoil to an increase in root respiration, while from Fig. 2 the increase in microbial respiration is even more significant and greatly contributes to the increase in total Rsoil. Why is this not mentioned in the text here?

P27 L11: I don't see how Fig. 3 shows that there is a greater root respiration.

P 27 L16 – 20: This formulation is confusing: in the first sentence you state that different precipitation scenarios led to little difference between Rsoil predicted using SS and NSS, while in the second sentence you state that precipitation regime affects the magnitude of Rsoil predicted by SS and NSS. Please re-formulate this.

P30 L6 – 8: from the data you show in the figures is seems like the difference in modelled Rsoil between SS and NSS at the timescale of a growing season is rather limited (e.g. the bars on the right side of Fig. 3), please clarify this. Also, in Fig. 3e I don't see substantial differences between SS and NSS after day 218.

P31 L1-4: I think this conclusion should be formulated less strong: the 'erroneous conclusions' depend on what you are modelling. Your results appear to show that using SS or NSS conditions does not have a large effect on e.g. the total amount of Rsoil over a whole growing season. However, if someone want to obtain detailed daily estimates of Rsoil on a (sub-)daily timescale, this is indeed important. I suggest the

authors re-formulate these sentences.

Technical comments

P2 L34: ... down to 1 m

P3 L51: ... precipitation inputs. The DETECT model...

P5 L8: ... coarse-grained

P5 L9: fast $CO_2$ diffusion rates

P5 L11: ... we expect coarse-grained soils

P5 L13: ... air-filled pore space

P6 L14: ... depth-invariant $CO_2$ production rates

P7 L 16: behavior and to (no comma)

P11 L12: remove the comma before 'and'

P18 L5: ... to 1 m depth

P20 L10: change to '(J previous time periods)'

P21 L20: if the SOC data you talk about is the same as shown in figure S4, you can refer to that figure here.

P23 L18:... 2013). These data were...

P30 L17: You could change this to: ... it may take about 15 minutes for a...

Figures and tables

Figure 1

Caption: everything after '... ,and temporally varying bulk $CO_2$ fluxes.' is redundant here. You could alternatively refer to the material and methods section where this is

also explained.

Figure 2

- Legend: add that root and microbial contributions are simulated using the DETECT model

- For easier comparison of the Rsoil between the two scenarios, you could indicate the Rsoil values shown in (a) on the bars in (b)

- Caption: 'see Table 2' should be Table 3 (also in Fig. 3, 4, 6, S1 and S2)

Figure 3

- Names of the scenarios in the sub-figures could be replaced with more intuitive names, followed by the scenario name between brackets, to increase readability.

- Include a legend for the grey and red lines

Figure 5

- Subplots (a) and (b): as you want to make the comparison between measurements and model results, you could choose only to show the timespan for which measurements are available (and show the entire timespan in the supplement)

- Legend: add 'depth': e.g. 3 cm depth

Table 1

- Instead of grouping the variables by 'Group 1', 'Group 2', etc, it would be more intuitive to provide the names to which the groups refer in the table (e.g. Group 1 = microbial submodel parameters, etc.)

- I would encourage the authors to include the references from where the parameter values were obtained in the table (where appropriate), now this is only described in the text

Table 3

- Bottom row, middle column: 'about' should be 'above'?

Supplementary information

Appendix S1

- Is there any evidence that root biomass varies between 0.5 and 1.5 times the amount measured in the middle of the growing season? Please include this.

- Last sentence of first paragraph: 'decays' should be 'declines'?

Figure S2

- Same remarks as for Fig. 3

---

## Author Comment (AC1) · 18 Jan 2018

Responses to reviewer RC1 (F. Moyano) on "Modelling soil CO2 production and transport with dynamic source and diffusion terms: Testing the steady-state assumption using DETECT v1.0" by Edmund Ryan et al.

Reviewer 's general comments The manuscript describes a modeling study with the main objective of determining the signiïṅẢcance of non-steady states for determining and understanding soil respiration ïṅĆuxes. The paper is well written, with a logi-

cal structure and clear sentences. Apart from some minor comments, We find the abstract correctly describes the study. The introduction is also complete and informative. The same is valid for the methods, which require a detailed description given the amount of equations and assumptions used. Overall, the study succeeds as posing a defined set of questions and methods that are then used to obtain the results. By making the data and model code available the authors make a valuable contribution to the community. The study is valid and provides some informative results as it is. However, the conclusions could be stronger with a slightly different focus. This considered, the below can be taken as suggestions for improvement unless a direct question or concern is stated.

Thank you for these positive comments.

Generally, the study could focus more on the specific question posed, i.e. when are NSS conditions relevant? It could discuss less the scenario comparisons not related to this, which make the article longer than required, since they are affected by a number of factors that are not analysed properly. For example, some discussions on the response of Rsoil that are due to the source part of the model (SK) require a more detailed analysis of the functions used and could be left out. This includes precipitation effects not related to CO2 transport (as We comment below). On the other hand, a closer look at how concentrations change in soils, the amounts of air-filled pore-space and how much/fast CO2 is displaced upon wetting would be a nice addition.

Thank you for these very helpful suggestions. We would like to keep the simulation experiments and the different scenarios, but we will amend the manuscript to better link them to the research questions. While there are many ways to create different simulation conditions (or scenarios), the scenarios that we selected were motivated by real data, from a real field site. The different scenarios lead to different soil conditions, thus allowing us to evaluate potential conditions or situations under which NSS conditions might be relevant.

Since the NSS and SS models do not differ in the production or source of CO2, the only difference should be where this CO2 remains after being produced. So it would be very informative to include the storage state variable, i.e. how much CO2 is in the soil. The total (Rsoil + storage) should be equal for both models (otherwise there is a mass balance problem, as there is no other output flux for CO2). This also makes clearer that a NSS is always a temporal condition, so any difference (at daily or seasonal scales) should be explained by changes in storage.

Thank you for this comment. A storage state variable for soil CO2 is already included in the DETECT model. After reading your comment in its entirety, we now realise that by a storage state variable, you mean total soil CO2 over the soil profile. You state the total (Rsoil + storage) should be equal for both models, but we think you meant to say that the total (Rsoil + change in storage) should be the same. We have checked this. Please see appendix S3 for details.

Because changes in CO2 storage can affect the net Rsoil, initial conditions that lead to a change in storage can affect the outcome. In that case it is better to get the model equilibrium to use as initial conditions instead of values fitted from data.

Thank you for this comment. We essentially did as the reviewer suggested. The DE-TECT model was run during the growing season of 2007 when measurements of soil CO2 concentrations were available for three different depths as well as above ground CO2 concentration. The initial values for this 2007 run (i.e. the soil CO2 concentrations for all depths) were estimated by fitting a simple function (described in appendix S2 of the supplemental material) to the CO2 data from near the start of the 2007 growing season. The initial conditions used for 2008 (i.e. soil CO2 concentration for 1st April, 2008) were taken from the soil CO2 simulated from DETECT from the final day of the growing season for 2007 (30th September, 2007). In a follow-up paper (Samuels-Crow et al., in revision), we found that it only takes about 1-2 weeks to achieve an equilibrium state, so the model output after this initial time period should not be affected by the initial conditions.

Reviewer 's specific comments Further questions and suggestions are given below as specific comments. Specific comments (Numbers are for the page and line) 3/47 The term moreover here does not seem to connect the two sentences. The second does not add to the previous. Agreed. We will remove the 'moreover'.

3/50-51 Integration time will surely also play a role, and NSS and SS differences will decrease for longer periods. Only a feedback of [CO2] on respiration or as a flux of dissolved inorganic C to groundwater (neither modeled) would result in different accumulated long-term Rsoil. We will include your above comments in the discussion. Thank-you for your insight.

4/4 A comparison with fossil fuels is misleading if not better clarified. Rsoil is part of the fast C cycle. Not necessarily a net addition of C. This comparison is purely to help the reader appreciate the size of the global scale Rsoil aggregated over a year. We will amend the text though to ensure that it is clearer.

5/22 The hypothesis that the Rsoil spike after re-wetting is caused by pores filling with water and displacing CO2, is presented here, but not quite tested in the study. This is something for a future study to address.

10/15 How is $\Psi e(z)$ calculated? Is $\theta sat(z)$ not the same as ÏŢT? The air-entry potential is calculated from measurements. We will add a reference to support this. The formulae we use are taken from the literature. We will add extra references, where required, to support these.

11/20 It is rather unusual to model the effects of volumetric moisture on respiration activity as an exponential function. This usually is an OK approximation only at the dry end of moisture content. Also strange is that when the $\theta$ and $\theta ant$ terms are 0 the function would equal 1. How does this make sense for a completely dry soil? There doesn't seem to be any information here or in the cited studies of why this function type was chosen (other than that it uses both current and antecedent inputs). Changes in the dynamics of soil moisture induced by modifying precipitation patterns will affect

Rsoil largely as a result of the shape of this function. It's non-linear shape partly would explain why changing the frequency of precipitation with the same total amount would lead to different seasonal fluxes. The discussion of those differences should include this. This point was raised by the other reviewer. Please see our response to that. In summary, $\theta$ at our field site never reached high enough values for respiration to decline. For completeness, however, we redid the control run using a respiration vs $\theta$ function that was bell shaped instead of exponential. We found that the time series of predicted soil respiration resulted in a very similar fit to the measurements. We will include extra discussion on this and all the points you make above.

13/eq.7 Here is another function that directly affects respiration activity and is strongly non-linearly related to moisture, as it includes the multiplier $\theta 3$. As with the f($\theta$, $\theta$ant) function, it changes Rsoil in response to changes in precipitation. This needs mentioning in the discussion. This formula was taken from the Davidson et al. (2012) paper (mentioned above this formula in the manuscript) which used field data to test its suitability. We have thus adopted this formula here, but we appreciate that there are other options. We will try to find space in the discussion to include your point, but the paper is already too long so this may not be possible given the other discussion points we need to include.

14/4 'time-varying' Thanks for spotting this. We'll change it.

15/11The expression is not an equality so it does not say how exactly Ndt is calculated. Fair point. We will clarify this in the revised version.

15/eq.10 Would be nice to see this derived in the appendix. There's nothing to derive. The expressions in equation 10 is just the discretised (or finite differenced) version of equation 1. We put equation 1 in this form in order to be able to numerically solve it. The Habernam book (that we reference) gives a great explanation of this.

16/9 Should actually cite the original derivation (by Cerling 1984) Okay, we will do.

16/eq.11 Since the only output is to the atmosphere, I'm guessing the depth terms are irrelevant and could be ignored in this model, unless the storage amount is of interest. I'm not sure I follow. This is the steady state solution to equation 1, so it has to involve a z term.

19/15 A reference for this procedure would be useful. Yes, of course.

22/5 Parameter p probably has a strong impact on Rsoil. Uncertainties in this parameter would be informative. Yes, you're right. Uncertainties are very important. For our study, we kept the parameters fixed, but when doing inverse modelling or uncertainty analysis we of course would want to assign a probability distribution to all parameters including this one.

22/10 Why without Cmic and CUE? The model fitting took place a period of time prior to the DETECT model being developed, and the formula (eqn 5) in that instance was used to estimate the soil respiration of CO2 from microbial sources. At the time, we did not have measurements of C_MIC or CUE so these were left out of that version of the submodel.

24 The paper makes texture a central point of the scenarios and discussion. However, the methods section did not make at all clear how texture affects the outcomes in the model. Presumably, texture is used in the HYDRUS model, thus affecting $\theta$. Maybe also affecting eq.2 (but it was not specified how). Given the discussion related to texture, this should be made clearer. Thanks for this. We'll update the methods to make this clearer.

26/1-2 The first sentence here is not clear. What effects? We'll use a different word to make it clearer what we mean.

32/3-23 This paragraph almost seems too out of topic. While the model could be used to explain some of the dynamics of post-wetting Rsoil, this does not seem to be the focus of the study. As commented above, these differences induced by changes

in precipitation are strongly affected by the functions using $\theta$, which are not really analyzed here. Since the paper is rather long, it would seem preferable to leave a more careful analysis of this topic for another paper. Thanks for this. We agree that the paper is rather long, so we'll remove this section as you suggest.

Please also note the supplement to this comment:
https://www.geosci-model-dev-discuss.net/gmd-2017-223/gmd-2017-223-AC1-supplement.pdf

---

## Author Comment (AC2) · 18 Jan 2018

Responses to reviewer RC2 (anonymous) on "Modelling soil CO2 production and transport with dynamic source and diffusion terms: Testing the steady-state assumption using DETECT v1.0" by Edmund Ryan et al.

Reviewer 's general comments In their manuscript, Ryan et al. study under which conditions soil CO2 production is in steady state with CO2 iňĆuxes at the soil surface using a modelling approach, in which they focus on the effects of grain size and antecedent

temperature and soil moisture conditions. Therefore, the authors present a new model of non-steady-state soil CO2 production (DETECT v1.0) and compare the model results with a simplified version of the model which assumes steady state conditions (no delay between subsoil production of CO2 and CO2 the flux at the soil surface), by applying the model to an experimental site in Wyoming (PHACE).

The authors address some important questions: which environments factors control subsoil CO2 production and how can these processes be correctly simulated using a modelling approach. Overall, the manuscript is well-written and has a good structure. The abstract is informative and provides a good overview of the questions the authors address and a brief overview of the set-up of the study. The introduction gives an overview of the studied subject and existing knowledge, although it could be shortened in my opinion (see specific comments). The methodology provides a complete overview of the structure of the DETECT model and the equations it uses. At some points, however, some information is still missing (see specific comments). In the results section, the authors present how they applied the model to assess the effect of different environmental factors supported by clear graphs. In the discussion section, in my opinion, the authors should focus more on the processes lying at the basis of their observations, such as the effect of soil moisture on microbial and root CO2 respiration (see specific comments). The fact that the authors provide the codes of their model together with a clear user manual increases the impact of their contribution.

Thank you for these comments. We will ensure we carefully and fully address the specific comments you refer to here.

Although I believe that this manuscript provides a valuable contribution to existing knowledge on how to model CO2 production in soils, I have some concerns and suggestions, as formulated below and in the specific comments. A main concern is that most of the different amounts of modelled Rsoil between the scenarios arise from the effect that soil moisture has on the production of CO2 from both sources (roots and microbes), e.g. as shown in Figure 2 between days 220 and 240. The effect of

soil moisture on CO2 production by both roots and microbes is regulated by equation 4a, which assumes an exponential relationship between $\theta$ and the amount of CO2 respiration. The conclusion that precipitation regime characteristics and/or including antecedent soil moisture and temperature conditions have an impact on the magnitude of the soil CO2 efflux (as formulated in the conclusion) is thus greatly affected by the use of eq. 4a. Using a different equation in which e.g. CO2 respiration rates decrease at very high soil moisture contents, might thus lead to a different conclusion. E.g., using a soil moisture – respiration response function in which CO2 production is inhibited at very high soil moisture levels might lead to less CO2 respiration using NSS conditions. Therefore, I would encourage a more elaborate discussion (in addition to P33 L11-13) on the effect of this equation on your results or, better, an assessment of how including a different soil moisture - respiration response function affects the model results. Moreover, it should be more clearly explained how eq. 4a and 4b affect the produced CO2 by roots and microbes, so this is more easily understandable for the reader.

Thank you for this comment, and we completely understand your concern. We have evaluated an alternative production versus soil water content function (i.e. an alternative to equation 4a). This alternative function simulates the production of soil CO2 versus soil water content as a bell shaped curved. In other words, production increases as soil water content increases but only up until an optimum soil water content value. When soil water is higher than this value, the production decreases. The formula for this alternative function is given in appendix S4 of the revised supplemental material. A graphical representation of the original function (equation 4a) and this alternative function (appendix S4) is shown in figure S7 of the supplemental material. For the Wyoming field site, however, soil water content never reached values that would have resulted in reduced ecosystem or soil CO2 flux or respiration rates . Hence, the graphical representation of this alternative soil CO2 production function shows production increasing for values of soil water content up to the optimum soil water content value. We ran the DETECT model for March-September, 2008, using this alternative produc-

tion versus soil water content function, and the time series of predicted soil respiration is given in figure S8. The predicted soil respiration fits the ecosystem respiration and microbial respiration measurements equally well, when comparing this figure with the corresponding figure in the manuscript (figure 2) which used an exponential function for the production versus soil water content relationship. Thus, either function (bell-shaped or exponential) is equally good at representing the relationship between soil CO2 production and soil water content at our well-drained, mid-latitude field site. To address your final point about how equations 4a and 4b affect the CO2 produced by roots and microbes, figure S9 in the supplemental material shows modelled S (production term in equation 1) against soil water content with the points colour coded according to three soil temperature bands.

The authors state that a correct simulation of CO2 respiration in soils can improve modelling soil C processes. Therefore it would be interesting to assess the effect of the NSS vs SS approach on the total SOC pool: does the increase in CO2 respiration using the NSS conditions lead to substantially decreasing SOC pool, or is this effect limited? Or in other words, is a correct simulation (NSS vs SS) of CO2 respiration necessary in order to correctly model changes in the total SOC pool?

This is an interesting question, but DETECT is not designed to follow slowly changing soil carbon pools. That is, we can't use DETECT to infer changes in the SOC pools (denoted C_SOM in model description). SOC is an input to the model (here, field data inform SOC, but other models focused on soil carbon pools could also be linked to DETECT), so the model does not predict changes to SOC. DETECT is most useful for understanding and modelling fast-time scale processes (e.g. fluxes) given known, measured, or hypothesized pool sizes (e.g. SOC). A future model development would be to couple DETECT to a dynamic soil C pool model, but this is unrealistic for the manuscript revision. Our view is that C_SOM will not differ between the SS and NSS models because the total amount of CO2 lost from the profiles was the same over of a growing season. The NSS is only important in clayey soils exposed to wetting/drying

cycles and only for CO2 efflux on periods of weeks-months. The C_SOM pool would take years to change.

Other suggestions and remarks are formulated in the specific comments below. Specific comments P 4 L17-18: in addition to delays due to CO2 transport times, is also something known about the effect on additional CO2 production (as this is one of the outcomes of the study)? We don't understand what the reviewer means by this question. In particular, we don't understand what 'effect on additional CO2 production' as an 'outcome of the study' means. The aim of the study is to determine if it is reasonable to assume that soil CO2 produced in the soil is respired at the same time point (steady-state), under different soil texture and precipitation regimes.

P5 L21: please clarify what you mean with 'displacement of CO2' We mean physical displacement. We will make this clearer in the revised version of the manuscript.

P6-7 L18-13: In my opinion, this detailed explanation of your set-up can be formulated much shorter here, as this is explained in detail in the methods section Okay, we will try to reduce this text for the revised version.

P8 L6-16: this is mostly a repeat of the last paragraph of the introduction and can be removed Fair point. We will remove this text or remove the part of last paragraph of the introduction where there is a repeat in text.

P8 L17 – P9 L2: If you want to shorten the manuscript I would remove this part, as this is also clear from the introduction and the rest of the methods section. Thank-you for this suggestion. We will consider this for the revised version.

P 11 L 20: please provide a reference for this equation Okay.

P11 eq 3: It's not clear to me how you obtained the value for RRbase, can this be stated explicitly? This is a parameter that we estimated. It is given in table 1. We will add a sentence to explain how we estimated it.

P13 L16-17: how were these different values for the constants obtained? Please pro-

vide a reference if appropriate Section 2.4.5 (page 21) explains how these parameters were estimated.

P14 L13-14: please provide the value for the atmospheric CO2 concentration that was used here. This is already given a few lines below equation 9c.

P16 L17 – P17 L8: This paragraph belongs to the introduction, not to the materials and methods section. We see your point. This text sets the scene for what comes after and it's specifically about the field site conditions. We will consider moving it to the introduction if it will fit (the introduction is already too long). Otherwise, we will shorten the text.

P18 L8: please be more specific about the data that was created We will improve the clarity of this sentence in the revised version.

P20 L9 – 22: It would be good if you could summarize the values of these parameters in a supplementary table, this would increase the readability and reduce the amount of text. These weight parameters are already included in table 1.

P21 L5 – 10: This can be removed in my opinion, this is also explained in the caption of the table Thanks for the suggestion. We're definitely looking for ways to reduce the length of the manuscript, and this portion of text is certainly a candidate.

P21 L12 – 20: this is also explained in Appendix S1, this can be removed either in the text or in the appendix. As above.

P22 L5 – 7: Here you state that you obtained a value for the parameter p as the ratio of Csol to Csom. However, in eq7 you state that you calculate Csol from the p parameter. This is rather confusing: is eq. 7 actually used in the model? We have measurement of C_SOL and C_SOM but only a very limited number. Equation (7) is simulating C_SOL for all 100 depths and all 732 time points. We'll amend the text to make this clearer.

P22 L9: It is not clear how both parameters (Vbase and Km) were obtained through fitting the microbial respiration submodel to data. Please clarify. Also, why are Cmic

and CUE left out? We'll modify the text to make this clearer. For your second question, the model fitting took place a period of time prior to the DETECT model being developed, and the formula (eqn 5) in that instance was use to estimate the soil respiration of CO2 from microbial sources. At the time, we did not have measurements of C_MIC or CUE so these were left out of that version of the submodel.

P22 L16: please clarify how these values were adjusted. Okay, we'll make this clearer for the revised version.

P24 L6: I agree with the comment from reviewer 1 here: please clarify how texture affect the model outcomes. The soil texture is an input into the HYDRUS model which was used to simulate soil water content and soil temperature for all depths and times. By varying the soil texture in HYDRUS, this resulted in different sets of soil water content and soil temperature values. We will make this clearer in the revised manuscript.

P24 L15: please provide the amount of precipitation in 2009 here. Okay, will do.

P25 L4 – 10: In my opinion, it's strange to already summarize the results before you have presented them, I would remove this paragraph as this is also clear from the rest of the results section This is just a stylist preference I think. We prefer to start the results section with bold statements about what the results revealed. The aim is to grab the attention of the reader. We take your point though, so we will consider dropping this text.

P26 L7 – 8: the fact that Rsoil is larger when including the antecedent effect is likely to be a result of relationship between soil moisture and respiration (eq 4a), another formulation of this relationship could lead to a different result, see comment above. I agree. You can think of the model with antecedent terms and the model without antecedent terms as two versions of the model. There are other versions that could be considered. A growing body of evidence (see introduction) suggests that antecedent conditions are important when simulating respiration (at different soil depths in this case). This is why we consider it here.

P26 L9 – 11: You attribute the greater Rsoil to an increase in root respiration, while from Fig. 2 the increase in microbial respiration is even more significant and greatly contributes to the increase in total Rsoil. Why is this not mentioned in the text here? We will address this in the revised version.

P27 L11: I don't see how Fig. 3 shows that there is a greater root respiration. Thanks for spotting this. An earlier version of figure 3 included the green and blue lines (like in fig.2). We will amend the text to remove any confusion.

P27 L16 – 20: This formulation is confusing: in the first sentence you state that different precipitation scenarios led to little difference between Rsoil predicted using SS and NSS, while in the second sentence you state that precipitation regime affects the magnitude of Rsoil predicted by SS and NSS. Please re-formulate this. We'll reword this text to improve clarity.

P30 L6 – 8: from the data you show in the figures is seems like the difference in modelled Rsoil between SS and NSS at the timescale of a growing season is rather limited (e.g. the bars on the right side of Fig. 3), please clarify this. Also, in Fig. 3e I don't see substantial differences between SS and NSS after day 218. We'll reword this text to make this clearer.

P31 L1-4: I think this conclusion should be formulated less strong: the 'erroneous conclusions' depend on what you are modelling. Your results appear to show that using SS or NSS conditions does not have a large effect on e.g. the total amount of Rsoil over a whole growing season. However, if someone want to obtain detailed daily estimates of Rsoil on a (sub-)daily timescale, this is indeed important. I suggest the authors re-formulate these sentences. Thank-you for this comment. We'll amend the text addressing the points you make.

Technical comments Thank-you for spotting all of these. We will fix all of these issues that you mentioned. P2 L34: ... down to 1 m P3 L51: ... precipitation inputs. The DETECT model... P5 L8: ... coarse-grained P5 L9: fast CO2 diffusion rates P5

L11: ... we expect coarse-grained soils P5 L13: ... air-filled pore space P6 L14: ... depth-invariant CO2 production rates P7 L 16: behavior and to (no comma) P11 L12: remove the comma before 'and' P18 L5: ... to 1 m depth P20 L10: change to '(J previous time periods)' P21 L20: if the SOC data you talk about is the same as shown in figure S4, you can refer to that figure here. P23 L18:... 2013). These data were... P30 L17: You could change this to: ... it may take about 15 minutes for a... Figures and tables Figure 1 Caption: everything after '... ,and temporally varying bulk CO2 fluxes.' is redundant here. You could alternatively refer to the material and methods section where this is also explained. Figure 2 - Legend: add that root and microbial contributions are simulated using the DETECT model - For easier comparison of the Rsoil between the two scenarios, you could indicate the Rsoil values shown in (a) on the bars in (b) - Caption: 'see Table 2' should be Table 3 (also in Fig. 3, 4, 6, S1 and S2) Figure 3 - Names of the scenarios in the sub-figures could be replaced with more intuitive names, followed by the scenario name between brackets, to increase readability. - Include a legend for the grey and red lines Figure 5 - Subplots (a) and (b): as you want to make the comparison between measurements and model results, you could choose only to show the timespan for which measurements are available (and show the entire timespan in the supplement) - Legend: add 'depth': e.g. 3 cm depth Table 1 -Instead of grouping the variables by 'Group1', 'Group2', etc, it would be more intuitive to provide the names to which the groups refer in the table (e.g. Group 1 = microbial submodel parameters, etc.) - I would encourage the authors to include the references from where the parameter values were obtained in the table (where appropriate), now this is only described in the text Table 3 - Bottom row, middle column: 'about' should be 'above'? Supplementary information Appendix S1 - Is there any evidence that root biomass varies between 0.5 and 1.5 times the amount measured in the middle of the growing season? Please include this. - Last sentence of first paragraph: 'decays' should be 'declines'? Figure S2 - Same remarks as for Fig. 3

Please also note the supplement to this comment:
https://www.geosci-model-dev-discuss.net/gmd-2017-223/gmd-2017-223-AC2-supplement.pdf

**Supplement:**

**Supporting Information**

This document contains the new text and figures that will be included in the revised version of the manuscript following the comments by the two GMD reviewers. Thus, the revised SI document will contain the original document + this new material.

**Appendix S3 Checking the mass balance of soil CO2 for the DETECT and DETECT-SS models.**

To mass balance of the DETECT and DETECT-SS models is theoretically guaranteed because equation 1 of the paper is actually the mass balance equation. The mass balance equation is defined as:

$$IN + PROD = OUT + ACC,$$

where for our model, IN and OUT are the inputs and outputs of CO2 from the boxes below and above it in the soil profile, PROD is the production of CO2, and ACC is the accumulation of CO2 over time. We can rearrange this mass balance equation to put it in the form of equation 1 from the manuscript:

$$ACC = (IN - OUT) + PROD$$

Where ACC is the dc/dt term from equation 1, (IN – OUT) is the d(Dgs*dC/dz)/dz term, and PROD is the S term. Similar comments can be made for the steady-state version of the DETECT model except that the ACC term in the above mass balance equation is equal to zero, i.e. there is no accumulation of soil CO2 over time (or the dC/dt term in equation 1 is set to zero).

As a practical check, we created a Matlab script which computes the total {Rsoil + change in CO2 storage} for both the DETECT and DETECT-SS models. Over the course of the year, {Rsoil + change in CO2 storage} was 497.1 gC/m$^2$ for the DETECT model and 497.1 gC/m$^2$ for the DETECT-SS model, under the control scenario.

**Appendix S4 Alternative formulations of the functions that describe how soil CO2 production changes with soil water content**

To test the robustness of the DETECT model, we try alternative formulations of the function $f$ that describe the production of soil CO2 from root and microbial sources for different soil water content ($\theta$) values. The formulation used in the paper (equation 4a) is an exponential function that depends on current and past soil water content. An alternative formulation is one where soil CO2 production increases as $\theta$ increases up to an optimum soil water content ($\theta_{opt}$) value. For values of $\theta$ greater than $\theta_{opt}$, soil CO2 production decreases. We represented this by a bell shaped curve:

$$f_R(\theta) = \frac{0.9}{\sqrt{0.01\pi}} \exp\left(-\frac{(\theta - \theta_{opt})^2}{0.01}\right)$$

$$f_M(\theta) = \frac{1}{\sqrt{0.0081\pi}} \exp\left(-\frac{(\theta - \theta_{opt})^2}{0.0081}\right)$$

where $f_R$ and $f_M$ refers to the function used as part of the calculations for the soil CO2 production from roots (R) and microbial (M) sources, and where $\theta_{opt} = 0.3$.

[Figure]

[Figure]

**Figure S7** The panel on the left of this figure shows the graphical representation of the equation that models RrBase as a function of θ, where RrBase is the base rate of Rr (production of root CO2) and θ is the soil water content.  This equation is one of the equations used to calculate microbial CO2 production, where the other equations use this RrBase value to allow production to vary according the specific temperature and microbial C content of a particular depth.  Here we show two options for modelling RrBase as a function of θ: (1) the exponential type function used in this analysis (see equation 4a of the paper); (2) an alternative to equation 4a, where RrBase increases as θ increases, but only up until a certain point given by $\theta_{opt}$; for values of θ higher than $\theta_{opt}$, RrBase decreases.  For the Wyoming field site that we use to make measurements, θ never got high enough that resulted in ecosystem respiration CO2 rates to decrease.  Hence, the graphical representation of this alternative RrBase function shows RrBase increasing for values of θ up to $\theta_{opt}$.  The description of the panel on the right of this figure is exactly the same as the left panel except that the y-axis shows RmBase (microbial production of CO2) instead of RrBase.

[Figure]

**Figure S8** The description for this figure is exactly the same as that of figure 2, except that the function used to simulate the production of soil CO2 from root and microbial sources is a bell shaped curve rather than an exponential function as used for the results of this analysis (see equation 4a of paper).

[Figure]

**Figure S9** Graphical representation of total production of CO2 from root and microbial sources (S, mg CO2 m-3 hr-1) as modelled by DETECT versus soil water content (θ, m3 m-3) at different soil depths.

Red=above 12°C
Black=between 4°C and 12°C.
Blue = below 4°C

---

## Author Response (AR1)

**Responses to reviewer RC1 (F. Moyano) on "Modelling soil CO2 production and transport with dynamic source and diffusion terms: Testing the steadystate assumption using DETECT v1.0" by Edmund Ryan et al.**

**Reviewer 's general comments**

The manuscript describes a modeling study with the main objective of determining the significance of non-steady states for determining and understanding soil respiration fluxes. The paper is well written, with a logical structure and clear sentences. Apart from some minor comments, We find the abstract correctly describes the study. The introduction is also complete and informative. The same is valid for the methods, which require a detailed description given the amount of equations and assumptions used. Overall, the study succeeds as posing a defined set of questions and methods that are then used to obtain the results. By making the data and model code available the authors make a valuable contribution to the community. The study is valid and provides some informative results as it is. However, the conclusions could be stronger with a slightly different focus. This considered, the below can be taken as suggestions for improvement unless a direct question or concern is stated.

Thank you for these positive comments.

**RC1 General Comment #1**

Generally, the study could focus more on the specific question posed, i.e. when are NSS conditions relevant? It could discuss less the scenario comparisons not related to this, which make the article longer than required, since they are affected by a number of factors that are not analysed properly. For example, some discussions on the response of Rsoil that are due to the source part of the model (SK) require a more detailed analysis of the functions used and could be left out. This includes precipitation effects not related to CO2 transport (as We comment below). On the other hand, a closer look at how concentrations change in soils, the amounts of air-filled pore-space and how much/fast CO2 is displaced upon wetting would be a nice addition.

**Author's Response**: Thank you for these very helpful suggestions. We would like to keep the simulation experiments and the different scenarios, but we will amend the manuscript to better link them to the research questions. While there are many ways to create different simulation conditions (or scenarios), the scenarios that we selected were motivated by real data, from a real field site. The different scenarios lead to different soil conditions, thus allowing us to evaluate potential conditions or situations under which NSS conditions might be relevant.

**Author's specific changes in the manuscript:**

1. We deleted 'and CO2 production rates' from page 7, line 21.

 In response to your comment on "some discussion on the response of Rsoil due to Sk being left out', we were unsure of all of the instances that you refer to so we have concentrated on deleting the parts related to 'precipitation effects not related to CO2 transport'. In particular, we deleted the entire second paragraph of section 4.3 reducing the length of the manuscript by a page.
 In response to your final point, we do already mention this in the second half of the second paragraph of section 4.2. We have added an extra sentence to make it clear about the change in CO2 concentration for the different soil texture types following a rain event.

**RC1 General Comment #2**

Since the NSS and SS models do not differ in the production or source of CO2, the only difference should be where this CO2 remains after being produced. So it would be very informative to include the storage state variable, i.e. how much CO2 is in the soil. The total (Rsoil + storage) should be equal for both models (otherwise there is a mass balance problem, as there is no other output flux for CO2). This also makes clearer that a NSS is always a temporal condition, so any difference (at daily or seasonal scales) should be explained by changes in storage.

**Author's Response**: Thank you for this comment. A storage state variable for soil CO2 is already included in the DETECT model. After reading your comment in its entirety, we now realise that by a storage state variable, you mean total soil CO2 over the soil profile. You state the total (Rsoil + storage) should be equal for both models, but we think you meant to say that the total (Rsoil + change in storage) should be the same. We have checked that they are the same. Please see appendix S3 for details.

**Author's specific changes in the manuscript:**

1. Appendix S3 in the supplementary information is new.

2. We have added new text in the manuscript at the bottom of page 8 in order to refer to appendix *S*3.

**RC1 General Comment #3**

Because changes in CO2 storage can affect the net Rsoil, initial conditions that lead to a change in storage can affect the outcome. In that case it is better to get the model equilibrium to use as initial conditions instead of values fitted from data.

**Author's Response**: Thank you for this comment. We essentially did as the reviewer suggested. The DETECT model was run during the growing season of 2007 when measurements of soil CO2 concentrations were available for three different depths as well as above ground CO2 concentration. The initial values for this 2007 run (i.e. the soil CO2 concentrations for all depths) were estimated by fitting a simple function (described in appendix S2 of the supplementary information) to the CO2 data from near the start of the 2007 growing season. The initial conditions used for 2008 (i.e. soil CO2 concentration for 1st April, 2008) were taken from the soil CO2 simulated from DETECT from the final day of the growing season for 2007 (30th September, 2007). In a follow-up paper (Samuels-Crow et al., in revision), we found that it only takes about 1-2 weeks to achieve an equilibrium state, so the model output after this initial time period should not be affected by the initial conditions.

**Author's specific changes in the manuscript:**

1. The text that directly following equation 9a-c (which describes how the initial conditions were estimated) has been updated to improve clarity.

2. Appendix S2 (referred to by the text in the manuscript that describe the initial conditions) has been updated to improve clarity.

**Reviewer 's specific comments**

Further questions and suggestions are given below as specific comments. Specific comments (Numbers are for the page and line)

**RC1 Specific Comment #1** (3/47) The term moreover here does not seem to connect the two sentences. The second does not add to the previous.

Agreed. We have removed 'moreover' from this sentence in the abstract.

**RC1 Specific Comment #2** (3/50-51) Integration time will surely also play a role, and NSS and SS differences will decrease for longer periods. Only a feedback of [CO2] on respiration or as a flux of dissolved inorganic C to groundwater (neither modeled) would result in different accumulated long-term Rsoil.

You are correct that NSS matters less for longer periods (e.g. years to decades), but over shorter periods (days-months) consideration of NSS condition is important. For example, when SS is not true this means that isotope methods (which can rely of this SS assumption) should not be used to partition soil respiration into its different components. This comment is similar to the second general comments from the second reviewer (RC2); please see the response this to see the changes we have made in the manuscript regarding this point.

**RC1 Specific Comment #3** (4/4) A comparison with fossil fuels is misleading if not better clarified. Rsoil is part of the fast C cycle. Not necessarily a net addition of C.

This comparison is purely to help the reader appreciate the size of the global scale Rsoil aggregated over a year, however we appreciate how this could be misleading. Hence, we have added text to this first sentence of the introduction to ensure that it is clearer.

**RC1 Specific Comment #4** (5/22) The hypothesis that the Rsoil spike after re-wetting is caused by pores filling with water and displacing CO2, is presented here, but not quite tested in the study.

This is something for a future study to address. We have included this soil CO2 transport process as a potential addition for a future version of the model (see the second paragraph of section 4.6.

**RC1 Specific Comment #5** (10/15) How is  $\Psi e(z)$  calculated? Is  $\theta sat(z)$  not the same as  $\phi T$ ?

The air-entry potential is calculated from measurements and the formulae we use are taken from the literature. We have added extra references in section 2.1.1 to support these.

**RC1 Specific Comment #6** (11/20) It is rather unusual to model the effects of volumetric moisture on respiration activity as an exponential function. This usually is an OK approximation only at the dry end of moisture content. Also strange is that when the  $\theta$  and  $\theta$ ant terms are 0 the function would equal 1. How does this make sense for a completely dry soil? There doesn't seem to be any information here or in the cited studies of why this function type was chosen (other than that it uses both current and antecedent inputs). Changes in the dynamics of soil moisture induced by modifying precipitation patterns will affect Rsoil largely as a result of the shape of this function. It's non-linear shape partly would explain why changing the frequency of precipitation with the same total amount would lead to different seasonal fluxes. The discussion of those differences should include this.

This point was raised by the other reviewer as one of the general comments. Please see our response to that which includes where we have made changes in the manuscript. In summary,  $\theta$  at our field site never reached high enough values for respiration to decline. For completeness, however, we redid the control run using a respiration vs  $\theta$  function that was bell shaped instead of exponential. We found that the time series of predicted soil respiration resulted in a very similar fit to the measurements.

**RC1 Specific Comment #7** (13/eq.7) Here is another function that directly affects respiration activity and is strongly non-linearly related to moisture, as it includes the multiplier  $\theta$ 3. As with the f( $\theta$ ,  $\theta$ ant)

function, it changes Rsoil in response to changes in precipitation. This needs mentioning in the discussion.

This formula was taken from the Davidson et al. (2012) paper (mentioned above this formula in the manuscript) which used field data to test its suitability. We have thus adopted this formula here, but we appreciate that there are other options. We have thus added text in the discussion (see second paragraph of section 4.6).

**RC1 Specific Comment #8 (14/4) 'time-varying'**

Changed.

**RC1 Specific Comment #9** (15/11) The expression is not an equality so it does not say how exactly Ndt is calculated.

Changed.

RC1 Specific Comment #10 (15/eq.10) Would be nice to see this derived in the appendix.

There's nothing to derive. The expressions in equation 10 are just the discretised (or finite differenced) version of equation 1. We put equation 1 in this form in order to be able to numerically solve it. The Haberman book (that we reference) gives a great explanation of this. New text has been added at the end of section 2.2 to make this clear.

**RC1 Specific Comment #11** (16/9) Should actually cite the original derivation (by Cerling 1984) *We have added this citation as well.*

**RC1 Specific Comment #12** (16/eq.11) Since the only output is to the atmosphere, I'm guessing the depth terms are irrelevant and could be ignored in this model, unless the storage amount is of interest.

I'm not sure I follow. This is the steady state solution to equation 1, so it has to involve a z term. In other words, there isn't one single C pool but 100 different C pools, one for each depth.

RC1 Specific Comment #13 (19/15) A reference for this procedure would be useful.

Reference added towards the end of section 2.4.3.

**RC1 Specific Comment #14** (22/5) Parameter p probably has a strong impact on Rsoil. Uncertainties in this parameter would be informative.

Yes, you're right. Uncertainties are very important. For our study, we kept the parameters fixed, but when doing inverse modelling or uncertainty analysis we of course would want to assign a probability distribution to all parameters including this one. We have added this point as an additional improvement to consider to future versions of DETECT (second paragraph of section 4.6).

RC1 Specific Comment #15 (22/10) Why without Cmic and CUE?

The model fitting took place a period of time prior to the DETECT model being developed, and the formula (eqn 5) in that instance was used to estimate the soil respiration of CO2 from microbial sources. At the time, we did not have measurements of C\_MIC or CUE so these were left out of that version of the submodel. We have added a sentence following the sentence in question (start of third paragraph of section 2.4.5) to give the reasons stated above.

**RC1 Specific Comment #16** (24) The paper makes texture a central point of the scenarios and discussion. However, the methods section did not make at all clear how texture affects the outcomes in the model. Presumably, texture is used in the HYDRUS model, thus affecting  $\theta$ . Maybe

also affecting eq.2 (but it was not specified how). Given the discussion related to texture, this should be made clearer.

Thanks for this. We've added two extra sentences to the methods (end of the first paragraph of section 2.5) to make this clearer.

RC1 Specific Comment #17 (26/1-2) The first sentence here is not clear. What effects?

We'll use a different word to make it clearer what we mean.

**RC1 Specific Comment #18** (32/3-23) This paragraph almost seems too out of topic. While the model could be used to explain some of the dynamics of post-wetting Rsoil, this does not seem to be the focus of the study. As commented above, these differences induced by changes in precipitation are strongly affected by the functions using  $\theta$ , which are not really analyzed here. Since the paper is rather long, it would seem preferable to leave a more careful analysis of this topic for another paper.

Thanks for this. We agree that the paper is rather long, so we have removed this second paragraph in section 4.3 as you suggested.

**Responses to reviewer RC2 (anonymous) on "Modelling soil CO2 production and transport with dynamic source and diffusion terms: Testing the steadystate assumption using DETECT v1.0" by Edmund Ryan et al.**

**Reviewer 's general comments**

In their manuscript, Ryan et al. study under which conditions soil CO2 production is in steady state with CO2 fluxes at the soil surface using a modelling approach, in which they focus on the effects of grain size and antecedent temperature and soil moisture conditions. Therefore, the authors present a new model of non-steady-state soil CO2 production (DETECT v1.0) and compare the model results with a simplified version of the model which assumes steady state conditions (no delay between subsoil production of CO2 and CO2 the flux at the soil surface), by applying the model to an experimental site in Wyoming (PHACE).

**RC2 General Comment #1**

The authors address some important questions: which environments factors control subsoil CO2 production and how can these processes be correctly simulated using a modelling approach. Overall, the manuscript is well-written and has a good structure. The abstract is informative and provides a good overview of the questions the authors address and a brief overview of the set-up of the study. The introduction gives an overview of the studied subject and existing knowledge, although it could be shortened in my opinion (see specific comments). The methodology provides a complete overview of the structure of the DETECT model and the equations it uses. At some points, however, some information is still missing (see specific comments). In the results section, the authors present how they applied the model to assess the effect of different environmental factors supported by clear graphs. In the discussion section, in my opinion, the authors should focus more on the processes lying at the basis of their observations, such as the effect of soil moisture on microbial and root CO2 respiration (see specific comments). The fact that the authors provide the codes of their model together with a clear user manual increases the impact of their contribution.

**Author's Response**: Thank you for these comments. We will ensure we carefully and fully address the specific comments you refer to here.

Although I believe that this manuscript provides a valuable contribution to existing knowledge on how to model CO2 production in soils, I have some concerns and suggestions, as formulated below and in the specific comments.

**RC2 General Comment #2**

A main concern is that most of the different amounts of modelled Rsoil between the scenarios arise from the effect that soil moisture has on the production of CO2 from both sources (roots and microbes), e.g. as shown in Figure 2 between days 220 and 240. The effect of soil moisture on CO2 production by both roots and microbes is regulated by equation 4a, which assumes an exponential relationship between θ and the amount of CO2 respiration. The conclusion that precipitation regime characteristics and/or including antecedent soil moisture and temperature conditions have an impact on the magnitude of the soil CO2 efflux (as formulated in the conclusion) is thus greatly affected by the use of eq. 4a. Using a different equation in which e.g. CO2 respiration rates decrease at very high soil moisture contents, might thus lead to a different conclusion. E.g., using a soil moisture – respiration response function in which CO2 production is inhibited at very high soil moisture levels might lead to less CO2 respiration using NSS conditions. Therefore, I would encourage a more elaborate discussion (in addition to P33 L11-13) on the effect of this equation on

your results or, better, an assessment of how including a different soil moisture - respiration response function affects the model results. Moreover, it should be more clearly explained how eq. 4a and 4b affect the produced CO2 by roots and microbes, so this is more easily understandable for the reader.

Author's Response: Thank you for this comment, and we completely understand your concern. We have evaluated an alternative production versus soil water content function (i.e. an alternative to equation 4a). This alternative function simulates the production of soil CO2 versus soil water content as a bell shaped curved. In other words, production increases as soil water content increases but only up until an optimum soil water content value. When soil water is higher than this value, the production decreases. The formula for this alternative function is given in appendix S4 of the revised supplementary information. A graphical representation of the original function (equation 4a) and this alternative function (appendix S4) is shown in figure S8 of the supplementary information. For the Wyoming field site, however, soil water content never reached values that would have resulted in reduced ecosystem or soil CO2 flux or respiration rates. Hence, the graphical representation of this alternative soil CO2 production function shows production increasing for values of soil water content up to the optimum soil water content value. We ran the DETECT model for March-September, 2008, using this alternative production versus soil water content function, and the time series of predicted soil respiration is given in figure S9. The predicted soil respiration fits the ecosystem respiration and microbial respiration measurements equally well, when comparing this figure with the corresponding figure in the manuscript (figure 2), which used an exponential function for the production versus soil water content relationship. Thus, either function (bell-shaped or exponential) is equally good at representing the relationship between soil CO2 production and soil water content at our welldrained, mid-latitude field site. To address your final point about how equations 4a and 4b affect the CO2 produced by roots and microbes, figure S10 in the supplementary information shows modelled S (production term in equation 1) against soil water content with the points colour coded according to three soil temperature bands.

**Author's specific changes in the manuscript:**

1. I have added text to the end of the second paragraph of section 2.1.2 in order to briefly describe the supplementary information to applying an alternative soil CO2 production versus soil water content function (i.e. an alternative version of equation 4a which has a bell-shaped form instead of an exponential form). The details of this alternative version of equation 4a as well as accompanying figures is given in the supplementary information (Appendix S4, figure S9).

2. In response to the final point of this comment, we refer to figure S10 in the new text at the end of third paragraph of 2.1.2.

**RC2 General Comment #3**

The authors state that a correct simulation of CO2 respiration in soils can improve modelling soil C processes. Therefore it would be interesting to assess the effect of the NSS vs SS approach on the total SOC pool: does the increase in CO2 respiration using the NSS conditions lead to substantially decreasing SOC pool, or is this effect limited? Or in other words, is a correct simulation (NSS vs SS) of CO2 respiration necessary in order to correctly model changes in the total SOC pool?

**Author's Response**: This is an interesting question, but DETECT is not designed to follow slowly changing soil carbon pools. That is, we can't use DETECT to infer changes in the SOC pools (denoted C\_SOM in model description). SOC is an input to the model (here, field data inform SOC, but other models focused on soil carbon pools could also be linked to DETECT), so the model does not predict changes to SOC. DETECT is most useful for understanding and modelling fast-time scale processes (e.g. fluxes) given known, measured, or hypothesized pool sizes (e.g. SOC). A future model development would be to couple DETECT to a dynamic soil C pool model, but this is unrealistic for the manuscript revision. Our view is that C\_SOM will not differ between the SS and NSS models because

the total amount of CO2 lost from the profiles was the same over of a growing season. The NSS is only important in clayey soils exposed to wetting/drying cycles and only for CO2 efflux on periods of weeks-months. The C\_SOM pool would take years to change.

**Author's specific changes in the manuscript:**

1. We have included a sentence as part of second paragraph of section 4.6 which describes potential future developments of the DETECT model. In particular, we describe the possibility of coupling DETECT with a dynamic soil C pool model.

**Other suggestions and remarks are formulated in the specific comments below.**

**RC2 Specific Comment #1** (P 4 L17-18): in addition to delays due to CO2 transport times, is also something known about the effect on additional CO2 production (as this is one of the outcomes of the study)?

We don't understand what the reviewer means by this question. In particular, we don't understand what 'effect on additional CO2 production' as an 'outcome of the study' means. The aim of the study is to determine if it is reasonable to assume that soil CO2 produced in the soil is respired at the same time point (steady-state), under different soil textures and precipitation regimes.

RC2 Specific Comment #2 (P5 L21: please clarify what you mean with 'displacement of CO2'

We have rephrased this sentence in the fourth paragraph of the introduction to make it clearer.

**RC2 Specific Comment #3** (P6-7 L18-13): In my opinion, this detailed explanation of your set-up can be formulated much shorter here, as this is explained in detail in the methods section

Thank-you for this. We have shortened the text in this second to last paragraph of the introduction.

**RC2 Specific Comment #4** (P8 L6-16): this is mostly a repeat of the last paragraph of the introduction and can be removed.

This first paragraph of the methods has been removed.

**RC2 Specific Comment #5** (P8 L17 – P9 L2): If you want to shorten the manuscript I would remove this part, as this is also clear from the introduction and the rest of the methods section.

This second paragraph of the methods has also been removed.

RC2 Specific Comment #6 P 11 L 20): please provide a reference for this equation

References are already provided, but I appreciate that they might have been missed. I have moved the relevant sentence to a better place in the text that following equations 4a-c.

**RC2 Specific Comment #7** P11 eq 3): It's not clear to me how you obtained the value for RRbase, can this be stated explicitly?

Thanks for highlighting this. RRbase should have been included with the list of alpha1, alpha2, etc... immediately following equation 4c. This has now been corrected. The description of how we estimated the values of the parameters is given in section 2.4.5. For RRbase, this is given in the third paragraph of section 2.4.5.

**RC2 Specific Comment #8** (P13 L16-17): how were these different values for the constants obtained? Please provide a reference if appropriate

Section 2.4.5 explains how these parameters were estimated. However, we appreciate that this should be made clearer to the reader. We have added a sentence to the end of the second to last paragraph of section 2.1.2.

**RC2 Specific Comment #9** (P14 L13-14): please provide the value for the atmospheric CO2 concentration that was used here.

This is already given a few lines below equation 9c.

**RC2 Specific Comment #10** (P16 L17 – P17 L8): This paragraph belongs to the introduction, not to the materials and methods section.

We see your point. It's challenging to know how to fit this first paragraph of section 2.4 into the introduction, so we have decided to delete it.

RC2 Specific Comment #11 (P18 L8): please be more specific about the data that was created

*This last sentence of section 2.4.2 has been replaced with two sentences that hopefully are a lot clearer.*

**RC2 Specific Comment #12** (P20 L9 – 22): It would be good if you could summarize the values of these parameters in a supplementary table, this would increase the readability and reduce the amount of text.

These weight parameters are already included in table 2. For clarity, we now refer to Table 2 in the final sentence of section 2.4.4.

**RC2 Specific Comment #13** (P21 L5 – 10): This can be removed in my opinion, this is also explained in the caption of the table.

Thanks for the suggestion. We have deleted this text from the first paragraph of section 2.4.5.

**RC2 Specific Comment #14** (P21 L12 – 20: this is also explained in Appendix S1, this can be removed either in the text or in the appendix.

We have only deleted the second half of this text as we felt it was important to define what the  $R^*$ ,  $M^*$ ,  $S^*$  and f functions are since they are listed in table 1.

**RC2 Specific Comment #15** (P22 L5 – 7: Here you state that you obtained a value for the parameter p as the ratio of Csol to Csom. However, in eq7 you state that you calculate Csol from the p parameter. This is rather confusing: is eq. 7 actually used in the model?

Thanks for spotting this. We have measurement of C\_SOL and C\_SOM but only a very limited number. Equation (7) is simulating C\_SOL for all 100 depths and all 732 time points. We'll amend the text to make sure there is no confusion.

**RC2 Specific Comment #16** (P22 L9: It is not clear how both parameters (Vbase and Km) were obtained through fitting the microbial respiration submodel to data. Please clarify. Also, why are Cmic and CUE left out?

We'll modify the text to make this clearer. For your second question, the model fitting took place a period of time prior to the DETECT model being developed, and the formula (eqn 5) in that instance was use to estimate the soil respiration of CO2 from microbial sources. At the time, we did not have measurements of C\_MIC or CUE so these were left out of that version of the submodel.

RC2 Specific Comment #17 (P22 L16: please clarify how these values were adjusted.

Okay, we'll make this clearer for the revised version.

**RC2 Specific Comment #18** (P24 L6: I agree with the comment from reviewer 1 here: please clarify how texture affect the model outcomes.

The soil texture is an input into the HYDRUS model which was used to simulate soil water content and soil temperature for all depths and times. By varying the soil texture in HYDRUS, this resulted in different sets of soil water content and soil temperature values. We will make this clearer in the revised manuscript.

**RC2 Specific Comment #19** (P24 L15: please provide the amount of precipitation in 2009 here. *Precipitation totals now given for 2008 and 2009.*

**RC2 Specific Comment #20** (P25 L4 – 10: In my opinion, it's strange to already summarize the results before you have presented them, I would remove this paragraph as this is also clear from the rest of the results section.

Agreed. We did this in order to grab the attention of the reader at the start of the results section, but this short text at the start of the results section is also a repeat of the abstract. Hence, we have deleted this text.

**RC2 Specific Comment #21** (P26 L7 – 8: the fact that Rsoil is larger when including the antecedent effect is likely to be a result of relationship between soil moisture and respiration (eq 4a), another formulation of this relationship could lead to a different result, see comment above.

We agree. You can think of the model with antecedent terms and the model without antecedent terms as two versions of the model. There are other versions that could be considered. A growing body of evidence (see introduction) suggests that antecedent conditions are important when simulating respiration (at different soil depths in this case). This is why we consider it here. The discussion section on antecedent conditions (section 4.4) goes into more detail about this and so hopefully addresses your comment.

**RC2 Specific Comment #22** (P26 L9 – 11: You attribute the greater Rsoil to an increase in root respiration, while from Fig. 2 the increase in microbial respiration is even more significant and greatly contributes to the increase in total Rsoil. Why is this not mentioned in the text here?

This is actually a plotting mistake. The green line is supposed to represent respiration from root sources and the blue line should represent from microbial sources. We have fixed this for the revised version of the manuscript.

**RC2 Specific Comment #23** (P27 L11: I don't see how Fig. 3 shows that there is a greater root respiration.

Thanks for spotting this. An earlier version of figure 3 included the green and blue lines (like in fig.2). We have amended this part of the text towards the end of section 3.2.

**RC2 Specific Comment #24** (P27 L16 – 20: This formulation is confusing: in the first sentence you state that different precipitation scenarios led to little difference between Rsoil predicted using SS and NSS, while in the second sentence you state that precipitation regime affects the magnitude of Rsoil predicted by SS and NSS. Please re-formulate this.

We're unsure why you think this is confusing, when the layout in section 3.3 is the same as section 3.2 (for example). In section 3.2, the first paragraph talks about SS vs NSS for soil texture, while the second paragraph talks about the magnitude of Rsoil for different soil textures. Section 3.3 is the same except for the different precipitation regimes + there is only one paragraph because of less to say on SS vs NSS. We have tried our best to reformulate this text at the start of section 3.3 by

deleting most of the first sentence and merging it with the second sentence. We hope this is sufficient.

**RC2 Specific Comment #25** (P30 L6 – 8: from the data you show in the figures is seems like the difference in modelled Rsoil between SS and NSS at the timescale of a growing season is rather limited (e.g. the bars on the right side of Fig. 3), please clarify this. Also, in Fig. 3e I don't see substantial differences between SS and NSS after day 218.

The wording in the text was 'greater differences' (compared to the control scenario) not 'substantial differences. The point we're making here is that Rsoil under SS vs NSS is larger for clay soils (whether we assume antecedent conditions or not). We do accept that with the figures being quite small, it's hard to see the daily differences. Hence, we've added an extra figure to the supplementary information to make this clearer (Fig. S3a, S3b).

**RC2 Specific Comment #26** (P31 L1-4: I think this conclusion should be formulated less strong: the 'erroneous conclusions' depend on what you are modelling. Your results appear to show that using SS or NSS conditions does not have a large effect on e.g. the total amount of Rsoil over a whole growing season. However, if someone want to obtain detailed daily estimates of Rsoil on a (sub-) daily timescale, this is indeed important. I suggest the authors re-formulate these sentences.

Thank-you for this comment. We've added extra text to this last paragraph of section 4.2, so hopefully the statement is more explicit.

**Technical comments**

Thank-you for spotting all of these. We have fixed all of these issues that you mentioned and highlighted the relevant text in yellow.

P2 L34: ... down to 1 m

- P3 L51: ... precipitation inputs. The DETECT model...
- P5 L8: ... coarse-grained
- P5 L9: fast CO2 diffusion rates
- P5 L11: ... we expect coarse-grained soils
- P5 L13: ... air-filled pore space
- P6 L14: ... depth-invariant CO2 production rates
- P7 L 16: behavior and to (no comma)
- P11 L12: remove the comma before 'and'

P18 L5: ... to 1 m depth

P20 L10: change to '(J previous time periods)'

P21 L20: if the SOC data you talk about is the same as shown in figure S4, you can refer to that figure here.

P23 L18:... 2013). These data were...

P30 L17: You could change this to: ... it may take about 15 minutes for a...

**Figures and tables**

**Figure 1 Caption**: everything after '... ,and temporally varying bulk CO2 fluxes.' is redundant here. You could alternatively refer to the material and methods section where this is also explained.

We've deleted this text in the figure 1 caption.

**Figure 2** - Legend: add that root and microbial contributions are simulated using the DETECT model -For easier comparison of the Rsoil between the two scenarios, you could indicate the Rsoil values shown in (a) on the bars in (b) - Caption: 'see Table 2' should be Table 3 (also in Fig. 3, 4, 6, S1 and S2)

We've modified the text in the caption to address your first point, so hopefully the caption is clearer now. We didn't understand what you meant by your second comment and so have ignored this.

**Figure 3** - Names of the scenarios in the sub-figures could be replaced with more intuitive names, followed by the scenario name between brackets, to increase readability. - Include a legend for the grey and red lines.

We chose the names of the scenarios after a lot of consideration. We wanted something that was short but easy to refer to (at least after consultation with table 3). We have, however, included a fuller version of the scenario names in the titles of each subplot.

**Figure 5** - Subplots (a) and (b): as you want to make the comparison between measurements and model results, you could choose only to show the timespan for which measurements are available (and show the entire timespan in the supplement) - Legend: add 'depth': e.g. 3 cm depth

We understand your point here, however we think that it's better to include the whole time span here for consistency with the other figures (might confuse reader if using different time windows for different figures). We made the 'depth' change as you suggested.

**Table 1** -Instead of grouping the variables by 'Group1', 'Group2', etc, it would be more intuitive to provide the names to which the groups refer in the table (e.g. Group 1 = microbial submodel parameters, etc.) - I would encourage the authors to include the references from where the parameter values were obtained in the table (where appropriate), now this is only described in the text.

We have added names following group 1, etc.. so that it's clearer for the reader. We understand that including references for each parameter would be useful here, but we are limited by space in table 1. We feel that referring to the equation where each parameter belongs is sufficient, since the reader can refer to that equation to seek the proper reference. Another reason for not including a reference for each parameter here is that some parameters are based on the references given in the text following calculations that we have described in the manuscript. Hence, it is far better for the reader to go directly to the equation where the text that follows that equation and also to section 2.4.5 (now included in the caption) to understand what each parameter is and how it was estimated.

Table 3 - Bottom row, middle column: 'about' should be 'above'?

Changed.

**Supplementary information**

**Appendix S1** - Is there any evidence that root biomass varies between 0.5 and 1.5 times the amount measured in the middle of the growing season? Please include this. - Last sentence of first paragraph: 'decays' should be 'declines'?

We've updated the text in appendix S1 to address this. 'Decays' has now changed to 'declines'.

Figure S2 - Same remarks as for Fig. 3

Titles for each subplot have been updated.

[revised manuscript text omitted]

**Commented [RE5]:** REVIEWER 2 specific comment #3. We have shortened this paragraph as far as possible by removing any detail that is already given in the methods section.

**Commented [RE6]:** REVIEWER 1 General comment #1. 'and CO2 production rates' has been deleted.

**1 2. Methods**

**2 2.1 Description of the Non Steady State DETECT Model**

The PDE that underlies the DETECT model (v1.0) accounts for time- and depth-varying CO2 diffusivity and CO2 production by root and microbial respiration (Fang & Moncrieff, 1999). We use a pair of PDEs, one describing the soil CO2 derived from root respiration (subscripted with *R*), and the other for CO2 derived from microbial respiration (*M*) such that for K = R or *M*:  $\partial C_{K}(z,t) = \partial (z, z, t)$

$$\frac{\partial c_K(z,t)}{\partial t} = \frac{\partial}{\partial z} \left( D_{gs}(z,t) \frac{\partial c_K(z,t)}{\partial z} \right) + S_K(z,t) \tag{1}$$

 $c_K(z,t)$  is CO2 concentration (mg CO2 m-3),  $D_{gs}(z,t)$  is the effective diffusivity of CO2 through the 7 soil (m2 s-1), and  $S_{k}(z,t)$  is the source (or production) term (mg CO2 m-3) (Fig. 1b), all of which 8 vary by depth z (meters) and time t (hours). Note that  $D_{gs}$  is assumed to be the same for root- and 9 microbial-derived  $CO_2$  and is thus not indexed by K. In this version of the model, we assumed 10 11 that CO2 transport within the soil profile and over time is solely governed by gaseous diffusion, and we ignored other types of  $CO_2$  transport—such as diffusion in the liquid state, convection, 12 13 and bulk transport via vertical movement of water-that have been shown to have a negligible contribution (Fang and Moncrieff, 1999; Kayler et al., 2010). Total soil CO2 and total CO2 14 production are given as  $c(z,t) = c_M(z,t) + c_R(z,t)$  and  $S(z,t) = S_M(z,t) + S_R(z,t)$ , respectively. Below 15 we describe the two main components of the PDE model: (1)  $CO_2$  diffusivity,  $D_{gs}$ , and (2) the 16 17 production terms,  $S_R(z,t)$  and  $S_M(z,t)$ . Finally, we note that equation 1 is the mass balance equation (see appendix S3 in the supplementary information for more information). Commented [E8]: REVIEWER 1 General comments #2. New text. 18

19 2.1.1 Soil CO2 diffusivity sub-model

20 The diffusivity of  $CO_2$  within the soil ( $D_{gs}$ ) depends on soil structure and water content; we

**Commented [E7]:** REVIEWER 2, specific comments #4 and #5. As suggested by these two comments, the first two paragraphs of the methods from the previous version of the manuscript have been deleted. modeled Dgs using the Moldrup function (Sala et al., 1992; Moldrup et al., 2004). We chose this
formulation because it is more accurate than other common models, such as the Millington and
Quirk (2000) and Penman (1981) models (Moldrup et al., 2004). Based on Moldrup et al. (2004),
Dgs (m2 s-1) is defined as:

5
$$D_{g_{s}}(z,t) = D_{g_{0}}(z,t) \cdot \left(2\phi_{g_{100}}(z)^{3} + 0.04\phi_{g_{100}}(z)\right) \cdot \left(\frac{\phi_{g}(z,t)}{\phi_{g_{100}}(z)}\right)^{2+\frac{3}{b(z)}},$$
 (2)

6 where
$$D_{g0}(z,t) = D_{stp} \cdot \left(\frac{T_s(z,t)}{T_0}\right)^{1.75} \cdot \left(\frac{P_0}{P(t)}\right)$$
 and  $D_{stp} = 1.39 \times 10^{-5} \text{ m}^2 \text{ s}^{-1}$  is the diffusion

coefficient for  $CO_2$  in air at standard temperature ( $T_0$ , 273 K) and pressure ( $P_0$ , 101.325 kPa); 7 8  $T_s(z,t)$  is the soil temperature (Kelvin) at depth z and time t, and P(t) is the air pressure (kPa) just 9 above the soil surface at time t. The remaining terms in Eqn 2 include  $\phi_g(z,t)$ , the air-filled soil porosity, which is related to the total soil porosity ( $\phi_T$ ) and volumetric soil water content ( $\theta$ ) 10 according to  $\phi_g(z,t) = \phi_T(z) - \theta(z,t)$ , and  $\phi_T(z)$  is defined as 1 - BD(z)/PD, where BD and PD are 11 the bulk density and particle density of the soil, respectively (Davidson et al., 2006);  $\phi_{g100}(z)$  is 12 the air-filled porosity at a soil water potential ( $\Psi$ ) of -100 cm H2O (about -10 kPa); b(z) is a 13 14 unitless parameter that is related to the pore size distribution of the soil based on the water retention curve given by  $\Psi = \Psi_e(\theta/\theta_{sat})^{-b}$ , where  $\Psi_e(z)$  is the air-entry potential – calculated from 15 measurements (Morgan et al., 2011) – and  $\theta_{sat}(z)$  is the saturated soil water content (v/v). 16

**17 2.1.2 CO2 source (production) terms**

- 18 Soil CO2 can be produced in the soil (*S* term in Eqn. 1) by five different biological processes: (i)
- 19 root growth respiration, (ii) root maintenance respiration, (iii) consumption of rhizodeposits by
- 20 root-associated microorganisms and associated microbial respiration, (iv) microbial

**Commented [E9]:** REVIEWER 1 specific comment #5. New reference to address this comment.

**Commented [E10]:** REVIEWER 1 specific comment #5. New text and reference to address this comment.

decomposition of newly produced plant litter that has been incorporated into the soil matrix, and 1 2 (v) microbial decomposition of older soil organic matter (SOM) (Pendall et al., 2004). Due to the general lack of sufficient data and process understanding to accurately separate all five sources, 3 the DETECT model treats CO2 production as the sum of two main contributions: CO2 respired 4 by (1) roots and closely associated microorganisms (the sum of (i)-(iii)), giving  $S_R(z,t)$ , and (2) 5 free-living soil microorganisms (the sum of (iv)-(v)), giving  $S_M(z,t)$ . Such simplification based on 6 root and microbial sources is common in models of soil CO2 transport and production (Šimůnek 7 and Suarez, 1993; Fang and Moncrieff, 1999; Hui and Luo, 2004). Although DETECT v1.0 8 assumes that root and microbial respiration are independent of one another, they both depend on 9 the same environmental data (e.g.,  $\theta$  and  $T_s$ ). 10

11 CO2 production by root respiration is represented as the product of three terms: (i) the 12 mass-specific base respiration rate ( $R_{Rbase}$ ) at a reference soil temperature of  $T_s = T_{ref}$  and at 13 average soil water and antecedent temperature conditions, (ii) root mass expressed as the amount 14 of root carbon,  $C_R(z,t)$ , and (iii) functions that rescale  $R_{Rbase}$  to account for the effect of soil water 15 ( $\theta$ ), temperature ( $T_s$ ), and their antecedent counterparts, which are determined separately for 16 roots and microbes. For roots, antecedent soil water and temperature are denoted as  $\theta_R^{ant}$  and 17  $T_s^{ant}$ , respectively. In general,  $S_R(z,t)$  is given by:

$$S_R(z,t) = R_{Rbase} \cdot C_R(z,t) \cdot f(\theta(z,t), \theta_R^{ant}(z,t)) \cdot g(T_S(z,t), T_S^{ant}(z,t))$$
(3)

The functional form of  $C_R(z,t)$  is informed by field data on root biomass C (see Appendix S1 for complete details). The functions f and g are given by:

20
$$f\left(\theta,\theta_{R}^{ant}\right) = \exp\left(\alpha_{1}\theta(z,t) + \alpha_{2}\theta_{R}^{ant}(z,t) + \alpha_{3}\theta(z,t) \cdot \theta_{R}^{ant}(z,t)\right)$$
(4a)

21
$$g(T_S, T_R^{ant}) = \exp\left(E_o(z, t) \cdot \left(\frac{1}{T_{ref} - T_o} - \frac{1}{T_S(z, t) - T_o}\right)\right)$$
(4b)

$$E_o(z,t) = E_o^* + \alpha_4 T_S^{ant}(z,t)$$
(4c)

| 1  | $R_{Rbase}$ , $\alpha_1$ , $\alpha_2$ , $\alpha_3$ , $\alpha_4$ , $T_o$ , and $E_o^*$ are parameters that require numerical values (Table 1; Ryan |
|----|---------------------------------------------------------------------------------------------------------------------------------------------------|
| 2  | et al. 2015), $\theta$ and $T_s$ are informed by field data, and $\theta_R^{ant}$ and $T_s^{ant}$ are computed from the field                     |
| 3  | data (described below). The temperature scaling function, g (Eqn 4b) was motivated by Lloyd                                                       |
| 4  | and Taylor (1994) has been successfully used to describe soil and ecosystem respiration (Luo                                                      |
| 5  | and Zhou, 2010; Cable et al., 2013; Ryan et al., 2015). $E_o(z,t)$ is analogous to an energy of                                                   |
| 6  | activation term that governs the apparent temperature sensitivity of $S_R$ (Davidson and Janssens,                                                |
| 7  | 2006; Cable et al., 2011; Tucker et al., 2013); we assume $E_o$ responds to antecedent temperature,                                               |
| 8  | reflecting a potential thermal acclimation response (Atkin and Tjoelker, 2003; Ryan et al., 2015).                                                |
| 9  | $T_o$ is also related to the apparent temperature sensitivity (Cable et al., 2011), and we assume that                                            |
| 10 | it is invariant with depth and time (Lloyd and Taylor, 1994; Cable et al., 2013; Barron-Gafford et                                                |
| 11 | al., 2014; Ryan et al., 2015). While the functional forms and choice of environmental drivers                                                     |
| 12 | used for $f$ and $g$ were motivated by previous analyses (Cable et al., 2013; Barron-Gafford et al.,                                              |
| 13 | 2014), the exact functions and parameter values were based on Ryan et al. (2015) and Cable et                                                     |
| 14 | al. (2013). Exponential functions are also used for the moisture $(f_3)$ and temperature $(g)$ scale                                              |
| 15 | functions to ensure $f > 0$ and $g > 0$ (Eqn 4a). The choice of an exponential form of the functions                                              |
| 16 | was based on Ryan et al. (2015), with graphical forms of the functions given in figure S10                                                        |
| 17 | (supplementary information). However, the DETECT model is flexible enough to accommodate                                                          |
| 18 | alternative functions for $f$ and $g$ . For example, we ran DETECT for the control scenario using a                                               |
| 19 | bell-shaped function that described how soil CO 2 production changes with $\theta$ (appendix S4 and                                    |
| 20 | Figure S8, supplementary information) as an alternative to equation 4a. For this alternative                                                      |
| 21 | model run, the modelled $R_{soil}$ was very similar to the modelled $R_{soil}$ from the results of this study                                     |
|    | (Figure SQ, supplementary information)                                                                                                            |

**Commented [E11]:** REVIEWER 2 Specific comment #6. This sentence is unchanged from the previous version but has been moved to appear slightly earlier in this paragraph.

**Commented [RE12]:** REVIEWER 2 General comments #2. New text to address this comment.

CO2 production by microbial respiration and SOM decomposition is represented by a
modified version of the Dual Arrhenius and Michaelis-Menten (DAMM) model (Davidson et al.,
2012). We exclude the O2 term, rendering the model relevant to systems that are typically
unlimited by O2 availability, such as the semi-arid site that we focus on, but we accounted for a
microbial C pool (*CMIC*) and a soluble soil-C pool (*CSOL*) (Todd-Brown et al., 2012) such that:

$$S_M(z,t) = V_{max}(z,t) \cdot \frac{C_{SOL}(z,t)}{K_m + C_{SOL}(z,t)} \cdot C_{MIC}(z,t) \cdot (1 - CUE)$$
(5)

7 Decomposition is assumed to be an enzymatic process that follows Michaelis-Menten kinetics, 8 where  $V_{max}$  is the maximum potential decomposition rate, and  $K_m$  (the half-saturation constant) is 9 the amount of substrate required for the decomposition rate to reach half of  $V_{max}$ . Carbon-use 10 efficiency (CUE) represents the proportion of total C assimilated by microbes that is allocated 11 for microbial growth (Tucker et al., 2013). We excluded a microbial death rate term (Todd-12 Brown *et al.*, 2012) because we had insufficient data on death rates, and  $C_{MIC}$  is only ~1% of 13  $C_{SOL}$  at our study site (Carrillo and Pendall, in review).

In contrast to the original DAMM formulation, we allowed  $S_M(z,t)$  and  $V_{max}(z,t)$  to vary by depth and time, whereas existing applications of the DAMM model are generally applied to "bulk" soil (i.e., do not vary with z). We also modeled  $V_{max}$  according to the modified energy of activation function described in Lloyd and Taylor (1994), which essentially parallels Eqns 4b-4c:

18
$$V_{max}(z,t) = V_{Base} \cdot f\left(\theta, \theta_M^{ant}\right) \cdot \exp\left(E_o(z,t) \cdot \left(\frac{1}{T_{ref} - T_o} - \frac{1}{T_S(z,t) - T_o}\right)\right)$$
(6)

19  $V_{Base}$  is the 'base'  $V_{max}$  at a reference soil temperature of  $T_{ref}$  and at mean values of current  $\theta$  and 20 antecedent  $\theta$  and  $T_S$  (i.e., mean values of  $\theta_M^{ant}$  and  $T_s^{(ant)}$ ).  $E_o(z,t)$  and  $f(\theta, \theta_M^{ant})$  follow the same 21 functional forms and interpretation as described for the root respiration submodel (Eqns 3 and 4a-c), except that θantM and TantM are used instead of θantR and TantR, respectively, and different
 values are specified for the parameters α1, α2, α3, α4, To, and Eo\* to reflect microbial respiration
 The values are given in table 1, and section 2.4.5 explains how the values were estimated.
 Finally, CSOL is modeled as a function of soil organic C content at depth z, CSOM(z), based
 on the fraction, p, of CSOM(z) that is soluble and the diffusivity of the substrate in liquid, Dliq
 (Davidson et al., 2012). The equation for CSOL is given by:

$$C_{SOL}(z,t) = C_{SOM}(z) \cdot p \cdot \theta(z,t)^3 \cdot D_{liq}$$
(7)

7 The values of p and  $D_{liq}$  were taken from laboratory analysis (see § 2.4.5) and Davidson et al. (2012), respectively. We assumed that  $C_{SOM}(z)$  and  $C_{MIC}(z)$  (see Eqn 5) are constant over time 8 given the relatively short simulation periods we explored here (a single growing season); but the 9 model could be easily modified to allow for time-varying  $C_{SOM}$  and  $C_{MIC}$ . Here,  $C_{SOM}(z)$  and 10  $C_{MIC}(z)$  are simple, empirical functions that were informed by data (see Appendix S1 for details). 11 Moreover, while assumption of time invariant  $C_{SOM}(z)$  and  $C_{MIC}(z)$  is an implicit SS assumption 12 13 about biological factors affecting soil CO2 dynamics, this assumption allows us to isolate the importance of NSS conditions that are primarily due to physical CO2 transport characteristics. 14

**15 2.1.3 Soil respiration**

16 The efflux of  $CO_2$  from the soil surface (soil respiration,  $R_{soil}$ ) is computed as:

17
$$R_{soil}(t) = \frac{D_{gs}(z=0.01,t)}{\Delta z} \left( c(z=0.01,t) - c_{atm}(t) \right)$$
(8)

18  $D_{gs}(z=0.01, t)$  is the diffusivity of CO2 in the soil and c(z=0.01, t) is the total CO2 concentration 19 (microbial- and root-derived), respectively, at z = 0.01 m depth and time t;  $c_{atm}(t)$  is the CO2 20 concentration in the atmosphere above the soil surface; and  $\Delta z$  is the depth increment that the 21 model solves for soil CO2 concentration (here,  $\Delta z = 0.01$  m). **Commented [E13]:** REVIEWER 2, specific comment #8. New text to address the comment.

Commented [E14]: REVIEWER 1 specific comment #8. Fixed typo.

**1 2.2 Numerical implementation of the DETECT model**

| 2  | The numerical solution to the NSS version of the DETECT model v1.0, as described in Eqns 1-8,                      |                                                          |                                   |   |
|----|--------------------------------------------------------------------------------------------------------------------|----------------------------------------------------------|-----------------------------------|---|
| 3  | requires an initial condition (IC) and two boundary conditions (BCs), which we specified as:                       |                                                          |                                   |   |
|    | IC: $c(z, t = 0) = c_0(z)$ (9a)                                                                                    |                                                          |                                   |   |
|    | Upper BC:                                                                                                          | $c(z=0,t)=c_{atm}(t)$                                    | (9b)                              |   |
|    | Lower BC:                                                                                                          | $\frac{\partial c(z=1,t)}{\partial z} = 0$               | (9c)                              |   |
| 4  | The function $c_0(z)$ is determined a                                                                              | nd parameterized in two stages: (1)                      | observed soil CO 2     |   |
| 5  | concentration data at three depths                                                                                 | from the start of the 2007 growing s                     | season were used to               |   |
| 6  | parametrize a simple function that                                                                                 | t described the change in CO 2 concer         | ntration for all depths; (2)      |   |
| 7  | the DETECT model was run forw                                                                                      | ard for the growing season of 2007,                      | then the modelled CO 2 |   |
| 8  | concentrations for all depths on the final day of the 2007 growing season (September 31, 2007)                     |                                                          |                                   |   |
| 9  | was used as the initial condition for running the DETECT model for 2008. See Appendix S2 in                        |                                                          |                                   |   |
| 10 | the supplementary information for specific details. We set $c_{atm}(t)$ equivalent to 356 ppm for all $t_{atm}(t)$ |                                                          |                                   | ( |
| 11 | which was the average near-surface, ambient atmospheric CO 2 concentration measured at the              |                                                          |                                   | c |
| 12 | PHACE site in the 2008 growing season. Following methods of Haberman (1998), we adopted a                          |                                                          |                                   |   |
| 13 | zero-flux lower BC (Eqn 9c) due t                                                                                  | to the lack of data at or near a depth                   | of 1 m.                           |   |
| 14 | We numerically solved the                                                                                          | e non-linear PDE (Eqn. 1) by employ                      | ying a forward Euler              |   |
| 15 | discretization with a centered diffe                                                                               | erence method for the depth derivati                     | ve at a depth increment of        |   |
| 16 | $\Delta z = 0.01$ m. To ensure numerical stability, we calculate model outputs at a numerical time-step            |                                                          |                                   |   |
| 17 | of $\Delta t = \frac{dt}{Ndt}$ , where $dt$ is the time                                                            | step at which the predicted outputs a                    | are stored (6 hours), and         |   |
| 18 | Ndt is the number of numerical tir                                                                          | me-steps. Ndt is computed based on                       | the fastest (largest)             |   |
| 19 | diffusion coefficient at each time                                                                                 | step such that $\frac{Ndt}{0.5 \times (\Delta z)^2}$ , y | where $\max(D_{gs})$ is the       | ( |

**Commented [E15]:** REVIEWER 1 General comments #3: we have modified this text to be clearer about how we estimated the initial conditions for the DETECT and DETECT-SS models.

**Commented [E16]:** REVIEWER 1 specific comment #9. Now an equation (was an inequality).

2 separately for both root- and microbial-derived CO2 concentrations, such that for K = R or M:  $\frac{c_K(z,t+\Delta t) - c_K(z,t)}{\Delta t} = D_{gs}(z,t) \left( \frac{c_K(z+\Delta z,t) - 2c_K(z,t) + c_K(z-\Delta z,t)}{(\Delta t)^2} \right)$ 3  $+\left(\frac{D_{gs}(z+\Delta z,t)-D_{gs}(z-\Delta z,t)}{2\Delta z}\right)\left(\frac{c_{K}(z-\Delta z,t)-c_{K}(z+\Delta z,t)}{2\Delta z}\right)$ 4  $+S_K(z,t)$ (10)5 We rearranged Eqn. 10 to solve for  $c_K(z,t+\Delta t)$ , which was iterated forward for all time-steps and 6 depth increments; total CO2 concentration at each time step and depth is calculated as  $c(z,t+\Delta t) =$ 7  $c_R(z,t+\Delta t) + c_M(z,t+\Delta t)$ . For clarity, we emphasize that equation 10 is the discretized version of 8 equation 1, which we require in order to numerically solve equation 1 (Haberman, 1998). We 9 10 programmed the DETECT model v.10 and the numerical solution method in Matlab 11 (Mathworks, 2016). 2.3 Steady-state (SS) solution to the DETECT model 12 A primary goal of this work was to test if soil CO2 and associated Rsoil predicted from the non-13 steady-state (NSS) model (DETECT) could be distinguished from that of the steady-state (SS) 14 solution. The SS version of Eqn 1, which we refer to as the SS-DETECT model, can be solved 15 analytically as an ordinary differential equation (ODE) by setting the  $\partial c/\partial z$  term to zero 16 (Amundson et al., 1998). As with the NSS model, we found the SS solution to Eqn. 1 separately 17 for root- and microbial-derived CO2 concentrations,  $c_R^*(z,t)$  and  $c_M^*(z,t)$ , respectively. Using the 18

maximum  $D_{gs}$  across all depth increments at time t (Haberman, 1998). We solved Eqn. 1

**19 upper and lower boundary conditions described for the NSS model (Eqns 9b and 9c), the**

- analytical SS solutions at time t and depth z are derived by Amundson et al. (1998) and Cerling
- 21 (1984). The solution is given by:

1

**Commented [E17]:** REVIEWER 1 specific comment #10. New text to address this comment.

**Commented [E18]:** REVIEWER 1 specific comment #11. New citation added to address comment.

$$c_{K}^{*}(z,t) = \frac{S_{K}^{*}(t)}{D_{gs}(z,t)} \left(z - \frac{z^{2}}{2}\right) + c_{atm}(t)$$
(11a)

$$S_K^*(t) = \frac{1}{100} \sum_{z=0.01}^{1m} S_K(z,t)$$
(11b)

3 where K=R and K=M refers to the soil CO2 from root (R) and microbial (M) sources,

4 respectively.  $S_K^*(t)$  is the depth-averaged source term for microbial or root production

5 (averaging over 100 0.01-m increments). The soil CO2 diffusivity term,  $D_{gs}(z,t)$ , and upper

6 boundary condition,  $c_{atm}(t)$ , are the same as previously defined (Eqns 2 and 9b, respectively;

7 Amundson *et al.* (1998)).

1

2

**8 2.4 Application of the DETECT and SS-DETECT models to the PHACE site**

9 In this subsection, we provide an overview of the study site, including the PHACE experiment,

10 and relevant data sources from PHACE that we used to drive the DETECT and SS-DETECT

11 models. We also summarize how we calibrated the models in the context of the PHACE site, and

12 we highlight data that we used to informally validate the general behavior of the models. We

13 conclude by describing the simulation experiments that we conducted to test the effects of soil

14 texture and precipitation variability on the importance of NSS versus SS soil CO2 conditions.

**15 2.4.1 Field site and PHACE experiment**

16 The Prairie Heating and CO2 Enrichment (PHACE) field experiment is located in south-central

17 Wyoming (latitude  $41^{\circ}$  50'N, longitude  $104^{\circ}$  42'W, elevation = 1930 m). The site is a mixed-

[revised manuscript text omitted]

11
$$\theta_M^{ant}(z,t) = \sum_{j=1}^J w(j) \cdot \theta(z,t-j)$$
(12)

12 The w's are the antecedent importance weights, which sum to 1 from j = 1 (previous time period) to j=J(J previous time periods). The weights were informed by results from an analysis of 13 ecosystem respiration at the PHACE site (Ryan et al., 2015). For microbes, J = 4 days and w =14 (0.75, 0.25, 0, 0), indicating the strong importance of  $\theta$  conditions occurring yesterday (i = 1) 15 (Oikawa et al., 2014). Similar equations were used to compute  $\theta_R^{ant}(z,t)$  and  $T_s^{ant}(z,t)$ , each with 16 their own set of weights (w's) and time-scales (J's). For example, the time step and J for  $\theta$  differ 17 among microbes (2 days) and roots (3 weeks); for roots,  $\theta_R^{ant}(z,t)$  was computed as a weighted 18 average of past, average weekly values of  $\theta$ , with *j* denoting weeks into the past, for J = 4 weeks, 19 20 and w = (0.2, 0.6, 0.2, 0), indicating a strong lag response to  $\theta$  conditions occurring two weeks ago (Cable et al., 2013; Ryan et al., 2015). For antecedent soil temperature, we assumed that 21 22 each of the past four days were equally important by setting the w vectors to (0.25, 0.25, 0.25,

[revised manuscript text omitted]

VIEWER 2, specific comment #12. New nt.

REVIEWER 2, specific comment 14. About ere has been deleted as it was already

| 1  | two quantities which were based on unfumigated extracts obtained for microbial biomass                                |
|----|-----------------------------------------------------------------------------------------------------------------------|
| 2  | estimations as above ( $C_{SOL}$ ) and on total C concentration in soil ( $C_{SOM}$ ).                                |
| 3  | The values used for the base microbial respiration rates and the half-saturation constant                             |
| 4  | $(V_{Base} [Eqn 6] and K_m [Eqn 5]; Table 1)$ were estimated by fitting the microbial respiration                     |
| 5  | submodel, but without the $C_{MIC}$ or $CUE$ terms (Eqn 5), to microbial respiration data from the                    |
| 6  | PHACE control plots (Fig. S7, supplementary information). The C MIC and CUE terms were not                 |
| 7  | included in this earlier version of S M submodel – which was used for model calibration purposes           |
| 8  | - because we did not have measurements of these two variables at the time. We estimated $V_{Base}$                    |
| 9  | and $K_m$ using a Markov Chain Monte Carlo approach, identical to the approach used in Ryan et                        |
| 10 | al. (2015). In the absence of root respiration data, we assumed that base root respiration ( $R_{Rbase}$              |
| 11 | [Eqn 3]; Table 1) was proportional to the microbial base rate term (Hanson et al., 2000). The                         |
| 12 | parameters denoting the effects of current soil moisture (e.g., $\alpha_1$ ; Eqn 4a), antecedent moisture             |
| 13 | $(\alpha_2)$ , and the interaction between current and antecedent moisture $(\alpha_3)$ on root and microbial         |
| 14 | respiration were derived from Ryan et al. (2015), also based on an analysis of ecosystem                              |
| 15 | respiration $(R_{eco})$ data from PHACE. However, we adjusted the values (Table 1) by trial and                       |
| 16 | error to reflect the expectation that the effects of current soil moisture should be stronger for                     |
| 17 | microbial compared to root respiration because microbes tend to respond more rapidly to                               |
| 18 | precipitation pulses (Risk et al., 2008), whereas root respiration is likely to show a delayed                        |
| 19 | response that depends more strongly on past moisture conditions (Cable et al., 2008; Cable et al.,                    |
| 20 | 2013). Of the remaining two parameters describing $S_M$ (Eqns 5-6; Table 1), the value of $CUE$                       |
| 21 | was based on results from a soil incubation study conducted at a nearby site (Tucker et al., 2013),                   |
| 22 | whilst our value for $D_{liq}$ was taken from Davidson et al. (2012). Three parameters ( $E_o^*$ , $T_o$ , and |
| 23 | $\alpha_4$ ; Eqns 4a-b) were shared between the $S_R$ and $S_M$ submodels, and their values were also                 |

**Commented [RE24]:** REVIEWER 2, specific comment 15. It used to say '(Eqn 7)' at the end of the sentence (where this new text is now), but the reviewer quite rightly pointed out that this was confusing because we used C\_SOL and C\_SOM to estimate parameter p (not equation 7). Equation 7 is used to estimate C\_SOL for all depths and times based on simulated values of C\_SOM which are informed from measurements.

Commented [RE25]: REVIEWER 1 specific comment #15 / REVIEWER 2 specific comment #16. New text to address this comment.

**Commented [E26]:** REVIEWER 1 specific comment #16. New text to address first part of comment.

**Commented [RE27]:** REVIEWER 2, specific comment #17. New text to address comment

obtained from Ryan et al. (2015). Finally, the parameters used for CO2 diffusivity (b, BD, and 1  $\phi_{e100}$ ; Eqn 2) were based on published, site-specific data (Morgan et al., 2011). 2 2.4.6 Informal model validation with soil respiration measurements 3 We evaluated the accuracy of the DETECT model by comparing (1) predicted Rsoil (Eqn 8) 4 5 against plot-level measurements of ecosystem respiration ( $R_{eco}$ ) (see below) and (2) predicted soil CO2 concentrations, c(z,t), versus observed concentrations; all observed data were from the 6 PHACE study. Since we did not rigorously parameterize the DETECT model with PHACE data, 7 8 we were simply looking for reasonable, qualitative agreement between the modelled variables and the observations (e.g., similar order of magnitude, comparable temporal trends). Observed 9 10  $R_{eco}$  was measured on control plots every 2-4 weeks during the target growing season, using a canopy gas exchange chamber, and instantaneous fluxes were scaled to daily rates using a linear, 11 empirical function (Jasoni et al., 2005; Bachman et al., 2010). We assumed that Rsoil was similar 12 to  $R_{eco}$  given that above ground biomass was <20% of total plant biomass (Mueller et al., 2016). 13 14 Measurements of microbial respiration were obtained by applying glyphosate herbicide to small subplots in May, 2008, limiting ecosystem CO2 efflux to microbial sources (Pendall et al., 2013), 15 Non-steady state soil chambers were used to estimate the resulting surface soil fluxes every two 16 17 weeks around midday (Oleson et al., 2013; Ogle et al., 2016). Soil CO2 concentrations were also 18 measured with non-dispersive infrared sensors (Vaisala GM222, Finland) installed at 3, 10, and 20 cm below the soil surface, averaged on an hourly basis (Risk et al., 2008; Vargas et al., 2011; 19 20 Brennan, 2013). Observations of soil [CO2] for control plots were compared against predictions of c(z,t) at z = 0.03, 0.1, and 0.2 m and at the corresponding times. 21

22 2.5 Simulation Experiments

[revised manuscript text omitted]

**Commented [E32]:** REVIEWER 2, specific comment #23. Text changed here. Specifically I removed reference to fig. 3 which doesn't show predicted root respiration lines (like fig. 2 does).

Commented [E33]: REVIEWER 2, specific comment #24. Text changed here to address this comment.

1 of the Ctrl scenario, which led to a reduction in total growing season Rsoil in the P-FM scenario

2 (Fig. S2c and S2e).

**3 3.4 Effects of antecedent responses**

When antecedent soil water content and soil temperature were included in the DETECT model 4 we found that predicted Rsoil was 15% greater for the control scenario and 29-37% greater for the 5 fine textured soil scenarios, compared to the corresponding scenarios that did not include 6 7 antecedent conditions. When the sand content was 80% or for any of the different precipitation regimes, there was a negligible difference between Rsoil predicted by the antecedent versus non-8 9 antecedent parametrizations of DETECT. Daily Rsoil predicted by the DETECT model based on the Ctrl and Ctrl-ant scenarios 10 11 agreed well with observed ecosystem respiration ( $R_{eco}$ ), but  $R_{eco}$  was slightly higher than predicted  $R_{soil}$  (Fig. 2a,b), which was expected since  $R_{eco} = R_{soil}$  + aboveground autotrophic 12 respiration. For the most part, this data-model agreement was similar whether the antecedent 13 model terms were included (Fig. 2b) or not (Fig. 2a). Unfortunately,  $R_{eco}$  data were not available 14 during the time period (days 230-250) associated with the greatest disagreement between the Ctrl 15 and Ctrl-ant scenarios. During this period, frequent hourly measurements of soil [CO2] were in 16 better agreement with predicted soil CO2 from the Ctrl-ant scenario compared to the Ctrl 17 scenario (Figs. 5a,b, S4a,b). After day ~250, based on the DETECT model, both scenarios (Ctrl 18 and Ctrl-ant) under-predicted the observed soil [CO2] by ~ 50% (Fig. 5). 19

**20 4. Discussion**

- The DETECT and SS-DETECT models provide a framework for evaluating the circumstances
  under which steady-state (SS) assumptions of soil CO2 production and surface soil respiration
- 23  $(R_{soil})$  are valid, and to identify the major physical (i.e., soil texture, soil moisture) and/or

1 biological (i.e., root and microbial respiration responses) factors that lead to non-steady-state

2 (NSS) conditions.

**3 4.1 Steady-state versus non-steady-state conditions**

4 At the seasonal scale, there was reasonable agreement between total growing season  $R_{soil}$

5 predicted under the assumption of SS versus NSS conditions, but the strength of this agreement

6 depended on soil texture (see  $\S4.2$ ). At the daily scale,  $R_{soil}$  predicted by the DETECT model

7 deviated from values expected under the assumption of SS conditions for 11 days or 4% of the

8 days during the April-September growing season (Fig 2, days 218-228). These discrepancies,

9 attributed to NSS conditions, were generally limited to periods following large rain events. For

10 applications that assume SS conditions, such as isotopic partitioning studies (Hui and Luo, 2004;

11 Ogle and Pendall, 2015), the SS assumption seemed reasonable during periods of minimal or no

12 precipitation, representative of times during which soil water content changes very little or

13 gradually. For sites or time periods characterized by pulsed precipitation patterns, our results

14 suggested that NSS conditions would be more likely over longer periods of time.

**15 **4.2 Effect of varying soil texture**

Our results indicated that soil texture exerts the strongest control over the prevalence of NSS soil CO2 conditions. For a predominantly (e.g., 60%) sandy or silty soil, soil CO2 transport and efflux generally aligned with the SS assumption (Fig. 2, Fig. 3a-b). This was consistent with previous work that used SS models to predict  $R_{soil}$  for similar soil types (Baldocchi et al., 2006; Vargas et al., 2010).

For very fine-texture soil dominated by clay, however, SS assumptions were far less appropriate. The larger difference – relative to the *Ctrl* scenario – in  $R_{soil}$  predicted under SS versus NSS conditions for fine-texture (i.e., 60% clay) soil was apparent at both the growing

**Commented [E34]:** REVIEWER 2, specific comment #25. New text to address comment.

| 1  | season scale and the daily scale following a large precipitation event (Fig. 3, S3a, S3b). In               | Commente           |
|----|-------------------------------------------------------------------------------------------------------------|---------------------------|
| 2  | general, the DETECT model predicted that $R_{soil}$ should be higher in clay compared to sandy soil         | (                         |
| 3  | after precipitation events, a result supported by field experiments (Cable et al., 2008), but this          |                           |
| 4  | texture effect is muted under assumptions of SS. Moreover, recovery of $R_{soil}$ to SS rates after a       |                           |
| 5  | large rain event took $\sim$ 30 days in the clay soil (Fig. 3c, days 218 to 248) compared to $\sim$ 10 days |                           |
| 6  | for the other coarser soil texture scenarios (Fig. 2, Fig. 3a-b, days 218 to ~230). These effects of        |                           |
| 7  | soil texture on the prevalence of NSS conditions can be attributed to soil physical properties and          |                           |
| 8  | their effects on air-filled porosity and CO2 diffusivity. Fine textured soils have smaller pores and        |                           |
| 9  | tend to retain water for longer (Bouma and Bryla, 2000), which has the effect of decreasing soil            |                           |
| 10 | CO2 diffusivity (Fig. 6). Thus, under moist conditions that follow a rain event, it may take about          |                           |
| 11 | 15 minutes for a $CO_2$ molecule produced at 0.5 m to diffuse to the surface in a clay soil                 |                           |
| 12 | compared to only 1-2 minutes for a sandy soil. This means that the increase in CO 2              |                           |
| 13 | concentration near the soil's surface will be almost immediate under a coarsely textured soil               |                           |
| 14 | (Fig. 6a), but slightly delayed under a finely texture soil. Finally, fine-textured soils have slower       | Commente           |
| 15 | infiltration rates (Hillel, 1998), delaying the exposure of more deeply distributed roots and               | (                         |
| 16 | microbes to increased moisture availability. While this effect may not directly impact the SS               |                           |
| 17 | assumption, it would lead to greater time lags between precipitation pulses and $R_{soil}$ peaks.           |                           |
| 18 | These findings have important implications for studies that rely on the SS assumption to                    |                           |
| 19 | predict subsurface soil CO 2 production. The SS assumption may be sufficient for systems         |                           |
| 20 | defined by coarse-textured soils, but it may lead to erroneous conclusions if applied to fine-              |                           |
| 21 | textured soils, especially at the very short-term scale (e.g. diurnal $R_{soil}$ ) during times of          |                           |
| 22 | precipitation. Our simulation experiments made the simplifying assumption that soil texture is              | Commente
text to addre |
| 23 | constant with depth, but in many ecosystems, texture may vary greatly with depth (Ogle et al.,              | (                         |

Commented [E35]: REVIEWER 2, specific comment #25. New text to address comment.

**Commented [RE36]:** REVIEWER 1 General Comment #1, third specific change. New text.

**Commented [E37]:** REVIEWER 2. Specific comment #26. New text to address comment.

| 1 | 2004). An important next step is to extend the simulations to explore the impacts of depth-   |
|---|-----------------------------------------------------------------------------------------------|
| 2 | varying soil texture on SS versus NSS conditions. The DETECT model can easily accommodate     |
| 3 | such modifications; allowing soil texture to vary by depth would have a direct effect on soil |
| 4 | water content, which is simulated outside of DETECT using HYDRUS (Chou et al., 2008;          |
| 5 | Šimůnek et al., 2008; Piao et al., 2009), that can accommodate such depth variation.          |

**6 4.3 Effect of varying the timing or frequency of precipitation**

7 Unlike soil texture, varying the timing, frequency, and magnitude of precipitation resulted in predicted Rsoil that was almost identical under SS and NSS assumptions, both at the growing 8 9 season and daily time-scales (Fig. S2). We had anticipated that such changes in the precipitation regime would impact SS conditions via impacts on soil air-filled porosity and potentially by 10 11 changing the covariance between soil water and soil temperature, both of which affect soil CO2 diffusivity (e.g., see Eqn 2). We did not explore, however, the effect of decreasing the frequency 12 while simultaneously increasing the magnitude of individual pulses. We hypothesize that this 13 latter scenario could produce more exaggerated or extended NSS conditions given that large rain 14 events would infiltrate deeper, reducing CO2 diffusivity across greater soil depths, thus slowing 15 the transport of more deeply derived CO2. Increasing the number of small events, as done in the 16 *P-FM* scenario, would generally confine water inputs to shallow layers, from which CO2 has 17 shorter distances to travel to reach the surface, creating less opportunity for Rsoil to exhibit NSS 18 behavior. 19

**20 **4.4 Effect of antecedent conditions**

| 21 The inclusion or exclusion of antecedent soil moisture and tem | perature effects on CO 2 |
|-------------------------------------------------------------------|-------------------------------------|
|-------------------------------------------------------------------|-------------------------------------|

- 22 production rates had little to no impact on the balance between SS versus NSS behavior of  $R_{soil}$ .
- However, incorporating antecedent effects generally increased the magnitude of  $R_{soil}$  as

**Commented [RE38]:** REVIEWER 1, General Comment #1 / specific comment #18. As per these comments, the paragraph that followed this paragraph has been deleted.

**Commented [RE39]:** REVIEWER 2, specific comment #21. Section 4.4 remains unchanged, but I am highlighting it here as a place in the manuscript which addressed this comment raised by the reviewer.

| 1  | microbial respiration was stimulated more during the initial onset of the main precipitation        |
|----|-----------------------------------------------------------------------------------------------------|
| 2  | period when antecedent effects were considered (Fig. 2b vs Fig 2a, day 218, blue line). This is     |
| 3  | expected because the instantaneous response of microbes to a rain event is expected to be greater   |
| 4  | following a dry period compared to during a wet period (Xu et al., 2004; Sponseller, 2007; Cable    |
| 5  | et al., 2008; Thomas et al., 2008; Cable et al., 2013). These dynamics are incorporated in the      |
| 6  | antecedent version of the models when the parameter corresponding to the interaction between        |
| 7  | current and antecedent soil water content is negative (e.g., $\alpha_3$ , Table 1). Secondly, root  |
| 8  | respiration was greatly enhanced following the end of this period of precipitation (Fig. 2b vs Fig. |
| 9  | 2a, days ~230-250, green line), despite there being little precipitation after day 230 (Fig. 2b).   |
| 10 | This likely occurred because our DETECT model assumed that soil water over relatively longer        |
| 11 | time periods (past 1-2 weeks, Eqn. 12) affects current root respiration rates. This partly reflects |
| 12 | the mechanism that roots are able to take up more soil water that has infiltrated to deeper depths  |
| 13 | (Cable et al., 2013). The microbes, however, are coupled to past conditions over comparatively      |
| 14 | short time periods (a couple days).                                                                 |
| 15 | The importance and benefit of including antecedent terms for modelling soil respiration             |
| 16 | or ecosystem respiration has been well documented (Cable et al., 2013; Barron-Gafford et al.,       |
| 17 | 2014; Ryan et al., 2015). Thus, we encourage future studies to include influences of past           |
| 18 | conditions when modelling subsurface and surface CO2 fluxes. Fortunately, our simulation            |
| 19 | experiments suggest that the lagged responses of microbial and root respiration to soil moisture    |
| 20 | and temperature do not have a notable impact on the SS assumption.                                  |
|    |                                                                                                     |

**21 4.5 Comparison of modelled soil CO2 with data**

22 The good agreement between modeled and observed soil CO2 concentrations—particularly when

23 including antecedent effects—was very encouraging because the DETECT model was not

| 1  | rigorously tuned or calibrated to fit data on soil [CO 2 ] or ecosystem CO 2 fluxes ( $R_{eco}$ ) (Figs. 5, |
|----|-----------------------------------------------------------------------------------------------------------------------------------|
| 2  | S4). However, there remained discrepancies between the predicted and observed CO 2 fluxes,                             |
| 3  | particularly after rain events. These discrepancies could be an artifact of the input data used to                                |
| 4  | calculate $CO_2$ production (i.e., the source term). Some parameter values were drawn from the                                    |
| 5  | literature and others were estimated by fitting a non-linear regression model to data. For                                        |
| 6  | example, the parameters describing the current and antecedent soil water content effects ( $\alpha$ 's)                           |
| 7  | were obtained by fitting a non-linear model to $R_{eco}$ data (Ryan et al., 2015). While measured $R_{eco}$                       |
| 8  | represents both root respiration and microbial respiration contributions, it also reflects                                        |
| 9  | aboveground respiration, which is not currently treated in the DETECT model. Moreover, we                                         |
| 10 | made further assumptions about how the $R_{eco}$ parameter estimates translate to component                                       |
| 11 | processes (root and microbial responses), and we relied on literature information about how                                       |
| 12 | microbes and roots respond to precipitation events (e.g., the timing, magnitude, and lags). Future                                |
| 13 | studies could rigorously fit the DETECT model to field data, such as observations of $R_{soil}$ , soil                            |
| 14 | CO 2 concentrations, and 13 C isotope fluxes. Using a Bayesian methodology to do this would                 |
| 15 | allow one to incorporate multiple data sets to inform all parameters in DETECT.                                                   |
| 16 | 4.6 Non-steady state model of soil CO 2 transport and production                                                       |
| 17 | An important contribution of this this study was the development of a non-steady state (NSS)                                      |
| 18 | model of soil CO 2 transport and production (the DETECT model version 1.0), which is                                   |
| 19 | particularly useful for systems that may frequently experience NSS conditions. Other comparable                                   |
| 20 | NSS models exist (e.g., Šimůnek and Suarez, 1993; Fang and Moncrieff, 1999; Hui and Luo,                                          |
| 21 | 2004), but they generally treat the production (source) terms-root/rhizosphere respiration and                                    |
| 22 | microbial decomposition of soil organic matter-simplistically, and accompanying model code                                        |

23 is not available. Our DETECT v1.0 model includes more detailed submodels for the production

| 1  | terms, inspired by recent studies (E.g. Lloyd and Taylor, 1994; Pendall et al., 2003; Davidson et             |
|----|---------------------------------------------------------------------------------------------------------------|
| 2  | al., 2012; Todd-Brown et al., 2012; Carrillo et al., 2014a); in contrast to these studies, which              |
| 3  | essentially described models for "bulk" soil, we applied the CO2 production models to every                   |
| 4  | depth increment. Additionally, we have provided model code, implemented in Matlab (see Code                   |
| 5  | Availability section), with the goal of making the DETECT model, and ability to accommodate                   |
| 6  | NSS conditions, more accessible to potential users.                                                           |
| 7  | Future versions of DETECT could include other characteristics of soil CO 2 production              |
| 8  | and transport not included in v1.0. These include: (1) a transport process that simulates the                 |
| 9  | physical displacement of CO 2 in the soil following a precipitation event; (2) alternative options |
| 10 | for some of the functions used, for example there are a number of ways of estimating soluble soil             |
| 11 | C from soil organic C and soil water content (equation 7); (3) estimation of the parameters and               |
| 12 | their associated uncertainties using formal methods (e.g. MCMC) that rely on measurements of                  |
| 13 | C stocks and C fluxes; (4) quantification of the uncertainty of the model outputs (soil CO 2       |
| 14 | concentration, soil respiration) by propagation of uncertainty from the parameters; (5) coupling              |
| 15 | DETECT with a dynamic soil C model in order for the C SOM pools to be dynamic rather than          |
| 16 | prescribed independently of DETECT.                                                                           |
|    |                                                                                                               |

**5. Conclusions 17**

Determining the conditions under which steady-state (SS) assumptions are appropriate for 18 modeling soil CO2 production, transport, and efflux is crucial for accurately modeling the 19 contribution of soils to the carbon cycle. We found that soil texture exerted the greatest control 20 21 over whether SS assumptions are appropriate. When the soil at a site is coarse (60% or more sand), SS assumptions appeared to be appropriate, and one could apply a simpler, more 22 computationally efficient SS model, such as SS-DETECT (see also Amundson et al., 1998). As 23

Commented [E40]: REVIEWER 1 specific comment #4. New text to address this comment.

**Commented [RE41]:** REVIEWER 1 specific comment #7. New text to address the comment.

Commented [E42]: REVIEWER 1 specific comment #14. New text to address the comment.

[revised manuscript text omitted]

- 33
- 34
- 57
- 35
- 36
- 37
- 38
- 39